# Modeling and Optimization Trade-off in Meta-learning

**Katelyn Gao**
Intel Labs

**Ozan Sener**
Intel Labs

## Abstract

By searching for shared inductive biases across tasks, meta-learning promises to accelerate learning on novel tasks, but with the cost of solving a complex bilevel optimization problem. We introduce and rigorously define the trade-off between accurate modeling and optimization ease in meta-learning. At one end, classic meta-learning algorithms account for the structure of meta-learning but solve a complex optimization problem, while at the other end domain randomized search (otherwise known as joint training) ignores the structure of meta-learning and solves a single level optimization problem. Taking MAML as the representative meta-learning algorithm, we theoretically characterize the trade-off for general non-convex risk functions as well as linear regression, for which we are able to provide explicit bounds on the errors associated with modeling and optimization. We also empirically study this trade-off for meta-reinforcement learning benchmarks.

## 1   Introduction

Arguably, the major bottleneck of applying machine learning to many practical problems is the cost associated with data and/or labeling. While the cost of labeling and data makes supervised learning problems expensive, the high sample complexity of reinforcement learning makes it downright inapplicable for many practical settings. Meta-learning (or in general multi-task learning) is designed to ease the sample complexity of these methods. It has had success stories on a wide range of problems including image recognition and reinforcement learning [14].

In the classical risk minimization setting, for a task $\gamma$, the learner solves the problem

$$\min_{\boldsymbol{\theta}} \mathcal{R}(\boldsymbol{\theta}; \gamma) \triangleq \mathbb{E}_{\xi} \left[ \hat{\mathcal{R}}(\boldsymbol{\theta}, \xi; \gamma) \right] \tag{1}$$

where $\mathcal{R}(\boldsymbol{\theta}; \gamma)$ is the risk function which the learner can only access via noisy evaluations $\hat{\mathcal{R}}(\boldsymbol{\theta}, \xi; \gamma)$. Meta-learning, or 'learning to learn' [24], makes the observation that if the learner has access to a collection of tasks sampled from a distribution $p(\gamma)$, it can utilize an offline meta-training stage to search for shared inductive biases that assist in learning future tasks from $p(\gamma)$. Under the PAC framework, Baxter [2] shows that given sufficiently many tasks and data per task during meta-training, there are guarantees on the generalization of learned biases to novel tasks.

Specifically, consider an optimization algorithm $\text{OPT}(\gamma, \boldsymbol{\theta}_{meta})$ which solves the problem of meta-test task $\gamma$ using the meta solution $\boldsymbol{\theta}_{meta}$. This meta solution is typically a policy initialization for reinforcement learning or shared features for supervised learning. However, it can be any useful knowledge which can be learned in the meta-training stage. The family of meta-learning methods solve, where in practice $\text{OPT}$ is approximated by $\hat{\text{OPT}}$ that uses $N$ calls to an oracle [1];

$$\min_{\boldsymbol{\theta}} \mathcal{R}^{meta}(\boldsymbol{\theta}_{meta}) \triangleq \mathbb{E}_{\gamma \sim p(\gamma), \xi} \left[ \hat{\mathcal{R}}(\text{OPT}(\gamma, \boldsymbol{\theta}_{meta}), \xi; \gamma) \right] \tag{2}$$

This setting is intuitive and theoretically sound. However, it corresponds to a complicated bilevel optimization problem. Bilevel optimization is a notoriously hard problem and even the case of a well-behaved inner problem (e.g. linear program as OPT) can be NP-hard [15] in the general case. Hence, one can rightfully ask, is it feasible to solve the meta problem in (2)? This question is rather more important for the case of reinforcement learning as even solving the empirical-risk minimization in (1) has prohibitively high sample complexity.

Meta-learning proposes to accurately use the structure of the problem by introducing a very costly optimization problem. One obvious question is, *can we trade off modeling accuracy for computational ease?* Unfortunately, there is no general principled approach for controlling this trade-off as it requires understanding domain specific properties of the meta problem. Instead, we focus on the case of meta-information $\boldsymbol{\theta}_{meta}$ as the initialization of an iterative optimizer for meta-test task $\gamma$, $\boldsymbol{\theta}_{meta} = \boldsymbol{\theta}_\gamma^0$ and drop the subscript *meta* as it is clear from the context. This covers many existing algorithms, including MAML [9], which is able to approximate any learning algorithm when combined with deep networks [8]. For this case of meta-learning the initialization, a simple and direct alternative would be solving the pseudo-meta problem

$$\min_{\boldsymbol{\theta}} \mathcal{R}^{drs}(\boldsymbol{\theta}) \triangleq \mathbb{E}_{\gamma \sim p(\gamma), \xi} \left[ \hat{\mathcal{R}}(\boldsymbol{\theta}, \xi; \gamma) \right] \tag{3}$$

We call this domain randomized search (DRS) since it corresponds to the domain randomization method from Tobin et al. [26] and it does direct search over a distribution of domains (tasks). [2]

It might not be clear to the reader how DRS solves meta-learning. It is important to reiterate that this is only the case if the meta-learned information is an initialization. However, we believe an approximate form of meta-learning without bilevel structure can be found in other cases with the help of domain knowledge. In this paper, we rigorously prove that DRS is an effective meta-learning algorithm for learning an initialization by showing DRS decreases sample complexity during meta-testing.

These two approaches correspond to the two extremes of the modeling and optimization trade-off. Meta-learning corresponds to an *accurate modeling* and *a computationally harder optimization*, whereas DRS corresponds to a *less accurate modeling* and *computationally easier optimization*. In this paper, we try to understand this trade-off and specifically attempt to answer the following question; *Given a fixed and finite budget for meta-training and meta-testing, which algorithm is more desirable?* In order to answer this question, we provide a collection of theoretical and empirical answers. Taking MAML to be the representative meta-learning algorithm;

- We empirically study this trade-off in meta-reinforcement learning (Section 2).
- We analyze the sample complexity of DRS and MAML for a general non-convex function, and illustrate the interplay of the modeling error and optimization error (Section 3).
- We theoretically analyze the meta-linear regression case, which is fully tractable, and explicitly characterize the trade-off with simulations that confirm our results (Section 4).

## 1.1 Formulation, Background and Summary of Results

We are interested in a distribution $p(\gamma)$ of problems with risk functions $\mathcal{R}(\boldsymbol{\theta}, \gamma)$ for the task ids $\gamma$. Given a specific task, we assume we have access to $2N$ i.i.d. samples from the risk function. This corresponds to sampling data-points and labels for the supervised learning problem, and sampling episodes with their corresponding reward values for reinforcement learning[3]. Classical empirical risk minimization separately solves for each $\gamma$

$$\min_{\boldsymbol{\theta}} \tilde{\mathcal{R}}(\boldsymbol{\theta}; \gamma) \triangleq \frac{1}{2N} \sum_{i=1}^{2N} \hat{\mathcal{R}}(\boldsymbol{\theta}, \xi_i; \gamma) \tag{4}$$

where $\xi_i$ are datapoints for supervised learning and episodes for RL. Empirical risk minimization for MAML can be defined over $M$ meta-training tasks sampled from $p(\gamma)$ as;

$$\min_{\boldsymbol{\theta}} \tilde{\mathcal{R}}^{maml}(\boldsymbol{\theta}) \triangleq \frac{1}{M} \sum_{j=1}^{M} \frac{1}{N} \sum_{i=1}^{N} \hat{\mathcal{R}}(\hat{\mathrm{OPT}}(\gamma_j, \boldsymbol{\theta}, N), \xi_{i,j}; \gamma_j) \tag{5}$$

Meta-learning needs samples for both the inner optimization and outer meta problem; usually half of the samples are used for each. Empirical risk minimization for DRS is

$$\min_{\boldsymbol{\theta}} \tilde{\mathcal{R}}^{drs}(\boldsymbol{\theta}) \triangleq \frac{1}{M} \sum_{i=1}^{M} \frac{1}{2N} \sum_{i=1}^{2N} \hat{\mathcal{R}}(\boldsymbol{\theta}, \xi_{i,j}; \gamma_j) \tag{6}$$

Denote the solutions of problems (2) and (3) by $\boldsymbol{\theta}^\star_{maml}$ and $\boldsymbol{\theta}^\star_{drs}$, which are the minimum population risk solutions. We call the risk of these solutions $\mathcal{R}^{drs}(\boldsymbol{\theta}^\star_{drs})$ and $\mathcal{R}^{maml}(\boldsymbol{\theta}^\star_{maml})$ the **modeling error** as they are the best each method can get with the best optimizer and infinite data and computation. However, we only have access to empirical risk in (5) and (6), as well as a finite computation budget for meta-training, $T^{tr}$. Hence, instead of optimal solutions, we have solutions $\boldsymbol{\theta}^{T^{tr}}_{maml}$ and $\boldsymbol{\theta}^{T^{tr}}_{drs}$. We call the difference between these empirical solutions and $\boldsymbol{\theta}^\star_{maml}$ and $\boldsymbol{\theta}^\star_{drs}$ the **optimization error**.

We expect the optimization error of MAML to be significantly higher as bilevel optimization is much harder. More specifically, for general non-convex risk functions, in order for stochastic gradient descent (SGD) to reach an $\epsilon$-stationary point, DRS needs $\mathcal{O}(1/\epsilon^4)$ samples [12] while MAML needs $\mathcal{O}(1/\epsilon^6)$ [6]. Moreover, MAML uses half of its samples for the inner optimization; hence, the effective number of samples is reduced resulting in additional optimization error. The optimization error strongly depends on the sample budget ($N$ and $M$) which can be afforded. As more samples will decrease optimization errors for both methods but with different rates. On the other hand, we know MAML has lower modeling error as it explicitly uses the problem geometry.

The key question is the trade-off of the modeling error versus the optimization error as a function of $N$ and $M$ since it characterizes which algorithm is better for a given problem and budget. We look at this trade-off in two different settings.

- **General Non-Convex Case:** We study this trade-off empirically for meta-reinforcement learning (Section 2) and theoretically for general non-convex risk functions (Section 3). Our empirical results suggest that this trade-off is rather nuanced and problem dependent. Our theoretical results shed light on these peculiarities and describe the trade-off.
- **Meta-linear regression:** (Section 4). Empirical risk can be minimized analytically, enabling us to directly compare optimization error and modeling error. Results suggest that there are cut-off values for $N$ and $M$ that determine when the different methods are desirable.

## 2   Trade-offs in Meta-Reinforcement Learning

We study the trade-off empirically in meta-reinforcement learning in this section. We compare DRS and MAML on on a wide range of meta-RL benchmarks used in the literature. We consider 1) four robotic locomotion environments introduced by Finn et al. [9] and Rothfuss et al. [21], with varying reward functions (Half Cheetah Rand Vel, Walker2D Rand Vel) or varying system dynamics (Hopper Rand Params, Walker2D Rand Params) and 2) four manipulation environments from MetaWorld [33] with varying reward functions (ML1-Push, ML1-Reach) or changing manipulation tasks and varying reward functions (ML10, ML45). [4] All environments utilize the Mujoco simulator [27].

We make two comparisons: 1) ProMP [21], which combines MAML with PPO [23], vs. DRS combined with PPO (DRS+PPO) 2) TRPO-MAML [9, 21], which combines MAML with TRPO [22], vs. DRS combined with TRPO (DRS+TRPO). Meta-training is controlled such that all algorithms utilize an equal number of steps generated from the simulator for each environment.

During evaluation, we first sample a set of meta-test tasks. For each meta-test task, starting at a trained policy, we repeat the following five times: generate a small number of episodes from the current policy and update the policy using the policy gradient algorithm from the inner optimization of ProMP and TRPO-MAML. We compute the average episodic reward after $t$ updates, for $t = 0, 1, \ldots, 5$. These statistics are then averaged over all sampled tasks. We evaluate the policies learned by all algorithms at various checkpoints during meta-training, in total five evaluations with different random seeds. For each checkpoint and test update, we compute the probability of DRS being better than MAML using the one-sided Welch t-test. We plot the filled contour plot of the probabilities with respect to the meta-training checkpoint and number of test updates in Figures 1 and 2. We include details of the experimental protocol, Welch t-test, and additional plots of the reward in Appendix A.

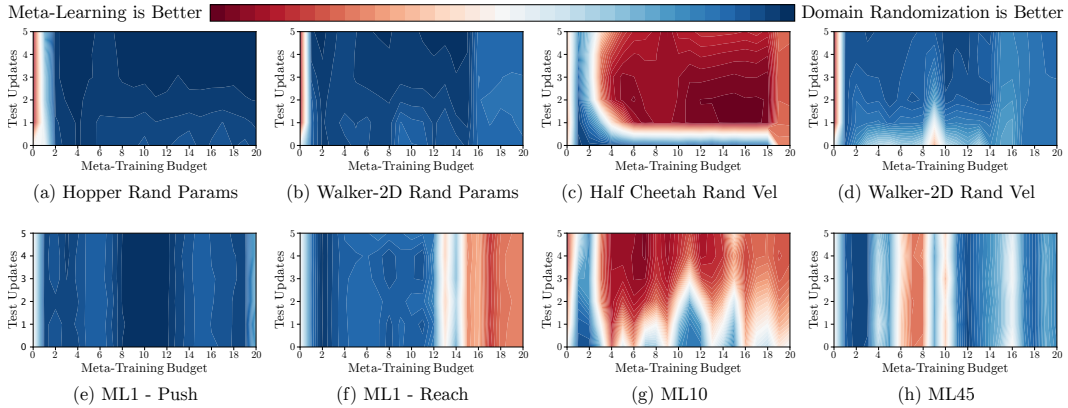

(a) Hopper Rand Params     (b) Walker-2D Rand Params     (c) Half Cheetah Rand Vel     (d) Walker-2D Rand Vel

(e) ML1 - Push     (f) ML1 - Reach     (g) ML10     (h) ML45

Figure 1: DRS+PPO vs. ProMP: Probability that the first method is better than the second.

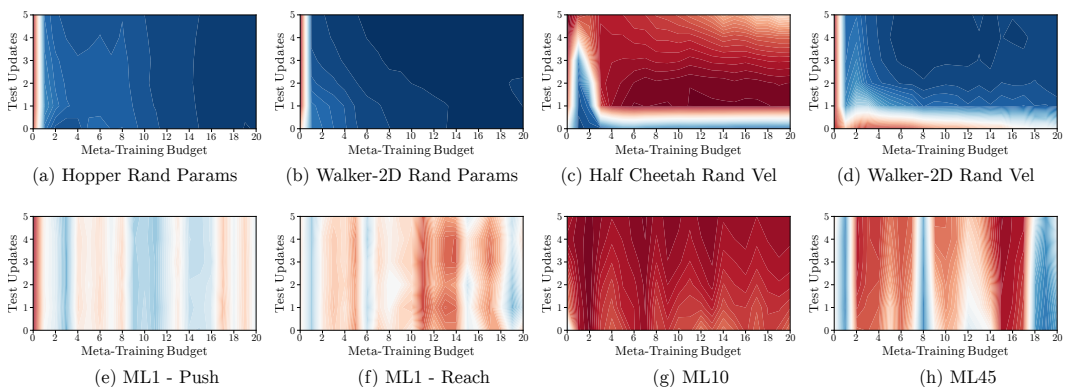

(a) Hopper Rand Params     (b) Walker-2D Rand Params     (c) Half Cheetah Rand Vel     (d) Walker-2D Rand Vel

(e) ML1 - Push     (f) ML1 - Reach     (g) ML10     (h) ML45

Figure 2: DRS+TRPO vs. TRPO-MAML: Probability that the first method is better than the second.

For the majority of environments, DRS is either better than or comparable to MAML, *(see Figure 1-(a,b,d,e,f) and Figure 2-(a,b,d,e,f))*. For two environments, specifically *Half Cheetah Rand Vel* and *ML10* in Figure 1-(c,g)&2-(c,g), MAML outperforms DRS for larger sample sizes and at least one test update. This confirms our thesis as the higher optimization error of MAML requires a larger sample size for successful learning, dominating its trade-off with modeling error for smaller sample sizes. Another surprising result is the case of ML45 (in Figure 1-(h)& 2-(h)) where as the sample size increases it alternates between the cases where DRS is better and MAML is better. We conjecture that this is because of the greater variance in meta training arising from the higher diversity in tasks.

Another interesting observation is that in the majority of environments, MAML fares better when combined with TRPO than when combined with PPO. This validates the importance of optimization as both TRPO and PPO use exactly the same model and are expected to behave similarly from a statistical perspective. We suspect that this behavior is due to the difference in how the trust region is used for optimization. In TRPO-MAML the constraint on the KL divergence between the current policy and updated policy is strictly enforced, while in ProMP it is transformed into a Lagrangian penalty and thus may not actually be satisfied. Satisfaction of the constraint is more helpful for MAML, whose empirical gradients generally have greater variance than those of DRS.

In conclusion, when the sample complexity is high and available sample size budget is low, DRS is the preferred method. MAML is only effective for large data sets, only in some problems. It is important for practitioners to understand the sample complexity of their problems before choosing which method to apply. In the next section, we provide theoretical analysis to clarify this phenomenon.

## 3    Trade-off through the Lens of Optimization Behavior

In this section, we analyze the sample complexity of meta-training and meta-test of MAML and DRS. In particular, we consider the interplay of meta-training and meta-testing for a complete analysis. Specifically, for MAML $\text{OPT}(\gamma, \boldsymbol{\theta})$ is one step of gradient descent on the task risk function $\mathcal{R}(\boldsymbol{\theta}; \gamma)$

starting at $\boldsymbol{\theta}$ with learning rate $\alpha$. The objectives of DRS and MAML are

$$\mathcal{R}^{drs}(\boldsymbol{\theta}) \triangleq \mathbb{E}_\gamma[\mathcal{R}(\boldsymbol{\theta};\gamma)], \qquad \mathcal{R}^{maml}(\boldsymbol{\theta};\alpha) \triangleq \mathbb{E}_\gamma[\mathcal{R}(\boldsymbol{\theta} - \alpha\nabla_{\boldsymbol{\theta}}\mathcal{R}(\boldsymbol{\theta};\gamma);\gamma)]. \qquad (7)$$

We analyze the trade-off between optimization error and modeling error of DRS and MAML for smooth non-convex risk functions from a sample complexity perspective. We denote the modeling errors by $\Lambda$ and, as in Section 1, define them as the expected risks at the globally optimal values of the corresponding objectives. Specifically, $\Lambda^{drs} = \mathcal{R}^{drs}(\boldsymbol{\theta}^\star_{drs})$ and $\Lambda^{maml}(\alpha) = \mathcal{R}^{maml}(\boldsymbol{\theta}^\star_{maml};\alpha)$.

Suppose that we have access to task stochastic gradient oracles and task stochastic Hessian oracles. During meta-training, DRS and MAML optimize $\mathcal{R}^{drs}(\boldsymbol{\theta})$ and $\mathcal{R}^{maml}(\boldsymbol{\theta};\alpha)$ using SGD for $T^{tr}$ steps. During meta-testing for a task $\gamma$, both carry out classical risk minimization (1) using SGD for $T^{te}$ steps, with the meta-training results $\boldsymbol{\theta}^{T^{tr}}_{drs}$ and $\boldsymbol{\theta}^{T^{tr}}_{maml}(\alpha)$ as warm starts. The metric we use is the Euclidean norm of the gradient at meta-testing since the best we can hope for a non-convex function is first-order stationarity.

Before we proceed with our analysis we set the notation and make some mild regularity assumptions. Consider task stochastic gradient oracles $\mathbf{g}(\cdot,\xi;\gamma)$ such that $\mathbb{E}_\xi[\mathbf{g}(\boldsymbol{\theta},\xi;\gamma)] = \nabla_{\boldsymbol{\theta}}\mathcal{R}(\boldsymbol{\theta};\gamma)$, and task stochastic Hessian oracles $\mathbf{h}(\cdot,\xi;\gamma)$ such that $\mathbb{E}_\xi[\mathbf{h}(\boldsymbol{\theta},\xi;\gamma)] = \nabla^2_{\boldsymbol{\theta}}\mathcal{R}(\boldsymbol{\theta};\gamma)$. We assume **B1:** The risk functions are nonnegative and bounded, $0 \leq \mathcal{R}(\boldsymbol{\theta};\gamma) \leq \Delta$ for all $\boldsymbol{\theta}$ and $\gamma$. **B2:** The risk functions are uniformly Lipschitz and smooth $\|\mathcal{R}(\boldsymbol{\theta}^1;\gamma)-\mathcal{R}(\boldsymbol{\theta}^2;\gamma)\|/\|\boldsymbol{\theta}^1-\boldsymbol{\theta}^2\| \leq L$ and $\|\nabla_{\boldsymbol{\theta}}\mathcal{R}(\boldsymbol{\theta}^1;\gamma)-\nabla_{\boldsymbol{\theta}}\mathcal{R}(\boldsymbol{\theta}^2;\gamma)\|/\|\boldsymbol{\theta}^1-\boldsymbol{\theta}^2\| \leq \mu$ for all $\gamma$. **B3:** The task stochastic gradient oracles have bounded variance, $\mathrm{tr}(\mathrm{Var}_\xi(\mathbf{g}(\cdot,\xi;\gamma))) \leq V^d$ for all $\gamma$, and the gradients of the risk functions have bounded variance, $\mathrm{tr}(\mathrm{Var}_\gamma(\nabla_{\boldsymbol{\theta}}\mathcal{R}(\boldsymbol{\theta};\gamma))) \leq V^t$. **B4:** the Hessians of the risk functions are Lipschitz $\|\nabla^2_{\boldsymbol{\theta}}\mathcal{R}(\boldsymbol{\theta}^1;\gamma)-\nabla^2_{\boldsymbol{\theta}}\mathcal{R}(\boldsymbol{\theta}^2;\gamma)\|/\|\boldsymbol{\theta}^1-\boldsymbol{\theta}^2\| \leq \mu^H$ for all $\gamma$. **B5:** The task stochastic Hessian oracles have bounded variance $\mathbb{E}_\xi\big[\big\|\mathbf{h}(\boldsymbol{\theta},\xi;\gamma) - \nabla^2_{\boldsymbol{\theta}}\mathcal{R}(\boldsymbol{\theta};\gamma)\big\|^2\big] \leq V^h$.

We state the sample complexity of DRS and MAML in Theorems 1 and 2, respectively. Proofs can be found in Appendix B.

**Theorem 1.** *Suppose that during each iteration of meta-training DRS, $M$ tasks are sampled each with $2N$ calls to their gradient oracles, and during each iteration of meta-testing, $N$ calls are made to the gradient oracle. Assume **B1-3**. Then, in expectation over the meta-test task $\gamma$,*

$$\sum_{t=0}^{T^{tr}-1} \left\|\nabla_{\boldsymbol{\theta}}\mathcal{R}^{drs}(\boldsymbol{\theta}^t)\right\|^2 + \sum_{t=0}^{T^{te}-1} \|\nabla_{\boldsymbol{\theta}}\mathcal{R}(\boldsymbol{\theta}^{t+T^{tr}};\gamma)\|^2 \leq \sqrt{0.5(\Delta + \Lambda^{drs})(C^{drs}_{tr}T^{tr} + C^{drs}_{te}T^{te})}$$

*where $C^{drs}_{tr} = \mu\left(L^2 + \frac{V^t}{M} + \frac{V^d}{2NM}\right)$ and $C^{drs}_{te} = \mu\left(L^2 + \frac{V^d}{N}\right)$.*

Before we continue with the analysis of MAML, we first discuss the implications of Theorem 1. Let us examine the bound when $T^{te}$ increases by one and $T^{tr}$, as well as all other variables, are fixed. The change in the left side is the meta-testing gradient after $T^{te}$ iterations, whereas that in the right side is approximated by its gradient wrt $T^{te}$, behaving approximately as $\mathcal{O}(\sqrt{\Delta+\Lambda^{drs}}/\sqrt{C^{drs}_{tr}T^{tr}+C^{drs}_{te}T^{te}})$. Hence, with more meta-training, we get closer to the stationary point of the meta-test task $\gamma$ with the same number of meta-test iterations. In other words our result shows that DRS, although it ignores the meta-learning problem structure as discussed in Section 1, provably solves the problem of meta-learning the initialization of an iterative optimization problem under sensible assumptions.

**Theorem 2.** *Following Algorithm 1 of Fallah et al. [6], suppose that during each iteration of meta-training MAML, $M$ tasks are sampled each with $2N$ calls to their gradient oracles, half of which are used in the inner optimization, and $D$ calls to their Hessian oracles. During each iteration of meta-testing, $N$ calls are made to the gradient oracle. Assume **B1-5**, $\alpha \in [0, 1/6\mu]$, and $D \geq 2\alpha^2 V^h$. Then, in expectation over the meta-test task $\gamma$,*

$$\sum_{t=0}^{T^{tr}-1} \left\|\nabla_{\boldsymbol{\theta}}\mathcal{R}^{maml}(\boldsymbol{\theta}^t;\alpha)\right\|^2 + \sum_{t=0}^{T^{te}-1} \|\nabla_{\boldsymbol{\theta}}\mathcal{R}(\boldsymbol{\theta}^{t+T^{tr}};\gamma)\|_2^2$$
$$\leq T^{tr}\alpha\mu(1+\alpha\mu)^2 L\sqrt{V^d/N} + \sqrt{0.5(\Delta + \Lambda^{maml}(\alpha) + \alpha L^2)(C^{maml}_{tr}T^{tr} + C^{maml}_{te}T^{te})}$$

*where $C^{maml}_{tr} = (4\mu + 2\mu^H\alpha L)\left(\left(2 + \frac{40}{M}\right)(1+\alpha\mu)^2 L^2 + \frac{14V^t}{M} + \frac{3V^d(1+\alpha^2\mu^2 M)}{MN}\right)$ and $C^{maml}_{te} = C^{drs}_{te}$.*

We keep the $\alpha$ as a free parameter since it makes the role of modeling error explicit. First consider a similar argument to discussion of Theorem 1; the meta-testing gradient after $T^{te}$ iterations behaves approximately as $\mathcal{O}(\sqrt{\Delta + \Lambda^{maml}}/\sqrt{C_{tr}^{maml}T^{tr} + C_{te}^{maml}T^{te}})$. Hence, unsurprisingly meta-training of MAML also provably improves the sample complexity in meta-testing. Next, if we ignore the modeling error by assuming $\Lambda^{maml}(\alpha) = \Lambda^{drs}$, the bound in Theorem 1 is lower than in Theorem 2 for all $\alpha > 0$. In other words, if the problem has no specific geometric structure that MAML can utilize, DRS will perform better. On the other hand, if the modeling error is dominant, i.e. $\Lambda^{maml}(\alpha) \ll \Lambda^{drs}$, MAML will perform better. For most practical cases, $\Lambda^{maml}(\alpha)$ will be less than $\Lambda^{drs}$, but not significantly. In these cases, the trade-off is governed by the values of $N$ and $M$.

Another important observation is the strong dependence of the sample complexity on oracle variances as well as smoothness constants. This partially explains the problem dependent behavior of DRS and MAML in Section 2. Choosing the right algorithm requires, in addition to the budget for $N$ and $M$, these problem and domain specific information that are often not practical to compute (or estimate). Hence, translating these results to practically relevant decision rules needs future work.

# 4   Trade-offs in Meta-Linear Regression

In this section, we study the linear regression case where we can explicitly characterize the trade-off of optimization error and modeling error. Since the empirical risk minimization problems corresponding to MAML and DRS in linear regression are analytically solvable, the optimization errors only consist of the statistical errors arising from using empirical risk minimization. We first analyze the optimization errors and then discuss the modeling errors of the two approaches. All proofs can be found in Appendix C.

## 4.1   Formal Setup and Preliminaries

For each task $\gamma$, we assume the following data model:

$$y_\gamma = \boldsymbol{\theta}_\gamma^\mathsf{T} \mathbf{x}_\gamma + \epsilon_\gamma, \qquad \epsilon_\gamma \sim (0, \sigma_\gamma^2), \qquad \mathbf{Q}_\gamma = \mathbb{E}[\mathbf{x}_\gamma \mathbf{x}_\gamma^\mathsf{T} \mid \gamma]. \tag{8}$$

where $\epsilon_\gamma$ and $\mathbf{x}_\gamma$ are independent and $\mathbf{x}_\gamma \in \mathbb{R}^p$. We assume the squared error loss;

$$\mathcal{R}(\boldsymbol{\theta}; \gamma) = \frac{1}{2}\mathbb{E}[(y_\gamma - \boldsymbol{\theta}^\mathsf{T} \mathbf{x}_\gamma)^2 \mid \gamma] = \frac{1}{2}\boldsymbol{\theta}^\mathsf{T} \mathbf{Q}_\gamma \boldsymbol{\theta} - \boldsymbol{\theta}_\gamma^\mathsf{T} \mathbf{Q}_\gamma \boldsymbol{\theta} + \frac{1}{2}\boldsymbol{\theta}_\gamma^\mathsf{T} \mathbf{Q}_\gamma \boldsymbol{\theta}_\gamma + \frac{1}{2}\sigma_\gamma^2. \tag{9}$$

The globally optimal (minimum risk) solutions for the DRS and MAML objectives (7) are (*see C.2 for derivations*);

$$\begin{aligned} \boldsymbol{\theta}_{drs}^* &= \mathbb{E}_\gamma[\mathbf{Q}_\gamma]^{-1}\mathbb{E}_\gamma[\mathbf{Q}_\gamma \boldsymbol{\theta}_\gamma] \\ \boldsymbol{\theta}_{maml}^*(\alpha) &= \mathbb{E}_\gamma[(\mathbf{I} - \alpha\mathbf{Q}_\gamma)\mathbf{Q}_\gamma(\mathbf{I} - \alpha\mathbf{Q}_\gamma)]^{-1}\mathbb{E}_\gamma[(\mathbf{I} - \alpha\mathbf{Q}_\gamma)\mathbf{Q}_\gamma(\mathbf{I} - \alpha\mathbf{Q}_\gamma)\boldsymbol{\theta}_\gamma]. \end{aligned} \tag{10}$$

Both solutions can be viewed as weightings of the task parameters $\boldsymbol{\theta}_\gamma$. Since the Hessian of $\mathcal{R}(\boldsymbol{\theta}; \gamma)$ is $\mathbf{Q}_\gamma$, the DRS solution gives greater weight to the tasks whose risk functions have higher curvature, i.e. are more sensitive to perturbations in $\boldsymbol{\theta}$. Compared to the DRS solution, the MAML solution puts more weight on the tasks with lower curvature. As the gradient of $\mathcal{R}(\boldsymbol{\theta}; \gamma)$ is $\mathbf{Q}_\gamma(\boldsymbol{\theta} - \boldsymbol{\theta}_\gamma)$, for tasks with lower curvature one gradient step on the task risk function takes us a smaller fraction of the distance from the current point $\boldsymbol{\theta}$ to the stationary point $\boldsymbol{\theta}_\gamma$; thus, starting from the MAML solution enables faster task adaptation overall if the risks are known exactly.

## 4.2   Bounds on Optimization Error

We consider a finite-sample setting where $M$ tasks, independently sampled from $p(\gamma)$ and $2N$ observations per task, sampled according to the model in (8) are given. We denote the resulting dataset as $\mathcal{D} \equiv (\mathbf{x}_{j,i}, y_{j,i}), j = 1, \ldots, M, i = 1, \ldots, 2N$.

During meta-training, from (6), DRS minimizes the average squared error over all data:

$$\hat{\boldsymbol{\theta}}_{drs} \triangleq \arg\min_{\boldsymbol{\theta}} \sum_{j=1}^{M} \sum_{i=1}^{2N} (y_{j,i} - \boldsymbol{\theta}^\mathsf{T} \mathbf{x}_{j,i})^2. \tag{11}$$

From (5), MAML optimizes for parameters such that, when optimized for each task using SGD on the average squared error over $N$ observations with learning rate $\alpha$, minimizes the average squared error over the remaining $N$ observations of all tasks:

$$\hat{\boldsymbol{\theta}}_{maml}(\alpha) \triangleq \arg\min_{\boldsymbol{\theta}} \sum_{j=1}^{M} \sum_{i=N+1}^{2N} (y_{j,i} - \tilde{\boldsymbol{\theta}}_j(\alpha)^\mathsf{T}\mathbf{x}_{j,i})^2, \tilde{\boldsymbol{\theta}}_j(\alpha) = \boldsymbol{\theta} - \frac{\alpha}{N}\sum_{i=1}^{N}(\mathbf{x}_{j,i}\mathbf{x}_{j,i}^\mathsf{T}\boldsymbol{\theta} - \mathbf{x}_{j,i}y_{j,i})$$

(12)

As a sub-optimality metric, we characterize the distances between the empirical solution and globally optimal solution, $\left\|\hat{\boldsymbol{\theta}}_{drs} - \boldsymbol{\theta}^*_{drs}\right\|$ and $\left\|\hat{\boldsymbol{\theta}}_{maml}(\alpha) - \boldsymbol{\theta}^*_{maml}(\alpha)\right\|$, in terms of the finite sample sizes $M$ and $N$ in Theorem 3&4. This is the error arising from using empirical samples instead of the population statistics, that is, the statistical error.

Before we state the theorems, we summarize the assumptions. **A1:** Bounded Hessian of the task loss, $\|\mathbf{Q}_\gamma\| \leq \beta$. **A2:** Bounded parameter, feature, and error space, $\|\boldsymbol{\theta}_\gamma - \boldsymbol{\theta}^*_{drs}\| \leq \tau$, $\|\boldsymbol{\theta}_\gamma - \boldsymbol{\theta}^*_{maml}(\alpha)\| \leq \tau'$ and $\|\boldsymbol{\theta}_\gamma\|$, $\|\mathbf{x}_{\gamma,i}\|$, and $|\epsilon_{\gamma,i}|$ are finite. **A3:** The distribution of $\mathbf{x}_{\gamma,i}$ conditional on $\gamma$ is **sub-Gaussian** with parameter $K$. In this setting, the following theorems characterize the statistical error for meta linear-regression.

**Theorem 3.** *Suppose that with probability* $1$, ***A1-3*** *holds. Let* $\omega$ *be logarithmic in* $\delta^{-1}$, $M$, *and* $p$, *and define the functions*

$$c_1(\Delta, r, s, \theta) = \|\theta\|\sqrt{2r\Delta} + \sqrt{2s\Delta}, \quad c_2(\Delta) = \beta CK^2\sqrt{p + \Delta}, \quad c_3(\Delta) = \sqrt{\mathrm{tr}(\mathbb{E}_\gamma[\sigma_\gamma^2\mathbf{Q}_\gamma])\Delta},$$

*where* $C$ *is a constant. If* $\lambda_{min}(\mathbb{E}_\gamma[\mathbf{Q}_\gamma]) - \tilde{o}(1) > 0$, *with probability at least* $1 - \delta$, *ignoring higher order terms,* $\left\|\hat{\boldsymbol{\theta}}_{drs} - \boldsymbol{\theta}^*_{drs}\right\|$ *is bounded above by* [5]

$$(\lambda_{min}(\mathbb{E}_\gamma[\mathbf{Q}_\gamma]) - \tilde{o}(1))^{-1}\left(\frac{c_1(\omega, \|\mathrm{Var}_\gamma[\mathbf{Q}_\gamma]\|, \mathrm{tr}(\mathrm{Var}_\gamma[\mathbf{Q}_\gamma\boldsymbol{\theta}_\gamma]), \boldsymbol{\theta}^*_{drs})}{\sqrt{M}} + \frac{\tau c_2(\omega)/\sqrt{2} + c_3(\omega)}{\sqrt{N}}\right)$$

**Theorem 4.** *With the same assumptions as Theorem 3, let* $\mathbf{S}_\gamma(\alpha) = (\mathbf{I} - \alpha\mathbf{Q}_\gamma)\mathbf{Q}_\gamma(\mathbf{I} - \alpha\mathbf{Q}_\gamma)$. *If* $\lambda_{min}(\mathbb{E}_\gamma[\mathbf{S}_\gamma(\alpha)]) - \tilde{o}(1) > 0$, *with probability at least* $1 - \delta$, *ignoring higher order terms,* $\left\|\hat{\boldsymbol{\theta}}_{maml}(\alpha) - \boldsymbol{\theta}^*_{maml}(\alpha)\right\|$ *is bounded above by*

$$(\lambda_{min}(\mathbb{E}_\gamma[\mathbf{S}_\gamma(\alpha)]) - \tilde{o}(1))^{-1}\bigg(\frac{c_1(\omega, \|\mathrm{Var}_\gamma[\mathbf{S}_\gamma(\alpha)]\|, \mathrm{tr}(\mathrm{Var}_\gamma[\mathbf{S}_\gamma(\alpha)\boldsymbol{\theta}_\gamma]), \boldsymbol{\theta}^*_{maml}(\alpha))}{\sqrt{M}}$$
$$+ \frac{(1 + 3\alpha\beta)^2\tau'c_2(\omega) + \sqrt{2}(1 + \alpha\beta)^2c_3(\omega)}{\sqrt{N}}\bigg).$$

Theorems 3&4 show the statistical errors for MAML and DRS scale similarly in terms of rates with respect to $N$ and $M$. However, the constants are significantly different. Compare the coefficients of $1/\sqrt{N}$. The coefficient of $c_2(\omega)$ for DRS is $\tau\sqrt{2}^{-1}\lambda_{min}(\mathbb{E}_\gamma[\mathbf{Q}_\gamma])^{-1}$ and for MAML it is $\tau'(1 + 3\alpha\beta)^2\lambda_{min}(\mathbb{E}_\gamma[\mathbf{S}_\gamma(\alpha)])^{-1}$. When $\alpha\beta \lessapprox 1$, the latter is larger than the former, since we expect that $\tau \approx \tau'$ and the eigenvalues of $\mathbb{E}_\gamma[\mathbf{S}_\gamma(\alpha)]$ to be shrunken compared to those of $\mathbb{E}_\gamma[\mathbf{Q}_\gamma]$. A similar observation holds for the coefficient of $c_3(\omega)$. In other words, the convergence behavior of the MAML estimate has a worse dependence on $N$, indicating that the statistical error for DRS has more favorable behavior.

## 4.3 Modeling Error

Modeling error affects the meta-testing performance of the globally optimal solutions. We expect MAML to perform better as it directly models the meta-learning problem whereas DRS does not. Modeling error by definition depends on the correct model of the world and thus is difficult to characterize without domain knowledge. We study this error in the following theorem assuming that the distribution of tasks is the same for meta-training and meta-testing and discuss its implications for practitioners.

**Theorem 5.** *For meta-test task $\gamma$ and arbitrary $\boldsymbol{\theta}$, let $\tilde{\boldsymbol{\theta}}_\gamma(\alpha)$ be the parameters optimized by one step of SGD using $N$ data points $\mathcal{O}_\gamma$ and learning rate $\alpha$, as in (12). Let $\mathbf{A}_\gamma(\alpha) = \mathbf{S}_\gamma(\alpha) + \alpha^2(\mathbb{E}[\mathbf{x}_{\gamma,i}\mathbf{x}_{\gamma,i}^\mathsf{T}\mathbf{Q}_\gamma\mathbf{x}_{\gamma,i}\mathbf{x}_{\gamma,i}^\mathsf{T}] - \mathbf{Q}_\gamma^3)/N$, where $\mathbf{S}_\gamma(\alpha)$ is defined in Theorem 4. The expected losses before and after optimization, as functions of $\boldsymbol{\theta}$, are*

$$\mathcal{R}^{drs}(\boldsymbol{\theta}) \equiv \mathbb{E}_\gamma\left[\|\boldsymbol{\theta} - \boldsymbol{\theta}_\gamma\|_{\mathbf{Q}_\gamma}^2\right] \quad \text{and} \quad \mathbb{E}_\gamma[\mathbb{E}_{\mathcal{O}_\gamma}[\mathcal{R}(\tilde{\boldsymbol{\theta}}_\gamma(\alpha);\gamma)]] \equiv \mathbb{E}_\gamma\left[\|\boldsymbol{\theta} - \boldsymbol{\theta}_\gamma\|_{\mathbf{A}_\gamma(\alpha)}^2\right],$$

*where we have ignored constants and terms that do not include $\boldsymbol{\theta}$. The former is minimized by $\boldsymbol{\theta}_{drs}^*$. As $N \to \infty$, the latter approaches $\mathcal{R}^{maml}(\boldsymbol{\theta};\alpha)$, is minimized by $\boldsymbol{\theta}_{maml}^*(\alpha)$ and, for $0 < \alpha \leq 1/\beta$, is at most $\mathcal{R}^{drs}(\boldsymbol{\theta}_{drs}^*)$, the minimum expected loss possible before meta-testing optimization.*

Theorem 5 shows that the smaller modeling error of meta-learning indeed translates to improved performance. Meta-learning can utilize the geometry implied by the distribution of the tasks to reduce the expected loss given the ability to optimize at meta-testing.

Combining the results of all theorems, the optimization error is worse for MAML when $\alpha\beta \lessapprox 1$ from Theorem 3&4. DRS does not model the meta structure, hence it has worse modeling error/greater expected loss by Theorem 5. As expected, there is no clear winner in practice as the choice depends on the trade-off between optimization error and modeling error. Next, we show this empirically.

### 4.4 Empirical Results

We carried out simulations to empirically study the trade-off in the linear regression case. We chose the following specific distribution of tasks and data model:

$$\begin{aligned}
y_\gamma &= \boldsymbol{\theta}_\gamma^\mathsf{T}\mathbf{x}_\gamma + \epsilon_\gamma & \epsilon_\gamma &\sim \mathcal{N}(0, \sigma_\gamma^2) & \mathbf{x}_\gamma &\sim \mathcal{N}(0, \mathbf{Q}_\gamma) \\
\boldsymbol{\theta}_\gamma &\sim U([0,2]^p) & \sigma_\gamma^2 &\sim U([0,2]) & \mathbf{Q}_\gamma &= \mathbf{V}\operatorname{diag}(\boldsymbol{\theta}_\gamma)\mathbf{V}^T & \mathbf{V} &\sim U(\mathbb{SO}(p))
\end{aligned} \tag{13}$$

We present the case for $p = 1$; larger $p$ lead to similar qualitative results. We compute approximations of $\mathcal{R}^{drs}(\boldsymbol{\theta})$ (expected loss before meta-testing optimization) as a function of $\boldsymbol{\theta}$ and $\mathbb{E}_\gamma[\mathbb{E}_{\mathcal{O}_\gamma}[\mathcal{R}(\gamma;\tilde{\boldsymbol{\theta}}_\gamma(\alpha))]]$ (expected loss after meta-testing optimization) as a function of $\boldsymbol{\theta}$ and $\alpha$. For various $M$ and $N$, we generate a collection of datasets $\mathcal{D}$; for each $\mathcal{D}$, over a grid of values for $\alpha \in [0,1]$, we compute the corresponding DRS and MAML estimates and, using the aforementioned functions, calculate whether the MAML estimate has lower expected loss than the DRS estimate before and after meta-testing optimization. Figure 3 shows contour plots of the fraction of the datasets for which the MAML estimate has lower expected loss before meta-testing optimization (left three figures) and after (right three figures), for several values of $\alpha$. The axes are the number of tasks ($M$) and the data set size for meta-testing optimization ($N$).

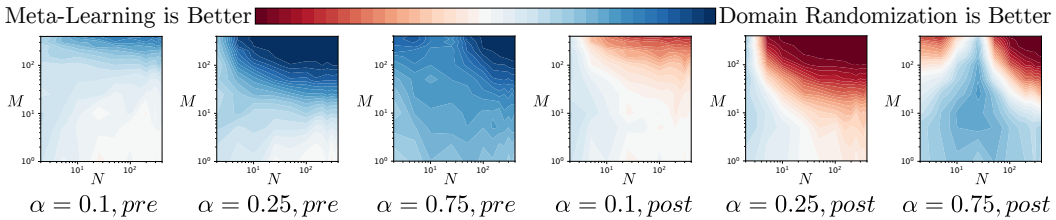

Figure 3: $p = 1$: Contour plots of the probability that the MAML estimate has lower expected loss than the DR estimate before meta-testing optimization (pre) and after (post). The axes are the number of tasks ($M$) and the number of data points used for meta-testing optimization ($N$), and $\alpha$ is fixed.

From Figure 3, for very high $M$ and $N$, the DRS estimate has lower expected loss before meta-testing optimization and the MAML estimate has lower expected loss after meta-testing optimization with very high certainty. This is expected as asymptotically meta-learning is expected to work well by Theorem 5; the practical question of interest is the finite sample case.

From Figure 3-post, we observe that for small $M$ and $N$, the probability that the MAML estimate has lower expected loss after meta-testing optimization can be substantially lower than $0.5$, i.e. the DRS estimate is superior. We conclude that in this case the increased optimization error from MAML dominates the decreased modeling error. Hence, unless the geometry is strongly skewed, DRS is desirable for smaller datasets in meta-linear regression.

# 5 Related Work

Recent work on few-shot image classification has shown that features from learning a deep network classifier on a large training set combined with a simple classifier at meta-testing may outperform many meta-learning algorithms [32, 4, 25]; a similar observation has been made for few-shot object detection [31]. Packer et al. [17] show that DRS outperforms RL$^2$ [5] on simple reinforcement learning environments where tasks correspond to different system dynamics. Our meta-RL experiments complement these works and theoretical studies partially explain them. We argue that there is a larger picture to be considered; the trade-off between modeling accuracy and optimization ease depend on characteristics of the dataset, model, and optimization, and should be studied on a case-by-case basis.

Previous theoretical studies of MAML have primarily focused on the meta-training stage. Finn et al. [10] prove that the MAML objective is smooth and convex if the task risk functions are smooth and convex and derive the DRS and MAML estimates if those functions are known for linear regression. Fallah et al. [6] characterize the meta-training sample complexity of SGD for MAML in supervised learning; Fallah et al. [7] provide analogous results for reinforcement learning. For regression with overparameterized neural networks, Wang et al. [29] show that gradient descent leads to the global optimum of the empirical MAML objective (5). Franceschi et al. [11] study the meta-learning problem (5) when $\mathtt{OPT}(\gamma, \boldsymbol{\theta})$ is a minimization operator instead of an iterative optimization procedure, proposing an algorithm with convergence guarantees. Rajeswaran et al. [20] propose implicit MAML with analysis of its training sample complexity, providing an alternative method to estimate the gradient of the MAML objective; Grazzi et al. [13] compare the quality of the estimate to that of the original MAML algorithm.

As a counterpart to Fallah et al. [6] and Fallah et al. [7], Wang et al. [30] provides bounds on the meta-testing performance of an $\epsilon$-stationary point of the MAML objective, assuming that the same set of tasks is used for meta-training and meta-testing. In contrast to these previous works, we analyze the meta-testing performance and overall optimization behavior of MAML when the meta-training and meta-testing tasks are not identical, merely drawn from the same distribution, and compare them to those of DRS.

# 6 Conclusion

This paper introduces an important trade-off in meta-learning, that of accurately modeling the meta-learning problem and complexity of the optimization problem. Classic meta-learning algorithms account for the structure of the problem space but define complex optimization objectives. Domain randomized search (DRS) does not account for the structure of the meta-learning problem and solves a single level optimization objective.

Taking MAML to be the representative meta-learning algorithm, we study this trade-off empirically and theoretically. On meta-reinforcement learning benchmarks, the optimization complexity appears to be more important; DRS is competitive with and often outperforms MAML, especially for fewer environment steps. Through an analysis of the sample complexity for smooth nonconvex risk functions, we show that DRS and MAML both solve the meta-learning problem and delineate the roles of optimization complexity and modeling accuracy. For meta-linear regression, we prove theoretically and verify in simulations that while MAML can utilize the geometry of the distribution of task losses to improve performance through meta-testing optimization, this modeling gain can be counterbalanced by its greater optimization error for small sample sizes. All three studies show that the balance of the trade-off is not only determined by the sample sizes but characteristics of the meta-learning problem, such as the smoothness of the task risk functions.

There are several interesting directions for future work. What is the trade-off exhibited by other algorithms, such as Reptile [16] or ANIL [19], that were designed to be more computationally efficient than MAML? Our theory may also be extended to more complex scenarios such as supervised learning with deep networks, the Linear Quadratic Regulator, more than one inner optimization step in MAML, or the use of part of the data to select hyperparameters.

## Broader Impact

The massive progress made by machine learning in artificial intelligence has been partly driven by massive amounts of data and compute [1]. By sharing inductive biases across tasks, meta-learning aims to speed up learning on novel tasks, thereby reducing their data and computational burden. In this paper, we have argued that the data and computational cost of the meta-training procedure matters in addition to that of the meta-test procedure. We have shown that domain randomized search, a computationally cheaper approach compared to classic meta-learning methods such as MAML, solves the meta-learning problem and is competitive with MAML when the budget of meta-training data/compute is small. Thus, it can be an effective meta-learning approach in practice when obtaining data is expensive and/or one would like to reduce the carbon footprint of meta-training, with some cost in performance at meta-test.

## Acknowledgments and Disclosure of Funding

We are very grateful to Vladlen Koltun for providing guidance during the project and to Charles Packer for help in setting up the RL experiments. We would like to thank the other members of the Intelligent Systems Lab at Intel and Amir Zamir for feedback on the first draft of this paper, and Sona Jeswani for assistance in running an early version of the RL experiments. There are no external funding or conflicts of interest to disclose.

## Footnotes

[1]For example, if $\text{OPT}(\gamma, \boldsymbol{\theta}_{meta})$ is gradient descent on the task risk function $\mathcal{R}(\boldsymbol{\theta}; \gamma)$, $\hat{\text{OPT}}(\gamma, \boldsymbol{\theta}_{meta}, N)$ would be SGD on the usual empirical risk minimization function (4).

[2]It has also been referred to as joint training in the literature [10].

[3]Risk can be seen as the chosen loss function for supervised learning and negative of reward for RL.

[4]We do not include ML1-Pick-and-Place because training was unsuccessful for all algorithms of interest.

[5]Here we abuse notation somewhat by defining the variance of a matrix $\mathbf{B}$ to be $\mathrm{Var}(\mathbf{B}) = \mathbb{E}(\mathbf{B} - \mathbb{E}(\mathbf{B})^2)$.

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
