[Supplementary Material]

# A  Details on meta-RL experiments

## A.1  Setup

**Environments**   We consider four robotic locomotion and four manipulation environments, all with continuous action spaces. The robotic locomotion environments, based on MuJoCo [27] and OpenAI Gym [3], fall into two categories.

- Varying reward functions: HalfCheetahRandVel, Walker2DRandVel

  The HalfCheetahRandVel environment was introduced in Finn et al. [9]. The distribution of tasks is a distribution of HalfCheetah robots with different goal velocities, and remains the same for meta-training and meta-testing. The Walker2DRandVel environment, defined similarly to HalfCheetahRandVel, is found in the codebase for Rothfuss et al. [21].

- Varying system dynamics: HopperRandParams, Walker2DRandParams

  The HopperRandParams and Walker2DRandParams were introduced in Rothfuss et al. [21]. For HopperRandParams, the distribution of tasks is a distribution of Hopper robots with different body mass, body inertia, damping, and friction, and remains the same for meta-training and meta-testing. The Walker2DRandParams environment is defined similarly.

We briefly describe the four manipulation environments from Metaworld; for more details please refer to Yu et al. [33].

- ML1-Push and ML1-Reach: ML1-Push considers the manipulation task of pushing a puck to a goal position. The distribution of tasks is a collection of initial puck and goal positions, and differs for meta-training and meta-testing. ML1-Reach is defined similarly to ML1-Push but with the manipulation task of reaching a goal position.

- ML10 and ML45: For both environments, the meta-training and meta-testing distributions of tasks are collections of manipulation tasks and the corresponding initial object/goal positions. The manipulation tasks in the meta-training versus meta-testing distributions do not overlap; there are 10 manipulation tasks in the training distribution for ML10, and 45 for ML45.

**Algorithms**   We consider four policy gradient algorithms, ProMP [21], which approximately combines MAML and PPO [23], DRS+PPO, a combination of DRS and PPO, TRPO-MAML [9], and DRS+TRPO, a combination of DRS and TRPO [22]. For full descriptions of the ProMP and TRPO-MAML algorithms, please refer to the cited papers. We use the implementations in the codebase provided by Rothfuss et al. [21]. To combine DRS and PPO/TRPO, it suffices to take the original PPO/TRPO algorithm and maximize the objective using generated trajectories from a sampled set of tasks instead of a single task. This follows from the fact that we can approximate an expectation over a distribution of tasks by a Monte Carlo sample of tasks.

**Meta-training**   ProMP and TRPO-MAML use the same meta-training procedure [21, 9]. At each iteration, a set of $M$ tasks are sampled from the meta-training distribution of tasks. For each task, ProMP (TRPO-MAML) generate $L$ episodes under the current policy, computes an adapted policy using policy gradient, and generates $L$ episodes under the adapted policy; all $M \times L$ episodes generated under adapted policies are used to compute its objective. For each task, DRS+PPO (DRS+TRPO) generate $L$ episodes under the current policy; all $M \times L$ episodes are used to compute its objective.

Each iteration of ProMP (TRPO-MAML) requires twice as many steps from the simulator as DRS+PPO (DRS+TRPO). Therefore, to ensure that each algorithm utilizes the same amount of data, we run ProMP (TRPO-MAML) for half as many iterations as DRS+PPO (DRS+TRPO). More specifically, for the robotic locomotion environments, we run ProMP (TRPO-MAML) for 1000 iterations and DRS+PPO (DRS+TRPO) for 2000. For the manipulation environments, we run ProMP (TRPO-MAML) for 10000 iterations and DRS+PPO (DRS+TRPO) for 20000. These go beyond the number of training steps used in Rothfuss et al. [21] and Yu et al. [33].

**Meta-testing**   The meta-testing procedure, described next, are carried out at 21 checkpoints during meta-training. We sample 1000 tasks from the meta-testing distribution of tasks. For each task and

ProMP/DRS+PPO (TRPO-MAML/DRS+TRPO), starting at a meta-trained policy, we repeat the following five times: 1) generate $L$ episodes from the current policy and 2) update the policy with the same policy gradient algorithms used to compute the adapted policies while training ProMP (TRPO-MAML). We compute the average episodic reward after $t$ policy updates, for $t = 0, 1, \ldots, 5$. These statistics are then compiled over all 1000 sampled tasks, using the procedure outlined below.

For ProMP and TRPO-MAML, the learning rate used to update the policy is the inner learning rate used to compute the adapted policies during meta-training. For DRS+PPO, it is the learning rate during meta-training (our heuristic), and for DRS+TRPO, it is zero.

**Hyperparameters**   To choose the meta-training learning rates, step sizes, and the inner learning rate used to compute the adapted policies, we conduct grid search. For the robotic locomotion environments, the learning rate for ProMP and DRS+PPO was chosen from $[0.0001, 0.001, 0.01]$, the step size for TRPO-MAML and DRS+TRPO from $[0.001, 0.01, 0.1]$, and the inner learning rate for ProMP and TRPO-MAML from $[0.01, 0.05, 0.1]$. For the manipulation environments, the learning rate for ProMP and DRS+PPO was chosen from $[0.000001, 0.00001, 0.0001, 0.001]$, the step size for TRPO-MAML and DRS+TRPO from $[0.001, 0.01, 0.1]$, and the inner learning rate for ProMP and TRPO-MAML from $[0.00001, 0.0001, 0.001, 0.01]$. These ranges include the values given in the codebases for Rothfuss et al. [21] and Yu et al. [33]. The chosen values are given in Table 1.

| Environment | ProMP | | DRS+PPO | TRPO-MAML | | DRS+TRPO |
|---|---|---|---|---|---|---|
| | LR | Inner LR | LR | Step Size | Inner LR | Step Size |
| HopperRandParams | 0.001 | 0.01 | 0.001 | 0.1 | 0.01 | 0.1 |
| Walker2DRandParams | 0.001 | 0.01 | 0.001 | 0.1 | 0.01 | 0.1 |
| HalfCheetahRandVel | 0.0001 | 0.01 | 0.01 | 0.001 | 0.01 | 0.01 |
| Walker2DRandVel | 0.001 | 0.01 | 0.001 | 0.1 | 0.01 | 0.01 |
| ML1-Push | 0.0001 | 0.0001 | 0.0001 | 0.1 | 0.0001 | 0.1 |
| ML1-Reach | 0.0001 | 0.0001 | 0.0001 | 0.001 | 0.00001 | 0.001 |
| ML10 | 0.0001 | 0.001 | 0.0001 | 0.01 | 0.001 | 0.001 |
| ML45 | 0.0001 | 0.00001 | 0.0001 | 0.1 | 0.001 | 0.01 |

Table 1: Learning rates (LR), step sizes, and inner learning rates chosen by grid search.

For the remaining hyperparameters, we used the values given in Rothfuss et al. [21] and Yu et al. [33]. We list below several of the main ones:

- $M$: 40 for the robotic locomotion environments, 20 for the manipulation environments
- $L$: 20 for the robotic locomotion environments, 10 for the manipulation environments
- Episode length: 200 for the robotic locomotion environments, 150 for the manipulation environments
- Policy architecture: a multi-layer perceptron with two hidden layers of 64 nodes for the robotic locomotion environments, and 100 nodes for the manipulation environments.
- A linear feature baseline is used to compute the advantage values.

**Result Compilation**   We run meta-training and meta-testing for all environments and algorithms for five random seeds. For a fixed environment, algorithm, and checkpoint, let the per seed estimates of the average rewards and their variances to be $\hat{R}_s, \hat{V}_s, s = 1, \ldots, 5$, and $R$ to be the corresponding random variable with mean $\mu$ and variance $\sigma^2$. The estimated mean of $R$, $\hat{\mu}$, is computed as the average of the $\hat{R}_s$. Using the formula $\mathrm{Var}[R] = \mathbb{E}[\mathrm{Var}[R \mid s]] + \mathrm{Var}[\mathbb{E}[R \mid s]]$, $\hat{\sigma}^2$, the estimated variance of $R$, is computed as the sum of 1) the average of the $\hat{V}_s$ and 2) the variance of the $\hat{R}_s$.

To compute the probability that DRS is better than MAML, we use the one sided Welch's t-test. Although the t-test makes the underlying assumption of Gaussianity, it is an acceptable assumption as reporting mean and variance is the common practice. Let the estimated average rewards be $\hat{\mu}_{drs}$

| (a) HopperRandParams | (b) Walker2DRandParams | (c) HalfCheetahRandVel | (d) Walker2DRandVel |
| (e) ML1-Push | (f) ML1-Reach | (g) ML10 | (h) ML45 |

Figure 4: DRS+PPO vs. ProMP: Average episodic rewards after one update of the policy during evaluation, as a function of the number of steps at meta-training. DRS+PPO is red and ProMP is blue.

and $\hat{\mu}_{maml}$ and their estimated standard deviations be $\hat{\sigma}_{drs}$ and $\hat{\sigma}_{maml}$. Since we use five random seeds, we compute the t-value and degree of freedom as

$$t = (\hat{\mu}_{drs} - \hat{\mu}_{maml}) \Big/ \sqrt{\frac{\hat{\sigma}_{drs}^2}{5} + \frac{\hat{\sigma}_{maml}^2}{5}} \quad \text{and} \quad \nu = \frac{\left(\hat{\sigma}_{drs}^2/5 + \hat{\sigma}_{maml}^2/5\right)^2}{(\hat{\sigma}_{drs}^2/5)^2/4 + (\hat{\sigma}_{maml}^2/5)^2/4} \tag{14}$$

We compute the probability of $\mu_{drs} > \mu_{maml}$ as one minus the CDF of a t-distribution with $\nu$ degrees of freedom at $t$.

## A.2 Additional Results

In this section we present results in a way that is more commonly seen in the meta-RL literature. We plot the average reward after one policy update at meta-testing as a function of the number of steps at meta-training. Figures 4 and 5 shows the plots for each environment for ProMP & DRS+PPO and TRPO-MAML & DRS+TRPO, respectively.

For the first two environments with variations in system dynamics only, seen in Figure 4-(a,b) and 5-(a,b), DRS is superior to MAML throughout training. For the next four environments with variations in reward functions only, either 1) DRS and MAML are comparable (Figure 4-(d) and 5-(e,f)), or 2) while DRS is initially superior, as the amount of training data increases the difference between the two algorithms usually diminishes, and eventually MAML may surpass DRS (Figure 4-(c,e,f) and 5-(c)). In the final two environments with variations in system dynamics and reward functions, the standard errors are generally too large to make a definite statement (see Figure 4-(g,h) and 5-(h)). This suggests that useful inductive biases are more difficult to learn when the system dynamics vary between tasks, a potentially interesting direction for further study.

## B Postponed Proofs from Section 3

### B.1 Proof of Theorem 1

*Proof.* We start with a generic bound on the gradient norm of a smooth function. Consider one step of SGD on a function $f(\cdot)$ that is $\mu$-smooth with learning rate $\beta^t$.

$$f(\boldsymbol{\theta}^{t+1}) = f(\boldsymbol{\theta}^t - \beta^t \mathbf{g}^t)$$
$$\leq f(\boldsymbol{\theta}^t) - \beta^t \nabla_{\boldsymbol{\theta}} f(\boldsymbol{\theta}^t)^\mathsf{T} \mathbf{g}^t + \frac{\mu \beta^{t^2}}{2} \|\mathbf{g}^t\|_2^2 \tag{15}$$

| (a) HopperRandParams | (b) Walker2DRandParams | (c) HalfCheetahRandVel | (d) Walker2DRandVel |

| (e) ML1-Push | (f) ML1-Reach | (g) ML10 | (h) ML45 |

Figure 5: DRS+TRPO vs. TRPO-MAML: Average episodic rewards after one update of the policy during evaluation, as a function of the number of steps at meta-training. DRS+TRPO is red and TRPO-MAML is blue.

where $\mathbf{g}^t$ is an estimate of the gradient of $f(\boldsymbol{\theta}^t)$. Dividing by $\beta^t$ and moving the gradient term,

$$\nabla_{\boldsymbol{\theta}} f(\boldsymbol{\theta}^t)^{\mathsf{T}} \mathbf{g}^t \leq \frac{f(\boldsymbol{\theta}^t) - f(\boldsymbol{\theta}^{t+1})}{\beta^t} + \frac{\mu \beta^t}{2} \|\mathbf{g}^t\|_2^2 \tag{16}$$

During meta-training we take $T^{tr}$ steps of SGD on the loss $\mathcal{R}^{drs}(\cdot) = \mathbb{E}_{\gamma}[\mathcal{R}(\cdot; \gamma)]$ with learning rate $\beta^{tr}$ and during meta-testing we take $T^{te}$ steps of SGD on the task loss $\mathcal{R}(\cdot; \gamma)$ with learning rate $\beta^{te}$. Both losses are $\mu$-smooth by assumption. Therefore, we can sum up the previous inequality over all $T$ steps, where $T = T^{tr} + T^{te}$.

$$\sum_{t=0}^{T^{tr}-1} \nabla_{\boldsymbol{\theta}} \mathcal{R}^{drs}(\boldsymbol{\theta}^t)^{\mathsf{T}} \mathbf{g}^t + \sum_{t=T^{tr}}^{T-1} \nabla_{\boldsymbol{\theta}} \mathcal{R}(\boldsymbol{\theta}^t; \gamma)^{\mathsf{T}} \mathbf{g}^t$$

$$\leq \frac{\mathcal{R}^{drs}(\boldsymbol{\theta}^0) - \mathcal{R}^{drs}(\boldsymbol{\theta}^{T^{tr}})}{\beta^{tr}} + \frac{\mathcal{R}(\boldsymbol{\theta}^{T^{tr}}; \gamma) - \mathcal{R}(\boldsymbol{\theta}^T; \gamma)}{\beta^{te}} + \frac{\mu \beta^{tr}}{2} \sum_{t=0}^{T^{tr}-1} \|\mathbf{g}^t\|_2^2 + \frac{\mu \beta^{te}}{2} \sum_{t=T^{tr}}^{T-1} \|\mathbf{g}^t\|_2^2$$

$$\leq \frac{\Delta}{\beta^{tr}} + \frac{\mathcal{R}(\boldsymbol{\theta}^{T^{tr}}; \gamma) - \mathcal{R}^{drs}(\boldsymbol{\theta}^{T^{tr}})}{\min\{\beta^{tr}, \beta^{te}\}} + \frac{\mathcal{R}(\boldsymbol{\theta}^*_{drs}; \gamma) - \mathcal{R}(\boldsymbol{\theta}^T; \gamma)}{\beta^{te}} + \frac{\mu \beta^{tr}}{2} \sum_{t=0}^{T^{tr}-1} \|\mathbf{g}^t\|_2^2 + \frac{\mu \beta^{te}}{2} \sum_{t=T^{tr}}^{T-1} \|\mathbf{g}^t\|_2^2$$

$$\leq \frac{\Delta}{\beta^{tr}} + \frac{\mathcal{R}(\boldsymbol{\theta}^{T^{tr}}; \gamma) - \mathcal{R}^{drs}(\boldsymbol{\theta}^{T^{tr}})}{\min\{\beta^{tr}, \beta^{te}\}} + \frac{\mathcal{R}(\boldsymbol{\theta}^*_{drs}; \gamma)}{\beta^{te}} + \frac{\mu \beta^{tr}}{2} \sum_{t=0}^{T^{tr}-1} \|\mathbf{g}^t\|_2^2 + \frac{\mu \beta^{te}}{2} \sum_{t=T^{tr}}^{T-1} \|\mathbf{g}^t\|_2^2$$

$$\leq \frac{\Delta + \mathcal{R}(\boldsymbol{\theta}^*_{drs}; \gamma)}{\min\{\beta^{tr}, \beta^{te}\}} + \frac{\mathcal{R}(\boldsymbol{\theta}^{T^{tr}}; \gamma) - \mathcal{R}^{drs}(\boldsymbol{\theta}^{T^{tr}})}{\min\{\beta^{tr}, \beta^{te}\}} + \frac{\mu \beta^{tr}}{2} \sum_{t=0}^{T^{tr}-1} \|\mathbf{g}^t\|_2^2 + \frac{\mu \beta^{te}}{2} \sum_{t=T^{tr}}^{T-1} \|\mathbf{g}^t\|_2^2$$

$$\tag{17}$$

Recall that the task losses are $L$-Lipschitz. For $t = 0$ to $t = T^{tr} - 1$,

$$\mathbf{g}^t = \frac{1}{2NM} \sum_{j=1}^{M} \sum_{i=1}^{2N} \mathbf{g}(\boldsymbol{\theta}^t, \xi_i; \gamma_j)$$

$$\mathbb{E}(\mathbf{g}^t) = \mathbb{E}_{\gamma}[\nabla_{\boldsymbol{\theta}} \mathcal{R}(\boldsymbol{\theta}^t; \gamma)] = \nabla_{\boldsymbol{\theta}} \mathcal{R}^{drs}(\boldsymbol{\theta}^t)$$

$$\mathbb{E} \left\| \mathbf{g}^t \right\|^2 = \left\| \mathbb{E}(\mathbf{g}^t) \right\|^2 + \text{tr}(\text{Var}(\mathbf{g}^t)) \leq L^2 + V^t/M + V^d/2NM$$

For $t = T^{tr}$ to $t = T - 1$,

$$\mathbf{g}^t = \frac{1}{N} \sum_{i=1}^{N} \mathbf{g}(\boldsymbol{\theta}^t, \xi_i; \gamma)$$

$$\mathbb{E}(\mathbf{g}^t) = \nabla_{\boldsymbol{\theta}} \mathcal{R}(\boldsymbol{\theta}^t; \gamma)$$

$$\mathbb{E}\left\|\mathbf{g}^t\right\|^2 = \left\|\mathbb{E}(\mathbf{g}^t)\right\|^2 + \mathrm{tr}(\mathrm{Var}(\mathbf{g}^t)) \le L^2 + V^d/N$$

Therefore, taking expectation of both sides of the previous equation over $\mathbf{g}^t$ and $\gamma$, and using the fact that $\mathbb{E}_\gamma[\mathcal{R}(\boldsymbol{\theta}^{T^{tr}}; \gamma)] = \mathcal{R}^{drs}(\boldsymbol{\theta}^{T^{tr}})$ as well as $\mathbb{E}_\gamma[\mathcal{R}(\boldsymbol{\theta}^*_{drs}; \gamma)] = \Lambda^{drs}$,

$$\sum_{t=0}^{T^{tr}-1} \left\|\nabla_{\boldsymbol{\theta}} \mathcal{R}^{drs}(\boldsymbol{\theta}^t)\right\|^2 + \sum_{t=0}^{T^{te}-1} \|\nabla_{\boldsymbol{\theta}} \mathcal{R}(\boldsymbol{\theta}^{t+T^{tr}}; \gamma)\|_2^2$$
$$\le \frac{\Delta + \Lambda^{drs}}{\min\{\beta^{tr}, \beta^{te}\}} + \frac{\mu(\beta^{tr} T^{tr}(\frac{V^d}{2NM} + \frac{V^t}{M} + L^2) + \beta^{te} T^{te}(\frac{V^d}{N} + L^2))}{2} \tag{18}$$

We can further simplify the bound by assuming $\beta^{tr} = \beta^{te}$ and optimizing it over the learning rate. Doing so, we obtain

$$\sum_{t=0}^{T^{tr}-1} \left\|\nabla_{\boldsymbol{\theta}} \mathcal{R}^{drs}(\boldsymbol{\theta}^t)\right\|^2 + \sum_{t=0}^{T^{te}-1} \|\nabla_{\boldsymbol{\theta}} \mathcal{R}(\boldsymbol{\theta}^{t+T^{tr}}; \gamma)\|^2 \le \sqrt{0.5(\Delta + \Lambda^{drs})(C_{tr}^{drs} T^{tr} + C_{te}^{drs} T^{te})} \tag{19}$$

where

$$C_{tr}^{drs} = \mu\left(L^2 + \frac{V^t}{M} + \frac{V^d}{2NM}\right) \text{ and } C_{te}^{drs} = \mu\left(L^2 + \frac{V^d}{N}\right) \tag{20}$$

$\square$

## B.2 Proof of Theorem 2

*Proof.* From Corollary A.1 of Fallah et al. [6], $\mathcal{R}^{maml}(\boldsymbol{\theta})$ has smoothness constant $\mu' = 4\mu + 2\mu^H \alpha L$. Thus, using a similar argument as in the proof of Theorem 1,

$$\sum_{t=0}^{T^{tr}-1} \nabla_{\boldsymbol{\theta}} \mathcal{R}^{maml}(\boldsymbol{\theta}^t; \alpha)^\intercal \mathbf{g}^t + \sum_{t=T^{tr}}^{T-1} \nabla_{\boldsymbol{\theta}} \mathcal{R}(\boldsymbol{\theta}^t; \gamma)^\intercal \mathbf{g}^t$$
$$\le \frac{\mathcal{R}^{maml}(\boldsymbol{\theta}^0; \alpha) - \mathcal{R}^{maml}(\boldsymbol{\theta}^{T^{tr}}; \alpha)}{\beta^{tr}} + \frac{\mathcal{R}(\boldsymbol{\theta}^{T^{tr}}; \gamma) - \mathcal{R}(\boldsymbol{\theta}^T; \gamma)}{\beta^{te}} + \frac{\mu' \beta^{tr}}{2} \sum_{t=0}^{T^{tr}-1} \|\mathbf{g}^t\|_2^2 + \frac{\mu \beta^{te}}{2} \sum_{t=T^{tr}}^{T-1} \|\mathbf{g}^t\|_2^2$$
$$\le \frac{\Delta + \mathcal{R}(\boldsymbol{\theta}^*_{maml}(\alpha) - \alpha \nabla_{\boldsymbol{\theta}} \mathcal{R}(\boldsymbol{\theta}^*_{maml}(\alpha); \gamma); \gamma)}{\min\{\beta^{tr}, \beta^{te}\}} + \frac{\mathcal{R}(\boldsymbol{\theta}^{T^{tr}}; \gamma) - \mathcal{R}^{maml}(\boldsymbol{\theta}^{T^{tr}}; \alpha)}{\min\{\beta^{tr}, \beta^{te}\}}$$
$$+ \frac{\mu' \beta^{tr}}{2} \sum_{t=0}^{T^{tr}-1} \|\mathbf{g}^t\|_2^2 + \frac{\mu \beta^{te}}{2} \sum_{t=T^{tr}}^{T-1} \|\mathbf{g}^t\|_2^2 \tag{21}$$

For $t = 0$ to $t = T^{tr} - 1$, using results from the proof of Theorem 5.12 in Fallah et al. [6],

$$\mathbf{g}^t = \frac{1}{M} \sum_{j=1}^{M} \mathbf{g}_j^t, \quad \mathbf{g}_j^t = (I - \frac{\alpha}{D} \sum_{d=1}^{D} \mathbf{h}(\boldsymbol{\theta}^t, \xi_d; \gamma_j)) \sum_{i=1}^{N} \mathbf{g}(\boldsymbol{\theta}^t - \frac{\alpha}{N} \sum_{j=1}^{N} \mathbf{g}(\boldsymbol{\theta}^t, \xi_j; \gamma_j), \xi_i; \gamma_j)$$

$$\mathbb{E}(\mathbf{g}^t) = \mathbb{E}(\mathbf{g}_j^t) = \nabla_{\boldsymbol{\theta}} \mathcal{R}^{maml}(\boldsymbol{\theta}^t; \alpha) + r^t$$

where

$$\left\|r^t\right\| \le (1 + \alpha\mu)\alpha\mu\sqrt{V^d/N}$$

$$\left\|\nabla_{\boldsymbol{\theta}} \mathcal{R}^{maml}(\boldsymbol{\theta}^t; \alpha)\right\| \le \left\|\mathbb{E}_\gamma[(I - \alpha \nabla_{\boldsymbol{\theta}}^2 \mathcal{R}(\boldsymbol{\theta}^t; \gamma)) \nabla_{\boldsymbol{\theta}} \mathcal{R}(\boldsymbol{\theta}^t - \alpha \nabla_{\boldsymbol{\theta}} \mathcal{R}(\boldsymbol{\theta}^t; \gamma); \gamma)]\right\| \le (1 + \alpha\mu)L$$

Define $\rho \triangleq (1 + \alpha\mu)$. Then,

$$\mathbb{E}\left\|\mathbf{g}^t\right\|^2 \leq \left\|\mathbb{E}(\mathbf{g}_j^t)\right\|^2 + \frac{1}{M}\mathbb{E}[\|\mathbf{g}_j^t\|^2]$$

$$\leq \left\|\nabla_{\boldsymbol{\theta}}\mathcal{R}^{maml}(\boldsymbol{\theta}^t;\alpha) + r^t\right\|^2 + \frac{1}{M}(40\left\|\nabla_{\boldsymbol{\theta}}\mathcal{R}^{maml}(\boldsymbol{\theta}^t;\alpha)\right\|^2 + 14V^t + \frac{3}{N}V^d)$$

$$\leq 2\rho^2 L^2 + 2\rho^2\alpha^2\mu^2\frac{V^d}{N} + \frac{1}{M}\left(40\rho^2 L^2 + 14V^t + \frac{3}{N}V^d\right)$$

$$\leq \left(2 + \frac{40}{M}\right)\rho^2 L^2 + \frac{14V^t}{M} + \frac{3V^d(1 + \alpha^2\mu^2 M)}{MN}$$

where we used $2\rho^2 \leq \sqrt{8} < 3$ following the assumption $\alpha \leq \frac{1}{6\mu}$. For $t = T^{tr}$ to $t = T - 1$,

$$\mathbf{g}^t = \frac{1}{N}\sum_{i=1}^N \mathbf{g}(\boldsymbol{\theta}^t, \xi_i; \gamma)$$

$$\mathbb{E}(\mathbf{g}^t) = \nabla_{\boldsymbol{\theta}}\mathcal{R}(\boldsymbol{\theta}^t; \gamma)$$

$$\mathbb{E}\left\|\mathbf{g}^t\right\|^2 = \left\|\mathbb{E}(\mathbf{g}^t)\right\|^2 + \mathrm{tr}(\mathrm{Var}(\mathbf{g}^t)) \leq L^2 + V^d/N$$

Using the Lipschitz property, $|\mathcal{R}(\boldsymbol{\theta} - \alpha\nabla\mathcal{R}(\boldsymbol{\theta};\gamma);\gamma) - \mathcal{R}(\boldsymbol{\theta})| \leq \alpha L^2$, and we can obtain the bound $|\mathbb{E}_\gamma[\mathcal{R}(\boldsymbol{\theta}^{T^{tr}};\gamma)] - \mathcal{R}^{maml}(\boldsymbol{\theta}^{T^{tr}})| \leq \alpha L^2$. Therefore, taking the expectation of both sides of (21) over $\mathbf{g}^t$ and $\gamma$ and using $\mathbb{E}_\gamma[\mathcal{R}(\boldsymbol{\theta}^*_{maml}(\alpha) - \alpha\nabla_{\boldsymbol{\theta}}\mathcal{R}(\boldsymbol{\theta}^*_{maml}(\alpha);\gamma);\gamma)] = \Lambda^{maml}(\alpha)$,

$$\sum_{t=0}^{T^{tr}-1}\left\|\nabla_{\boldsymbol{\theta}}\mathcal{R}^{maml}(\boldsymbol{\theta}^t;\alpha)\right\|^2 + \sum_{t=0}^{T^{te}-1}\|\nabla_{\boldsymbol{\theta}}\mathcal{R}(\boldsymbol{\theta}^{t+T^{tr}};\gamma)\|_2^2$$

$$\leq \frac{\Delta + \Lambda^{maml}(\alpha) + \alpha L^2}{\min\{\beta^{tr}, \beta^{te}\}} + \sum_{t=0}^{T^{tr}-1}\left\|\nabla_{\boldsymbol{\theta}}\mathcal{R}^{maml}(\boldsymbol{\theta}^t;\alpha)\right\|\left\|r^t\right\| + \beta^{te}\frac{\mu T^{te}(L^2 + V^d/N)}{2}$$

$$+ \beta^{tr}(2\mu + \mu^h\alpha L)T^{tr}\left(\left(2 + \frac{40}{M}\right)\rho^2 L^2 + \frac{14V^t}{M} + \frac{3V^d(1 + \alpha^2\mu^2 M)}{MN}\right) \qquad (22)$$

$$\leq \frac{\Delta + \Lambda^{maml}(\alpha) + \alpha L^2}{\min\{\beta^{tr}, \beta^{te}\}} + T^{tr}\alpha\mu\rho^2 L\sqrt{\frac{V^d}{N}} + \beta^{te}\frac{\mu T^{te}(L^2 + V^d/N)}{2}$$

$$+ \beta^{tr}(2\mu + \mu^H\alpha L)T^{tr}\left(\left(2 + \frac{40}{M}\right)\rho^2 L^2 + \frac{14V^t}{M} + \frac{3V^d(1 + \alpha^2\mu^2 M)}{MN}\right)$$

Assuming $\beta^{tr} = \beta^{te}$ and optimizing the bound over the learning rate, we obtain

$$\sum_{t=0}^{T^{tr}-1}\left\|\nabla_{\boldsymbol{\theta}}\mathcal{R}^{maml}(\boldsymbol{\theta}^t;\alpha)\right\|^2 + \sum_{t=0}^{T^{te}-1}\|\nabla_{\boldsymbol{\theta}}\mathcal{R}(\boldsymbol{\theta}^{t+T^{tr}};\gamma)\|_2^2$$

$$\leq T^{tr}\alpha\mu\rho^2 L\sqrt{V^d/N} + \sqrt{0.5(\Delta + \Lambda^{maml}(\alpha) + \alpha L^2)(C_{tr}^{maml}T^{tr} + C_{te}^{maml}T^{te})} \qquad (23)$$

where

$$C_{tr}^{maml} = \mu'\left(\left(2 + \frac{40}{M}\right)\rho^2 L^2 + \frac{14V^t}{M} + \frac{3V^d(1 + \alpha^2\mu^2 M)}{MN}\right) \text{ and } C_{te}^{maml} = \mu\left(L^2 + \frac{V^d}{N}\right)$$

$$\tag{24}$$

$\square$

## C  Postponed Proofs and Derivations from Section 4

### C.1  Derivation for risk function in Equation 9

$$\mathcal{R}(\boldsymbol{\theta}; \gamma) = \frac{1}{2}\mathbb{E}[(y_\gamma - \boldsymbol{\theta}^\mathsf{T}\mathbf{x}_\gamma)^2 \mid \gamma]$$

$$= \frac{1}{2}\mathbb{E}[(\epsilon_\gamma + (\boldsymbol{\theta}_\gamma - \boldsymbol{\theta})^\mathsf{T}\mathbf{x}_\gamma)^2 \mid \gamma]$$

$$= \frac{1}{2}\mathbb{E}[\epsilon_\gamma^2 + 2(\boldsymbol{\theta}_\gamma - \boldsymbol{\theta})^\mathsf{T}\mathbf{x}_\gamma\epsilon_\gamma + (\boldsymbol{\theta}_\gamma - \boldsymbol{\theta})^\mathsf{T}\mathbf{x}_\gamma\mathbf{x}_\gamma^\mathsf{T}(\boldsymbol{\theta}_\gamma - \boldsymbol{\theta}) \mid \gamma]$$

$$= \frac{1}{2}[\sigma_\gamma^2 + (\boldsymbol{\theta}_\gamma - \boldsymbol{\theta})^\mathsf{T}\mathbf{Q}_\gamma(\boldsymbol{\theta}_\gamma - \boldsymbol{\theta})]$$

$$= \frac{1}{2}\boldsymbol{\theta}^\mathsf{T}\mathbf{Q}_\gamma\boldsymbol{\theta} - \boldsymbol{\theta}_\gamma^\mathsf{T}\mathbf{Q}_\gamma\boldsymbol{\theta} + \frac{1}{2}\boldsymbol{\theta}_\gamma^\mathsf{T}\mathbf{Q}_\gamma\boldsymbol{\theta}_\gamma + \frac{1}{2}\sigma_\gamma^2$$

where the third equality uses the model (8).

### C.2  Derivation for optimal solutions in Equation 10

Recall that $\boldsymbol{\theta}_{drs}^* \triangleq \arg\min_\theta \mathbb{E}_\gamma[\mathcal{R}(\boldsymbol{\theta}; \gamma)]$ and $\boldsymbol{\theta}_{maml}^*(\alpha) \triangleq \arg\min_\theta \mathbb{E}_\gamma[\mathcal{R}(\boldsymbol{\theta} - \alpha\nabla_{\boldsymbol{\theta}}\mathcal{R}(\boldsymbol{\theta}; \gamma); \gamma)]$. Using Equation 9,

$$\mathbb{E}_\gamma[\mathcal{R}(\boldsymbol{\theta}; \gamma)] = \mathbb{E}_\gamma[\frac{1}{2}\boldsymbol{\theta}^\mathsf{T}\mathbf{Q}_\gamma\boldsymbol{\theta} - \boldsymbol{\theta}_\gamma^\mathsf{T}\mathbf{Q}_\gamma\boldsymbol{\theta} + \frac{1}{2}\boldsymbol{\theta}_\gamma^\mathsf{T}\mathbf{Q}_\gamma\boldsymbol{\theta}_\gamma + \frac{1}{2}\sigma_\gamma^2]$$

$$= \frac{1}{2}\boldsymbol{\theta}^\mathsf{T}\mathbb{E}_\gamma[\mathbf{Q}_\gamma]\boldsymbol{\theta} - \mathbb{E}_\gamma[\mathbf{Q}_\gamma\boldsymbol{\theta}_\gamma]^\mathsf{T}\boldsymbol{\theta} + \frac{1}{2}\mathbb{E}_\gamma[\boldsymbol{\theta}_\gamma^\mathsf{T}\mathbf{Q}_\gamma\boldsymbol{\theta}_\gamma] + \frac{1}{2}\mathbb{E}_\gamma[\sigma_\gamma^2]$$

$$\nabla_{\boldsymbol{\theta}}\mathbb{E}_\gamma[\mathcal{R}(\boldsymbol{\theta}; \gamma)] = \mathbb{E}_\gamma[\mathbf{Q}_\gamma]\boldsymbol{\theta} - \mathbb{E}_\gamma[\mathbf{Q}_\gamma\boldsymbol{\theta}_\gamma]$$

Since $\mathbb{E}_\gamma[\mathcal{R}(\boldsymbol{\theta}; \gamma)]$ is quadratic in $\boldsymbol{\theta}$, it is minimized at a first-order stationary point. Thus, setting $\nabla_{\boldsymbol{\theta}}\mathbb{E}_\gamma[\mathcal{R}(\boldsymbol{\theta}; \gamma)]$ equal to zero, we obtain $\boldsymbol{\theta}_{drs}^* = \mathbb{E}_\gamma[\mathbf{Q}_\gamma]^{-1}\mathbb{E}_\gamma[\mathbf{Q}_\gamma\boldsymbol{\theta}_\gamma]$.

Similarly,

$$\mathbb{E}_\gamma[\mathcal{R}(\boldsymbol{\theta} - \alpha\nabla_{\boldsymbol{\theta}}\mathcal{R}(\boldsymbol{\theta}; \gamma); \gamma)]$$

$$= \mathbb{E}_\gamma[\mathcal{R}(\boldsymbol{\theta} - \alpha(\mathbf{Q}_\gamma\boldsymbol{\theta} - \mathbf{Q}_\gamma\boldsymbol{\theta}_\gamma); \gamma)]$$

$$= \mathbb{E}_\gamma[\mathcal{R}((\mathbf{I} - \alpha\mathbf{Q}_\gamma)\boldsymbol{\theta} + \alpha\mathbf{Q}_\gamma\boldsymbol{\theta}_\gamma; \gamma)]$$

$$= \frac{1}{2}\mathbb{E}_\gamma[((\mathbf{I} - \alpha\mathbf{Q}_\gamma)\boldsymbol{\theta} + \alpha\mathbf{Q}_\gamma\boldsymbol{\theta}_\gamma)^\mathsf{T}\mathbf{Q}_\gamma((\mathbf{I} - \alpha\mathbf{Q}_\gamma)\boldsymbol{\theta} + \alpha\mathbf{Q}_\gamma\boldsymbol{\theta}_\gamma)]$$

$$- \mathbb{E}_\gamma[\boldsymbol{\theta}_\gamma^\mathsf{T}\mathbf{Q}_\gamma((\mathbf{I} - \alpha\mathbf{Q}_\gamma)\boldsymbol{\theta} + \alpha\mathbf{Q}_\gamma\boldsymbol{\theta}_\gamma)] + \frac{1}{2}\mathbb{E}_\gamma[\boldsymbol{\theta}_\gamma^\mathsf{T}\mathbf{Q}_\gamma\boldsymbol{\theta}_\gamma] + \frac{1}{2}\mathbb{E}_\gamma[\sigma_\gamma^2]$$

$$= \frac{1}{2}\boldsymbol{\theta}^\mathsf{T}\mathbb{E}_\gamma[(\mathbf{I} - \alpha\mathbf{Q}_\gamma)\mathbf{Q}_\gamma(\mathbf{I} - \alpha\mathbf{Q}_\gamma)]\boldsymbol{\theta} - \mathbb{E}_\gamma[\boldsymbol{\theta}_\gamma^\mathsf{T}(\mathbf{I} - \alpha\mathbf{Q}_\gamma)\mathbf{Q}_\gamma(\mathbf{I} - \alpha\mathbf{Q}_\gamma)]\boldsymbol{\theta}$$

$$+ \frac{1}{2}\mathbb{E}_\gamma[\boldsymbol{\theta}_\gamma^\mathsf{T}(\mathbf{I} - \alpha\mathbf{Q}_\gamma)\mathbf{Q}_\gamma(\mathbf{I} - \alpha\mathbf{Q}_\gamma)\boldsymbol{\theta}_\gamma] + \frac{1}{2}\mathbb{E}_\gamma[\sigma_\gamma^2]$$

$$\nabla_{\boldsymbol{\theta}}\mathbb{E}_\gamma[\mathcal{R}(\boldsymbol{\theta} - \alpha\nabla_{\boldsymbol{\theta}}\mathcal{R}(\boldsymbol{\theta}; \gamma); \gamma)]$$

$$= \mathbb{E}_\gamma[(\mathbf{I} - \alpha\mathbf{Q}_\gamma)\mathbf{Q}_\gamma(\mathbf{I} - \alpha\mathbf{Q}_\gamma)]\boldsymbol{\theta} - \mathbb{E}_\gamma[(\mathbf{I} - \alpha\mathbf{Q}_\gamma)\mathbf{Q}_\gamma(\mathbf{I} - \alpha\mathbf{Q}_\gamma)\boldsymbol{\theta}_\gamma]$$

$\mathbb{E}_\gamma[\mathcal{R}(\boldsymbol{\theta} - \alpha\nabla_{\boldsymbol{\theta}}\mathcal{R}(\boldsymbol{\theta}; \gamma); \gamma)]$ is also quadratic in $\boldsymbol{\theta}$, so we obtain its minimizer by setting its gradient equal to zero. Thus, $\boldsymbol{\theta}_{maml}^*(\alpha) = \mathbb{E}_\gamma[(\mathbf{I} - \alpha\mathbf{Q}_\gamma)\mathbf{Q}_\gamma(\mathbf{I} - \alpha\mathbf{Q}_\gamma)]^{-1}\mathbb{E}_\gamma[(\mathbf{I} - \alpha\mathbf{Q}_\gamma)\mathbf{Q}_\gamma(\mathbf{I} - \alpha\mathbf{Q}_\gamma)\boldsymbol{\theta}_\gamma]$.

### C.3  Proof of Theorem 3 and 4

#### C.3.1  Useful Lemmas and Preliminary Results

We start with stating a few useful lemmas to be used in the proof of the main statements. Following them, we analyze $\hat{\boldsymbol{\theta}}_{drs}$ and $\hat{\boldsymbol{\theta}}_{maml}(\alpha)$.

**Lemma 1.** *Let $\mathbf{X}_{j1}$ be the $p \times N$ matrix with columns $\mathbf{x}_{j,i}, i = 1, \ldots, N$ and $\mathbf{X}_{j2}$ be the $p \times N$ matrix with columns $\mathbf{x}_{j,i}, i = N + 1, \ldots, 2N$. Let $\mathbf{Y}_{j1}$ be the $N$-dimensional vector*

*with entries $y_{j,i}, i = 1, \ldots, N$, and $\mathbf{Y}_{j2}$ be the $N$-dimensional vector with entries $y_{j,i}, i = N+1, \ldots, 2N$. Let $\mathbf{X} = [\ldots \ \mathbf{X}_{j1} \ \mathbf{X}_{j2} \ \ldots]$ and $\mathbf{Y} = [\ldots \ \mathbf{Y}_{j1}^{\mathsf{T}} \ \mathbf{Y}_{j2}^{\mathsf{T}} \ \ldots]^{\mathsf{T}}$. Let $\mathbf{W}(\alpha) = \left[\ldots \ (\mathbf{I} - \frac{\alpha}{N}\mathbf{X}_{j1}\mathbf{X}_{j2}^{\mathsf{T}})\mathbf{X}_{j2} \ \ldots\right]$ and $\mathbf{Z}(\alpha) = \left[\ldots \ (\mathbf{Y}_{j2} - \frac{\alpha}{N}\mathbf{X}_{j2}^{\mathsf{T}}\mathbf{X}_{j1}\mathbf{Y}_{j1})^{\mathsf{T}} \ \ldots\right]^{\mathsf{T}}$. Then,*

$$\hat{\boldsymbol{\theta}}_{drs} = (\mathbf{X}^{\mathsf{T}})^{+}\mathbf{Y} \quad and \quad \hat{\boldsymbol{\theta}}_{maml}(\alpha) = (\mathbf{W}(\alpha)^{\mathsf{T}})^{+}\mathbf{Z}(\alpha)$$

*where $()^{+}$ denotes the Moore-Penrose pseudoinverse of a matrix.*

*Proof.* Using (11), we can write the DR estimate as minimizing the objective

$$\frac{1}{4MN}\sum_{j=1}^{M}\sum_{i=1}^{2N}(y_{j,i} - \boldsymbol{\theta}^{\mathsf{T}}\mathbf{x}_{j,i})^2 = \frac{1}{4MN}\sum_{j=1}^{M}\left(\|\mathbf{Y}_{j1} - \mathbf{X}_{j1}^{T}\boldsymbol{\theta}\|_2^2 + \|\mathbf{Y}_{j2} - \mathbf{X}_{j2}^{T}\boldsymbol{\theta}\|_2^2\right)$$

$$= \frac{1}{4MN}\|\mathbf{Y} - \mathbf{X}^{\mathsf{T}}\boldsymbol{\theta}\|_2^2$$

From Penrose [18], the value of $\boldsymbol{\theta}$ that minimizes the above is $(\mathbf{X}^{\mathsf{T}})^{+}\mathbf{Y}$.

Using (12), we can write the MAML estimate as minimizing the objective

$$\frac{1}{2MN}\sum_{j=1}^{M}\sum_{i=N+1}^{2N}(y_{j,i} - \tilde{\boldsymbol{\theta}}_j(\alpha)^{\mathsf{T}}\mathbf{x}_{j,i})^2$$

$$= \frac{1}{2MN}\sum_{j=1}^{M}\|\mathbf{Y}_{j2} - \mathbf{X}_{j2}^{\mathsf{T}}\tilde{\boldsymbol{\theta}}_j(\alpha)\|_2^2$$

$$= \frac{1}{2MN}\sum_{j=1}^{M}\|\mathbf{Y}_{j2} - \mathbf{X}_{j2}^{\mathsf{T}}(\boldsymbol{\theta} - \frac{\alpha}{N}(\mathbf{X}_{j1}\mathbf{X}_{j1}^{\mathsf{T}}\boldsymbol{\theta} - \mathbf{X}_{j1}\mathbf{Y}_{j1}))\|_2^2$$

$$= \frac{1}{2MN}\sum_{j=1}^{M}\|\mathbf{Y}_{j2} - \frac{\alpha}{N}\mathbf{X}_{j2}^{\mathsf{T}}\mathbf{X}_{j1}\mathbf{Y}_{j1} - ((\mathbf{I} - \frac{\alpha}{N}\mathbf{X}_{j1}\mathbf{X}_{j1}^{T})\mathbf{X}_{j2})^{\mathsf{T}}\boldsymbol{\theta}\|_2^2$$

$$= \frac{1}{2MN}\|\mathbf{Z}(\alpha) - \mathbf{W}(\alpha)^{\mathsf{T}}\boldsymbol{\theta}\|_2^2$$

With the same reasoning as for the DR estimate, this is minimized when $\boldsymbol{\theta}$ equals $(\mathbf{W}(\alpha)^{\mathsf{T}})^{+}\mathbf{Z}(\alpha)$. $\square$

Next, we obtain some useful high probability concentration inequalities as a direct consequence of matrix Bernstein's inequality [28].

**Lemma 2.** *Assume $\|\mathbf{Q}_\gamma\| \leq \beta$ with probability $1$. With probability at least $1 - \varrho$,*

$$\left\|\sum_{j=1}^{M}(\mathbf{Q}_j - \mathbb{E}_\gamma[\mathbf{Q}_\gamma])\right\| \leq \frac{2\beta}{3}\log\frac{2p}{\varrho} + \sqrt{2M\|\mathrm{Var}_\gamma[\mathbf{Q}_\gamma]\|\log\frac{2p}{\varrho}}$$

*Proof.* Notice that $\mathbf{Q}_j - \mathbb{E}_\gamma[\mathbf{Q}_\gamma]$ are $M$ independent, mean zero, symmetric random matrices. From the matrix Bernstein's inequality [28], for any $t \geq 0$,

$$\mathbb{P}\left\{\left\|\sum_{j=1}^{M}(\mathbf{Q}_j - \mathbb{E}_\gamma[\mathbf{Q}_\gamma])\right\| \geq t\right\} \leq 2p\exp\left\{-\frac{t^2/2}{s_1^2 + \beta t/3}\right\}$$

where

$$s_1^2 = \left\|\sum_{j=1}^{M}\mathbb{E}[(\mathbf{Q}_j - \mathbb{E}_\gamma[\mathbf{Q}_\gamma])^2]\right\| \leq M\left\|\mathbb{E}_\gamma[(\mathbf{Q}_\gamma - \mathbb{E}_\gamma[\mathbf{Q}_\gamma])^2]\right\| = M\|\mathrm{Var}_\gamma[\mathbf{Q}_\gamma]\|.$$

Therefore,

$$\mathbb{P}\left\{\left\|\sum_{j=1}^{M}(\mathbf{Q}_j - \mathbb{E}_\gamma[\mathbf{Q}_\gamma])\right\| \geq t\right\} \leq 2p \exp\left\{-\frac{t^2/2}{M\left\|\mathrm{Var}_\gamma[\mathbf{Q}_\gamma]\right\| + \beta t/3}\right\}.$$

Setting $\omega = \dfrac{t^2/2}{M\left\|\mathrm{Var}_\gamma[\mathbf{Q}_\gamma]\right\| + \beta t/3}$, we solve for $t$. $t$ satisfies the quadratic equation $3t^2 - 2\beta\omega t - 6M\left\|\mathrm{Var}_\gamma[\mathbf{Q}_\gamma]\right\|\omega = 0$, which has the positive root $(\beta\omega + \sqrt{\beta^2\omega^2 + 18M\left\|\mathrm{Var}_\gamma[\mathbf{Q}_\gamma]\right\|\omega})/3$. Therefore, the previous inequality becomes

$$\mathbb{P}\{\left\|\sum_{j=1}^{M}(\mathbf{Q}_j - \mathbb{E}_\gamma[\mathbf{Q}_\gamma])\right\| \geq \frac{\beta\omega + \sqrt{\beta^2\omega^2 + 18M\left\|\mathrm{Var}_\gamma[\mathbf{Q}_\gamma]\right\|\omega}}{3}\} \leq 2p\exp\{-\omega\}$$

and since $(\beta\omega + \sqrt{\beta^2\omega^2 + 18M\left\|\mathrm{Var}_\gamma[\mathbf{Q}_\gamma]\right\|\omega})/3 \leq 2\beta\omega/3 + \sqrt{2M\left\|\mathrm{Var}_\gamma[\mathbf{Q}_\gamma]\right\|\omega}$,

$$\mathbb{P}\{\left\|\sum_{j=1}^{M}(\mathbf{Q}_j - \mathbb{E}_\gamma[\mathbf{Q}_\gamma])\right\| \geq \frac{2\beta\omega}{3} + \sqrt{2M\left\|\mathrm{Var}_\gamma[\mathbf{Q}_\gamma]\right\|\omega}\} \leq 2p\exp\{-\omega\}.$$

Set $w = \log\frac{2p}{\varrho}$ to obtain the final result. $\qquad\square$

**Remark.** A similar proof will also show that if $\|\mathbf{Q}_\gamma\| \leq \beta$ and $\|\mathbf{I} - \alpha\mathbf{Q}_\gamma\| \leq \mu$ with probability 1, $\left\|\sum_{j=1}^{M}((\mathbf{I} - \alpha\mathbf{Q}_j)\mathbf{Q}_j(\mathbf{I} - \alpha\mathbf{Q}_j) - \mathbb{E}_\gamma[(\mathbf{I} - \alpha\mathbf{Q}_\gamma)\mathbf{Q}_\gamma(\mathbf{I} - \alpha\mathbf{Q}_\gamma)])\right\| \leq \dfrac{2\beta\mu^2}{3}\log\frac{2p}{\varrho} + \sqrt{2M\left\|\mathrm{Var}_\gamma[(\mathbf{I} - \alpha\mathbf{Q}_\gamma)\mathbf{Q}_\gamma(\mathbf{I} - \alpha\mathbf{Q}_\gamma)]\right\|}\log\frac{2p}{\varrho}$ with probability $1 - \varrho$.

**Lemma 3.** *Assume $\|\mathbf{Q}_\gamma\| \leq \beta$ and $\|\boldsymbol{\theta}_\gamma\| \leq \eta$ with probability 1. With probability at least $1 - \varrho$,*

$$\left\|\sum_{j=1}^{M}(\mathbf{Q}_j\boldsymbol{\theta}_j - \mathbb{E}_\gamma[\mathbf{Q}_\gamma\boldsymbol{\theta}_\gamma])\right\| \leq \frac{2\beta\eta}{3}\log\frac{2(p+1)}{\varrho} + \sqrt{2M\,\mathrm{tr}(\mathrm{Var}_\gamma[\mathbf{Q}_\gamma\boldsymbol{\theta}_\gamma])\log\frac{2(p+1)}{\varrho}}$$

*Proof.* Notice that $\mathbf{Q}_j\boldsymbol{\theta}_j - \mathbb{E}_\gamma[\mathbf{Q}_\gamma\boldsymbol{\theta}_\gamma]$ are $M$ independent, mean zero, random vectors. From the matrix Bernstein's inequality for rectangular matrices [28], since $\|\mathbf{Q}_\gamma\boldsymbol{\theta}_\gamma\| \leq \|\mathbf{Q}_\gamma\|\|\boldsymbol{\theta}_\gamma\| \leq \beta\eta$, for any $t \geq 0$,

$$\mathbb{P}\left\{\left\|\sum_{j=1}^{M}(\mathbf{Q}_j\boldsymbol{\theta}_j - \mathbb{E}_\gamma[\mathbf{Q}_\gamma\boldsymbol{\theta}_\gamma])\right\| \geq t\right\} \leq 2(p+1)\exp\left\{-\frac{t^2/2}{s_2^2 + \beta\eta t/3}\right\}$$

where

$$\begin{aligned}
s_2^2 &= \max\left(\left\|\sum_{j=1}^{M}\mathbb{E}[(\mathbf{Q}_j\boldsymbol{\theta}_j - \mathbb{E}_\gamma[\mathbf{Q}_\gamma\boldsymbol{\theta}_\gamma])(\mathbf{Q}_j\boldsymbol{\theta}_j - \mathbb{E}_\gamma[\mathbf{Q}_\gamma\boldsymbol{\theta}_\gamma])^\mathsf{T}]\right\|,\right.\\
&\qquad\qquad\left.\left\|\sum_{j=1}^{M}\mathbb{E}[(\mathbf{Q}_j\boldsymbol{\theta}_j - \mathbb{E}_\gamma[\mathbf{Q}_\gamma\boldsymbol{\theta}_\gamma])^\mathsf{T}(\mathbf{Q}_j\boldsymbol{\theta}_j - \mathbb{E}_\gamma[\mathbf{Q}_\gamma\boldsymbol{\theta}_\gamma])]\right\|\right)\\
&\leq M\max(\left\|\mathbb{E}_\gamma[(\mathbf{Q}_\gamma\boldsymbol{\theta}_\gamma - \mathbb{E}_\gamma[\mathbf{Q}_\gamma\boldsymbol{\theta}_\gamma])(\mathbf{Q}_\gamma\boldsymbol{\theta}_\gamma - \mathbb{E}_\gamma[\mathbf{Q}_\gamma\boldsymbol{\theta}_\gamma])^\mathsf{T}]\right\|,\\
&\qquad\qquad\left\|\mathbb{E}_\gamma[(\mathbf{Q}_\gamma\boldsymbol{\theta}_\gamma - \mathbb{E}_\gamma[\mathbf{Q}_\gamma\boldsymbol{\theta}_\gamma])^\mathsf{T}(\mathbf{Q}_\gamma\boldsymbol{\theta}_\gamma - \mathbb{E}_\gamma[\mathbf{Q}_\gamma\boldsymbol{\theta}_\gamma])]\right\|)\\
&= M\max(\left\|\mathrm{Var}_\gamma[\mathbf{Q}_\gamma\boldsymbol{\theta}_\gamma]\right\|, \left\|\mathrm{tr}(\mathrm{Var}_\gamma[\mathbf{Q}_\gamma\boldsymbol{\theta}_\gamma])\right\|)\\
&\leq M\,\mathrm{tr}(\mathrm{Var}_\gamma[\mathbf{Q}_\gamma\boldsymbol{\theta}_\gamma])
\end{aligned}$$

Therefore,

$$\mathbb{P}\left\{\left\|\sum_{j=1}^{M}(\mathbf{Q}_j\boldsymbol{\theta}_j - \mathbb{E}_\gamma[\mathbf{Q}_\gamma\boldsymbol{\theta}_\gamma])\right\| \geq t\right\} \leq 2(p+1)\exp\left\{-\frac{t^2/2}{M\,\mathrm{tr}(\mathrm{Var}_\gamma[\mathbf{Q}_\gamma\boldsymbol{\theta}_\gamma]) + \beta\eta t/3}\right\}.$$

Using the same argument as in the proof of Lemma 2, we have that

$$\mathbb{P}\left\{\left\|\sum_{j=1}^{M}(\mathbf{Q}_j\boldsymbol{\theta}_j - \mathbb{E}_\gamma[\mathbf{Q}_\gamma\boldsymbol{\theta}_\gamma])\right\| \geq \frac{2\beta\eta\omega}{3} + \sqrt{2M\operatorname{tr}(\operatorname{Var}_\gamma[\mathbf{Q}_\gamma\boldsymbol{\theta}_\gamma])\omega}\right\} \leq 2(p+1)\exp\{-\omega\}.$$

Set $w = \log\frac{2(p+1)}{\varrho}$ to obtain the final result.

$\square$

**Remark.** A similar argument will also show that if $\|\mathbf{Q}_\gamma\| \leq \beta$, $\|\mathbf{I} - \alpha\mathbf{Q}_\gamma\| \leq \mu$, and $\|\boldsymbol{\theta}_\gamma\| \leq \eta$,
$\left\|\sum_{j=1}^{M}((\mathbf{I} - \alpha\mathbf{Q}_j)\mathbf{Q}_j(\mathbf{I} - \alpha\mathbf{Q}_j)\boldsymbol{\theta}_j - \mathbb{E}_\gamma[(\mathbf{I} - \alpha\mathbf{Q}_\gamma)\mathbf{Q}_\gamma(\mathbf{I} - \alpha\mathbf{Q}_\gamma)\boldsymbol{\theta}_\gamma])\right\| \leq \frac{2\beta\mu^2\eta}{3}\log\frac{2(p+1)}{\varrho} +$
$\sqrt{2M\operatorname{tr}(\operatorname{Var}_\gamma[(\mathbf{I} - \alpha\mathbf{Q}_\gamma)\mathbf{Q}_\gamma(\mathbf{I} - \alpha\mathbf{Q}_\gamma)\boldsymbol{\theta}_\gamma])\log\frac{2(p+1)}{\varrho}}$ with probability at least $1 - \varrho$.

**Lemma 4.** *Fix the task $j$. Assume $\|\mathbf{Q}_\gamma\| \leq \beta$ with probability 1 and the distribution of $\mathbf{x}_{\gamma,i}$ conditional on $\gamma$ is sub-Gaussian with parameter $K$. With probability at least $1 - \varrho$,*

$$\left\|\frac{\mathbf{X}_{j1}\mathbf{X}_{j1}^\mathsf{T}}{N} - \mathbf{Q}_j\right\| \leq \beta CK^2\left(\sqrt{\frac{p + \log\frac{2}{\varrho}}{N}} + \frac{p + \log\frac{2}{\varrho}}{N}\right)$$

*Proof.* From results on covariance estimation from Vershynin [28],

$$\mathbb{P}\left\{\left\|\frac{\mathbf{X}_{j1}\mathbf{X}_{j1}^\mathsf{T}}{N} - \mathbf{Q}_j\right\| \geq CK^2\left(\sqrt{\frac{p + \omega}{N}} + \frac{p + \omega}{2N}\right)\|\mathbf{Q}_j\|\right\} \leq 2\exp\{-\omega\}.$$

Since $\mathbf{Q}_j \leq \beta$ with probability 1,

$$\mathbb{P}\left\{\left\|\frac{\mathbf{X}_{j1}\mathbf{X}_{j1}^\mathsf{T}}{N} - \mathbf{Q}_j\right\| \geq \beta CK^2\left(\sqrt{\frac{p + \omega}{N}} + \frac{p + \omega}{N}\right)\right\} \leq 2\exp\{-\omega\}.$$

Set $w = \log\frac{2}{\varrho}$ to obtain the final result.

$\square$

**Lemma 5.** *Fix the task $j$ and let $\mathbf{e}_{j1}$ be the $N$-dimensional vector with entries $\epsilon_{j,i}, i = 1, \dots, N$. If $\|\mathbf{x}_{\gamma,i}\| \leq \xi$ and $|\epsilon_{\gamma,i}| \leq \phi$ with probability 1, $\|\mathbf{X}_{j1}\mathbf{e}_{j1}\| \geq \frac{2\xi\phi\omega}{3} + \sqrt{2N\operatorname{tr}(\mathbb{E}_\gamma[\sigma_\gamma^2\mathbf{Q}_\gamma])\omega}$ with probability at most $2(p+1)e^{-\omega}$.*

*Proof.* Notice that $\mathbf{X}_{j1}\mathbf{e}_{j1} = \sum_{i=1}^{N}\mathbf{x}_{j,i}\epsilon_{j,i}$ is the sum of $N$ independent, mean zero, random vectors. From the matrix Bernstein's inequality for rectangular matrices, for any $t \geq 0$,

$$\mathbb{P}\{\|\mathbf{X}_{j1}\mathbf{e}_{j1}\| \geq t\} \leq 2(p+1)\exp\{-\frac{t^2}{s_3^2 + \xi\phi t/3}\}$$

for

$$s_3^2 = \max\left(\left\|\sum_{i=1}^{N}\mathbb{E}[\mathbf{x}_{j,i}\epsilon_{j,i}^2\mathbf{x}_{j,i}^\mathsf{T}]\right\|, \left\|\sum_{i=1}^{N}\mathbb{E}[\epsilon_{j,i}\mathbf{x}_{j,i}^\mathsf{T}\mathbf{x}_{j,i}\epsilon_{j,i}]\right\|\right)$$
$$\leq N\max(\left\|\mathbb{E}_\gamma[\sigma_\gamma^2\mathbf{Q}_\gamma]\right\|, \left\|\mathbb{E}_\gamma[\sigma_\gamma^2\mathbb{E}[\mathbf{x}_{\gamma,i}^\mathsf{T}\mathbf{x}_{\gamma,i} \mid \gamma]]\right\|)$$
$$= N\max(\left\|\mathbb{E}_\gamma[\sigma_\gamma^2\mathbf{Q}_\gamma]\right\|, \left\|\mathbb{E}_\gamma[\sigma_\gamma^2\operatorname{tr}(\mathbf{Q}_\gamma)]\right\|)$$
$$= N\max(\left\|\mathbb{E}_\gamma[\sigma_\gamma^2\mathbf{Q}_\gamma]\right\|, \operatorname{tr}(\mathbb{E}_\gamma[\sigma_\gamma^2\mathbf{Q}_\gamma]))$$
$$= N\operatorname{tr}(\mathbb{E}_\gamma[\sigma_\gamma^2\mathbf{Q}_\gamma])$$

where the last equality follows from the fact that for a symmetric positive semidefinite matrix such as $\mathbb{E}_\gamma[\sigma_\gamma^2\mathbf{Q}_\gamma]$, the norm is the largest eigenvalue and the trace is the sum of the eigenvalues. Therefore,

$$\mathbb{P}\{\|\mathbf{X}_{j1}\mathbf{e}_{j1}\| \geq t\} \leq 2(p+1)\exp\{-\frac{t^2/2}{N\operatorname{tr}(\mathbb{E}_\gamma[\sigma_\gamma^2\mathbf{Q}_\gamma]) + \xi\phi t/3}\}.$$

Using the same argument as Lemma 2, we have that

$$\mathbb{P}\{\|\mathbf{X}_{j1}\mathbf{e}_{j1}\| \geq \frac{2\xi\phi\omega}{3} + \sqrt{2N\operatorname{tr}(\mathbb{E}_\gamma[\sigma_\gamma^2\mathbf{Q}_\gamma])\omega}\} \leq 2(p+1)\exp\{-\omega\}.$$

$\square$

### C.3.2 Proof of Theorem 3

*Proof.* Let $\mathbf{X}_j = \begin{bmatrix} \mathbf{X}_{j1} & \mathbf{X}_{j2} \end{bmatrix}$ and $\mathbf{Y}_j = \begin{bmatrix} \mathbf{Y}_{j1}^\mathsf{T} & \mathbf{Y}_{j2}^\mathsf{T} \end{bmatrix}^\mathsf{T}$. Let $\mathbf{e}_j$ be the $2N$-dimensional vector with entries $\epsilon_{j,i}, i = 1, \ldots, 2N$. Let the singular value decomposition of $\mathbf{X}$ be $UDV^\mathsf{T}$. Then, from Proposition 1 and using the identity $\mathbf{Y}_j = \mathbf{X}_j^\mathsf{T}\boldsymbol{\theta}_j + \mathbf{e}_j$,

$$\hat{\boldsymbol{\theta}}_{drs} - \boldsymbol{\theta}_{drs}^* = (\mathbf{X}^\mathsf{T})^+\mathbf{Y} - \boldsymbol{\theta}_{drs}^*$$

$$= (\mathbf{X}^\mathsf{T})^+ \begin{bmatrix} \vdots \\ \mathbf{X}_j^\mathsf{T}\boldsymbol{\theta}_j + \mathbf{e}_j \\ \vdots \end{bmatrix} - \boldsymbol{\theta}_{drs}^*$$

$$= (\mathbf{X}^\mathsf{T})^+ \begin{bmatrix} \vdots \\ \mathbf{X}_j^\mathsf{T}(\boldsymbol{\theta}_j - \boldsymbol{\theta}_{drs}^*) + \mathbf{e}_j \\ \vdots \end{bmatrix} + (\mathbf{X}^\mathsf{T})^+ \begin{bmatrix} \vdots \\ \mathbf{X}_j^\mathsf{T}\boldsymbol{\theta}_{drs}^* \\ \vdots \end{bmatrix} - \boldsymbol{\theta}_{drs}^*$$

$$= (\mathbf{X}\mathbf{X}^\mathsf{T})^+\mathbf{X} \begin{bmatrix} \vdots \\ \mathbf{X}_j^\mathsf{T}(\boldsymbol{\theta}_j - \boldsymbol{\theta}_{drs}^*) + \mathbf{e}_j \\ \vdots \end{bmatrix} + (\mathbf{X}\mathbf{X}^\mathsf{T})^+\mathbf{X}\mathbf{X}^\mathsf{T}\boldsymbol{\theta}_{drs}^* - \boldsymbol{\theta}_{drs}^*$$

$$= U(DD^\mathsf{T})^+U^\mathsf{T}\sum_{j=1}^M \left(\mathbf{X}_j\mathbf{X}_j^\mathsf{T}(\boldsymbol{\theta}_j - \boldsymbol{\theta}_{drs}^*) + \mathbf{X}_j\mathbf{e}_j\right)$$

$$+ U(DD^\mathsf{T})^+(DD^\mathsf{T})U^\mathsf{T}\boldsymbol{\theta}_{drs}^* - \boldsymbol{\theta}_{drs}^*$$

Note that $DD^\mathsf{T}$ is a diagonal matrix of eigenvalues of $\mathbf{X}\mathbf{X}^\mathsf{T}$; $(DD^\mathsf{T})^+$ is a diagonal matrix with entries the reciprocals of the nonzero eigenvalues of $\mathbf{X}\mathbf{X}^\mathsf{T}$ and the rest zeros. Then, $(DD^\mathsf{T})^+(DD^\mathsf{T})$ is a diagonal matrix with $r$ ones, where $r$ is the number of nonzero eigenvalues of $\mathbf{X}\mathbf{X}^\mathsf{T}$, and $U(DD^\mathsf{T})^+(DD^\mathsf{T})U^\mathsf{T} = diag(I_r, 0)$. Let $\lambda_{min}()$ denote the smallest eigenvalue of a matrix. Thus,

$$\left\|\hat{\boldsymbol{\theta}}_{drs} - \boldsymbol{\theta}_{drs}^*\right\|_2 \leq \lambda_{min}(\mathbf{X}\mathbf{X}^\mathsf{T})^{-1}\left\|\sum_{j=1}^M (\mathbf{X}_j\mathbf{X}_j^\mathsf{T}(\boldsymbol{\theta}_j - \boldsymbol{\theta}_{drs}^*) + \mathbf{X}_j\mathbf{e}_j)\right\| \tag{25}$$

$$+ \|\boldsymbol{\theta}_{drs}^*\|\,\mathbb{I}\{\lambda_{min}(\mathbf{X}\mathbf{X}^\mathsf{T}) = 0\}$$

where

$$\lambda_{min}(\mathbf{X}\mathbf{X}^\mathsf{T}) = \lambda_{min}\left(\sum_{j=1}^M \mathbf{X}_j\mathbf{X}_j^\mathsf{T}\right)$$

$$= \lambda_{min}\left(2N\sum_{j=1}^M (\frac{\mathbf{X}_j\mathbf{X}_j^\mathsf{T}}{2N} - \mathbf{Q}_j) + 2N\sum_{j=1}^M (\mathbf{Q}_j - \mathbb{E}_\gamma[\mathbf{Q}_\gamma]) + 2NM\mathbb{E}_\gamma[\mathbf{Q}_\gamma]\right)$$

$$\geq 2NM\lambda_{min}(\mathbb{E}_\gamma[\mathbf{Q}_\gamma]) + 2N\lambda_{min}(\sum_{j=1}^M (\mathbf{Q}_j - \mathbb{E}_\gamma[\mathbf{Q}_\gamma])) + 2N\sum_{j=1}^M \lambda_{min}(\frac{\mathbf{X}_j\mathbf{X}_j^\mathsf{T}}{2N} - \mathbf{Q}_j)$$

$$\geq 2NM\lambda_{min}(\mathbb{E}_\gamma[\mathbf{Q}_\gamma]) - 2N\left\|\sum_{j=1}^M (\mathbf{Q}_j - \mathbb{E}_\gamma[\mathbf{Q}_\gamma])\right\| - 2N\sum_{j=1}^M \left\|\frac{\mathbf{X}_j\mathbf{X}_j^\mathsf{T}}{2N} - \mathbf{Q}_j\right\|, \tag{26}$$

$$\left\|\sum_{j=1}^{M} \mathbf{X}_j(\mathbf{X}_j^{\mathsf{T}}(\boldsymbol{\theta}_j - \boldsymbol{\theta}_{drs}^*) + \mathbf{e}_j)\right\| = 2N \left\|\sum_{j=1}^{M} (\frac{\mathbf{X}_j\mathbf{X}_j^{\mathsf{T}}}{2N} - \mathbf{Q}_j)(\boldsymbol{\theta}_j - \boldsymbol{\theta}_{drs}^*) + \sum_{j=1}^{M} \mathbf{Q}_j(\boldsymbol{\theta}_j - \boldsymbol{\theta}_{drs}^*) + \sum_{j=1}^{M} \frac{\mathbf{X}_j\mathbf{e}_j}{2N}\right\|$$

$$\leq 2N \sum_{j=1}^{M} \left\|\frac{\mathbf{X}_j\mathbf{X}_j^{\mathsf{T}}}{2N} - \mathbf{Q}_j\right\| \|\boldsymbol{\theta}_j - \boldsymbol{\theta}_{drs}^*\| + 2N \left\|\sum_{j=1}^{M}(\mathbf{Q}_j\boldsymbol{\theta}_j - \mathbb{E}_\gamma[\mathbf{Q}_\gamma\boldsymbol{\theta}_\gamma])\right\|$$

$$+ 2N \left\|\sum_{j=1}^{M}(\mathbf{Q}_j - \mathbb{E}_\gamma[\mathbf{Q}_\gamma])\right\| \|\boldsymbol{\theta}_{drs}^*\| + \sum_{j=1}^{M} \|\mathbf{X}_j\mathbf{e}_j\|.$$

$$(27)$$

Let $\omega > 0$. Consider the following probabilistic events. Using union bound, at least one of them occurs with probability at most $2(2p + 1 + M + (p+1)M)e^{-\omega}$

- (E1): $\left\|\sum_{j=1}^{M}(\mathbf{Q}_j - \mathbb{E}_\gamma[\mathbf{Q}_\gamma])\right\| \geq \frac{2\beta\omega}{3} + \sqrt{2M \|\mathrm{Var}_\gamma[\mathbf{Q}_\gamma]\| \omega}$, where $\|\mathbf{Q}_\gamma\| \leq \beta$ with probability 1. This event occurs with probability at most $2pe^{-\omega}$, by Lemma 2.

- (E2): $\left\|\sum_{j=1}^{M}(\mathbf{Q}_j\boldsymbol{\theta}_j - \mathbb{E}_\gamma[\mathbf{Q}_\gamma\boldsymbol{\theta}_\gamma])\right\| \geq \frac{2\beta\eta\omega}{3} + \sqrt{2M \mathrm{tr}(\mathrm{Var}_\gamma[\mathbf{Q}_\gamma\boldsymbol{\theta}_\gamma])\omega}$. Occurs with probability at most $2(p+1)e^{-\omega}$, by Lemma 3.

- (E3-1, …, E3-M): For $j = 1, \ldots, M$, $\left\|\frac{\mathbf{X}_j\mathbf{X}_j^{\mathsf{T}}}{2N} - \mathbf{Q}_j\right\| \geq \beta CK^2(\sqrt{\frac{p+\omega}{2N}} + \frac{p+\omega}{2N})$, where $\|\mathbf{Q}_\gamma\| \leq \beta$ with probability 1, the distribution of $\mathbf{x}_{\gamma,i}$ conditional on $\gamma$ is sub-Gaussian with parameter $K$, and $C$ is a constant. For each $j$, this occurs with probability at most $2e^{-\omega}$, from extending Lemma 4 to $\mathbf{X}_j$.

- (E4-1, …, E4-M): For $j = 1, \ldots, M$, $\|\mathbf{X}_j\mathbf{e}_j\| \geq \frac{2\xi\phi\omega}{3} + \sqrt{4N \mathrm{tr}(\mathbb{E}_\gamma[\sigma_\gamma^2\mathbf{Q}_\gamma])\omega}$, where $\|\mathbf{x}_{\gamma,i}\| \leq \xi$ and $|\epsilon_{\gamma,i}| \leq \phi$ with probability 1. For each $j$, this occurs with probability at most $2(p+1)e^{-\omega}$, from Lemma 5 to $\mathbf{X}_j\mathbf{e}_j$.

From (26) and (27), with probability at least $1 - 2(pM + 2M + 2p + 1)e^{-\omega}$,

$$\lambda_{min}(\mathbf{X}\mathbf{X}^{\mathsf{T}}) \geq 2NM\lambda_{min}(\mathbb{E}_\gamma[\mathbf{Q}_\gamma]) - 2N(\frac{2\beta\omega}{3} + \sqrt{2M \|\mathrm{Var}_\gamma[\mathbf{Q}_\gamma]\| \omega} + M(\beta CK^2(\sqrt{\frac{p+\omega}{2N}} + \frac{p+\omega}{2N}))))$$

$$= 2NM\left(\lambda_{min}(\mathbb{E}_\gamma[\mathbf{Q}_\gamma]) - (\sqrt{\frac{2 \|\mathrm{Var}_\gamma[\mathbf{Q}_\gamma]\| \omega}{M}} + \frac{2\beta\omega}{3M}) - \beta CK^2(\sqrt{\frac{p+\omega}{2N}} + \frac{p+\omega}{2N})\right),$$

$$\left\|\sum_{j=1}^{M}(\mathbf{X}_j\mathbf{X}_j^{\mathsf{T}}(\boldsymbol{\theta}_j - \boldsymbol{\theta}_{drs}^*) + \mathbf{X}_j\mathbf{e}_j)\right\|$$

$$\leq 2NM\tau\beta CK^2(\sqrt{\frac{p+\omega}{2N}} + \frac{p+\omega}{2N}) + 2N(\frac{2\beta\eta\omega}{3} + \sqrt{2M \mathrm{tr}(\mathrm{Var}_\gamma[\mathbf{Q}_\gamma\boldsymbol{\theta}_\gamma])\omega})$$

$$+ 2N \|\boldsymbol{\theta}_{drs}^*\| (\frac{2\beta\omega}{3} + \sqrt{2M \|\mathrm{Var}_\gamma[\mathbf{Q}_\gamma]\| \omega}) + M(\frac{2\xi\phi\omega}{3} + \sqrt{4N \mathrm{tr}(\mathbb{E}_\gamma[\sigma_\gamma^2\mathbf{Q}_\gamma])\omega})$$

$$\leq 2NM\left(\tau\beta CK^2(\sqrt{\frac{p+\omega}{2N}} + \frac{p+\omega}{2N}) + (\sqrt{\frac{2 \mathrm{tr}(\mathrm{Var}_\gamma[\mathbf{Q}_\gamma\boldsymbol{\theta}_\gamma])\omega}{M}} + \frac{2\beta\eta\omega}{3M})\right.$$

$$\left. + \|\boldsymbol{\theta}_{drs}^*\| (\sqrt{\frac{2 \|\mathrm{Var}_\gamma[\mathbf{Q}_\gamma]\| \omega}{M}} + \frac{2\beta\omega}{3M}) + (\sqrt{\frac{\mathrm{tr}(\mathbb{E}_\gamma[\sigma_\gamma^2\mathbf{Q}_\gamma])\omega}{N}} + \frac{\xi\phi\omega}{3N})\right).$$

Thus, by (25), if $\lambda_{min}(\mathbb{E}_\gamma[\mathbf{Q}_\gamma]) - (\sqrt{\frac{2 \|\mathrm{Var}_\gamma[\mathbf{Q}_\gamma]\| \omega}{M}} + \frac{2\beta\omega}{3M}) - \beta CK^2(\sqrt{\frac{p+\omega}{2N}} + \frac{p+\omega}{2N}) > 0$,

with probability at least $1 - 2(pM + 2M + 2p + 1)e^{-\omega}$, $\left\|\hat{\boldsymbol{\theta}}_{drs} - \boldsymbol{\theta}_{drs}^*\right\|$ is bounded above by

$$\frac{\tau\beta CK^2(\sqrt{\frac{p+\omega}{2N}} + \frac{p+\omega}{2N}) + \sqrt{\frac{2\operatorname{tr}(\operatorname{Var}_\gamma[\mathbf{Q}_\gamma\boldsymbol{\theta}_\gamma])\omega}{M}} + \frac{2\beta\eta\omega}{3M} + \|\boldsymbol{\theta}_{drs}^*\|(\sqrt{\frac{2\|\operatorname{Var}_\gamma[\mathbf{Q}_\gamma]\|\omega}{M}} + \frac{2\beta\omega}{3M}) + \sqrt{\frac{\operatorname{tr}(\mathbb{E}_\gamma[\sigma_\gamma^2\mathbf{Q}_\gamma])\omega}{N}} + \frac{\xi\phi\omega}{3N}}{\lambda_{min}(\mathbb{E}_\gamma[\mathbf{Q}_\gamma]) - (\sqrt{\frac{2\|\operatorname{Var}_\gamma[\mathbf{Q}_\gamma]\|\omega}{M}} + \frac{2\beta\omega}{3M}) - \beta CK^2(\sqrt{\frac{p+\omega}{2N}} + \frac{p+\omega}{2N})}$$

Defining $\delta = 2(pM + 2M + 2p + 1)e^{-\omega}$, we let $\omega = \ln(pM + 2M + 2p + 1) - \ln(\delta/2)$. We have that $c_1(\omega, \|\operatorname{Var}_\gamma[\mathbf{Q}_\gamma]\|, \operatorname{tr}(\operatorname{Var}_\gamma[\mathbf{Q}_\gamma\boldsymbol{\theta}_\gamma]), \boldsymbol{\theta}_{drs}^*) = \|\boldsymbol{\theta}_{drs}^*\|\sqrt{2\|\operatorname{Var}_\gamma[\mathbf{Q}_\gamma]\|\omega} + \sqrt{2\operatorname{tr}(\operatorname{Var}_\gamma[\mathbf{Q}_\gamma\boldsymbol{\theta}_\gamma])\omega}$, $c_2(\omega) = \beta CK^2\sqrt{p+\omega}$, and $c_3(\omega) = \sqrt{\operatorname{tr}(\mathbb{E}_\gamma[\sigma_\gamma^2\mathbf{Q}_\gamma])\omega}$.

Hence, if $\lambda_{min}(\mathbb{E}_\gamma[\mathbf{Q}_\gamma]) - \tilde{o}(1) > 0$, with probability at least $1 - \delta$, $\left\|\hat{\boldsymbol{\theta}}_{drs} - \boldsymbol{\theta}_{drs}^*\right\|$ is bounded by

$$(\lambda_{min}(\mathbb{E}_\gamma[\mathbf{Q}_\gamma]) - \tilde{o}(1))^{-1}(\frac{c_1(\omega, \|\operatorname{Var}_\gamma[\mathbf{Q}_\gamma]\|, \operatorname{tr}(\operatorname{Var}_\gamma[\mathbf{Q}_\gamma\boldsymbol{\theta}_\gamma]), \boldsymbol{\theta}_{drs}^*)}{\sqrt{M}} + \frac{\tau}{\sqrt{2}}\frac{c_2(\omega)}{\sqrt{N}} + \frac{c_3(\omega)}{\sqrt{N}}$$

$$+ \tilde{o}(\frac{1}{\sqrt{M}}) + \tilde{o}(\frac{1}{\sqrt{N}}))$$

where $\omega = \ln(pM + 2M + 2p + 1) - \ln(\delta/2)$. $\qquad\square$

### C.3.3 Theorem 4

*Proof.* Let $\mathbf{e}_{j1}$ be the $N$-dimensional vector with entries $\epsilon_{j,i}, i = 1, \ldots, N$ and $\mathbf{e}_{j2}$ be the $N$-dimensional vector with entries $\epsilon_{j,i}, i = N+1, \ldots, 2N$. Let the SVD of $\mathbf{W}$ be $UDV^\intercal$. Then, from Proposition 1 and using the identities $\mathbf{Y}_{j1} = \mathbf{X}_{j1}^\intercal\boldsymbol{\theta}_j + \mathbf{e}_{j1}$ and $\mathbf{Y}_{j2} = \mathbf{X}_{j2}^\intercal\boldsymbol{\theta}_j + \mathbf{e}_{j2}$,

$$\hat{\boldsymbol{\theta}}_{maml}(\alpha) - \boldsymbol{\theta}_{maml}^*(\alpha)$$
$$= (\mathbf{W}^\intercal)^+\mathbf{Z} - \boldsymbol{\theta}_{maml}^*(\alpha)$$
$$= (\mathbf{W}^\intercal)^+\begin{bmatrix} \vdots \\ \mathbf{X}_{j2}^\intercal\boldsymbol{\theta}_j + \mathbf{e}_{j2} - \alpha\mathbf{X}_{j2}^\intercal\mathbf{X}_{j1}(\mathbf{X}_{j1}^\intercal\boldsymbol{\theta}_j + \mathbf{e}_{j1})/N \\ \vdots \end{bmatrix} - \boldsymbol{\theta}_{maml}^*(\alpha)$$
$$= (\mathbf{W}^\intercal)^+\begin{bmatrix} \vdots \\ \mathbf{X}_{j2}^\intercal(\mathbf{I} - \frac{\alpha}{N}\mathbf{X}_{j1}\mathbf{X}_{j1}^\intercal)(\boldsymbol{\theta}_j - \boldsymbol{\theta}_{maml}^*(\alpha)) + \mathbf{e}_{j2} - \frac{\alpha}{N}\mathbf{X}_{j2}^\intercal\mathbf{X}_{j1}\mathbf{e}_{j1} \\ \vdots \end{bmatrix}$$
$$+ (\mathbf{W}^\intercal)^+\begin{bmatrix} \vdots \\ \mathbf{X}_{j2}^\intercal(\mathbf{I} - \frac{\alpha}{N}\mathbf{X}_{j1}\mathbf{X}_{j1}^\intercal)\boldsymbol{\theta}_{maml}^*(\alpha) \\ \vdots \end{bmatrix} - \boldsymbol{\theta}_{maml}^*(\alpha)$$
$$= (\mathbf{W}\mathbf{W}^\intercal)^+\mathbf{W}\begin{bmatrix} \vdots \\ \mathbf{X}_{j2}^\intercal(\mathbf{I} - \frac{\alpha}{N}\mathbf{X}_{j1}\mathbf{X}_{j1}^\intercal)(\boldsymbol{\theta}_j - \boldsymbol{\theta}_{maml}^*(\alpha)) + \mathbf{e}_{j2} - \frac{\alpha}{N}\mathbf{X}_{j2}^\intercal\mathbf{X}_{j1}\mathbf{e}_{j1} \\ \vdots \end{bmatrix}$$
$$+ (\mathbf{W}\mathbf{W}^\intercal)^+\mathbf{W}\mathbf{W}^\intercal\boldsymbol{\theta}_{maml}^*(\alpha) - \boldsymbol{\theta}_{maml}^*(\alpha)$$
$$= (UDD^\intercal U^\intercal)^+\sum_{j=1}^M((\mathbf{I} - \frac{\alpha}{N}\mathbf{X}_{j1}\mathbf{X}_{j1}^\intercal)\mathbf{X}_{j2}\mathbf{X}_{j2}^\intercal(\mathbf{I} - \frac{\alpha}{N}\mathbf{X}_{j1}\mathbf{X}_{j1}^\intercal)(\boldsymbol{\theta}_j - \boldsymbol{\theta}_{maml}^*(\alpha))$$
$$+ (\mathbf{I} - \frac{\alpha}{N}\mathbf{X}_{j1}\mathbf{X}_{j1}^\intercal)(\mathbf{X}_{j2}\mathbf{e}_{j2} - \frac{\alpha}{N}\mathbf{X}_{j2}\mathbf{X}_{j2}^\intercal\mathbf{X}_{j1}\mathbf{e}_{j1}))$$
$$+ (UDD^\intercal U^\intercal)^+ UDD^\intercal U^\intercal\boldsymbol{\theta}_{maml}^*(\alpha) - \boldsymbol{\theta}_{maml}^*(\alpha)$$

Note that $DD^\intercal$ is a diagonal matrix of eigenvalues of $\mathbf{W}\mathbf{W}^\intercal$; $(DD^\intercal)^+$ is a diagonal matrix with entries the reciprocals of the nonzero eigenvalues of $\mathbf{W}\mathbf{W}^\intercal$ and the rest zeros. Then, $(DD^\intercal)^+(DD^\intercal)$

is a diagonal matrix with $r$ ones, where $r$ is the number of nonzero eigenvalues of $\mathbf{W}\mathbf{W}^\intercal$, and $U(DD^\intercal)^+(DD^\intercal)U^\intercal = diag(I_r, 0)$. Let $\lambda_{min}()$ denote the smallest eigenvalue of a matrix. Thus,

$$
\begin{aligned}
&\left\| \hat{\boldsymbol{\theta}}_{maml}(\alpha) - \boldsymbol{\theta}^*_{maml}(\alpha) \right\| \\
&\leq \lambda_{min}(\mathbf{W}\mathbf{W}^\intercal)^{-1} \left\| \sum_{j=1}^{M} ((\mathbf{I} - \frac{\alpha}{N}\mathbf{X}_{j1}\mathbf{X}_{j1}^\intercal)\mathbf{X}_{j2}\mathbf{X}_{j2}^\intercal(\mathbf{I} - \frac{\alpha}{N}\mathbf{X}_{j1}\mathbf{X}_{j1}^\intercal)(\boldsymbol{\theta}_j - \boldsymbol{\theta}^*_{maml}(\alpha)) \right. \\
&\qquad \left. + (\mathbf{I} - \frac{\alpha}{N}\mathbf{X}_{j1}\mathbf{X}_{j1}^\intercal)(\mathbf{X}_{j2}\mathbf{e}_{j2} - \frac{\alpha}{N}\mathbf{X}_{j2}\mathbf{X}_{j2}^\intercal\mathbf{X}_{j1}\mathbf{e}_{j1})) \right\| \\
&\quad + \| \boldsymbol{\theta}^*_{maml}(\alpha) \| \, \mathbb{I}\{\lambda_{min}(\mathbf{W}\mathbf{W}^\intercal) = 0\}
\end{aligned}
\tag{28}
$$

where

$$
\begin{aligned}
&\lambda_{min}(\mathbf{W}\mathbf{W}^\intercal) \\
&= \lambda_{min}\left( \sum_{j=1}^{M}(\mathbf{I} - \frac{\alpha}{N}\mathbf{X}_{j1}\mathbf{X}_{j1}^\intercal)\mathbf{X}_{j2}\mathbf{X}_{j2}^\intercal(\mathbf{I} - \frac{\alpha}{N}\mathbf{X}_{j1}\mathbf{X}_{j1}^\intercal) \right) \\
&\geq \lambda_{min}\left( N\sum_{j=1}^{M}(\mathbf{I} - \alpha\mathbf{Q}_j)\mathbf{Q}_j(\mathbf{I} - \alpha\mathbf{Q}_j) - N\alpha\sum_{j=1}^{M}(\mathbf{I} - \alpha\mathbf{Q}_j)\mathbf{Q}_j(\frac{\mathbf{X}_{j1}\mathbf{X}_{j1}^\intercal}{N} - \mathbf{Q}_j) \right. \\
&\quad - N\alpha\sum_{j=1}^{M}(\frac{\mathbf{X}_{j1}\mathbf{X}_{j1}^\intercal}{N} - \mathbf{Q}_j)\mathbf{Q}_j(\mathbf{I} - \alpha\mathbf{Q}_j) + N\sum_{j=1}^{M}(\mathbf{I} - \alpha\mathbf{Q}_j)(\frac{\mathbf{X}_{j2}\mathbf{X}_{j2}^\intercal}{N} - \mathbf{Q}_j)(\mathbf{I} - \alpha\mathbf{Q}_j) \\
&\quad - N\alpha\sum_{j=1}^{M}(\mathbf{I} - \alpha\mathbf{Q}_j)(\frac{\mathbf{X}_{j2}\mathbf{X}_{j2}^\intercal}{N} - \mathbf{Q}_j)(\frac{\mathbf{X}_{j1}\mathbf{X}_{j1}^\intercal}{N} - \mathbf{Q}_j) \\
&\quad - N\alpha\sum_{j=1}^{M}(\frac{\mathbf{X}_{j1}\mathbf{X}_{j1}^\intercal}{N} - \mathbf{Q}_j)(\frac{\mathbf{X}_{j2}\mathbf{X}_{j2}^\intercal}{N} - \mathbf{Q}_j)(\mathbf{I} - \alpha\mathbf{Q}_j) \\
&\quad \left. + N\alpha^2\sum_{j=1}^{M}(\frac{\mathbf{X}_{j1}\mathbf{X}_{j1}^\intercal}{N} - \mathbf{Q}_j)\frac{\mathbf{X}_{j2}\mathbf{X}_{j2}^\intercal}{N}(\frac{\mathbf{X}_{j1}\mathbf{X}_{j1}^\intercal}{N} - \mathbf{Q}_j) \right) \\
&\geq MN\lambda_{min}(\mathbb{E}_\gamma[(\mathbf{I} - \alpha\mathbf{Q}_\gamma)\mathbf{Q}_\gamma(\mathbf{I} - \alpha\mathbf{Q}_\gamma)]) \\
&\quad - N\left\| \sum_{j=1}^{M}((\mathbf{I} - \alpha\mathbf{Q}_j)\mathbf{Q}_j(\mathbf{I} - \alpha\mathbf{Q}_j) - \mathbb{E}_\gamma[(\mathbf{I} - \alpha\mathbf{Q}_\gamma)\mathbf{Q}_\gamma(\mathbf{I} - \alpha\mathbf{Q}_\gamma)]) \right\| \\
&\quad - 2N\alpha\sum_{j=1}^{M}\|\mathbf{I} - \alpha\mathbf{Q}_j\| \, \|\mathbf{Q}_j\| \left\| \frac{\mathbf{X}_{j1}\mathbf{X}_{j1}^\intercal}{N} - \mathbf{Q}_j \right\| \\
&\quad - N\sum_{j=1}^{M}\|\mathbf{I} - \alpha\mathbf{Q}_j\|^2 \left\| \frac{\mathbf{X}_{j2}\mathbf{X}_{j2}^\intercal}{N} - \mathbf{Q}_j \right\| \\
&\quad - 2N\alpha\sum_{j=1}^{M}\|\mathbf{I} - \alpha\mathbf{Q}_j\| \left\| \frac{\mathbf{X}_{j1}\mathbf{X}_{j1}^\intercal}{N} - \mathbf{Q}_j \right\| \left\| \frac{\mathbf{X}_{j2}\mathbf{X}_{j2}^\intercal}{N} - \mathbf{Q}_j \right\|
\end{aligned}
$$

and

$$
\left\| \sum_{j=1}^{M} ((\mathbf{I} - \frac{\alpha}{N}\mathbf{X}_{j1}\mathbf{X}_{j1}^{\mathsf{T}})\mathbf{X}_{j2}\mathbf{X}_{j2}^{\mathsf{T}}(\mathbf{I} - \frac{\alpha}{N}\mathbf{X}_{j1}\mathbf{X}_{j1}^{\mathsf{T}})(\boldsymbol{\theta}_j - \boldsymbol{\theta}_{maml}^*(\alpha)) \right.
$$
$$
\left. + (\mathbf{I} - \frac{\alpha}{N}\mathbf{X}_{j1}\mathbf{X}_{j1}^{\mathsf{T}})(\mathbf{X}_{j2}\mathbf{e}_{j2} - \frac{\alpha}{N}\mathbf{X}_{j2}\mathbf{X}_{j2}^{\mathsf{T}}\mathbf{X}_{j1}\mathbf{e}_{j1})) \right\|
$$
$$
\leq N \left\| \sum_{j=1}^{M} ((\mathbf{I} - \alpha\mathbf{Q}_j)\mathbf{Q}_j(\mathbf{I} - \alpha\mathbf{Q}_j)\boldsymbol{\theta}_j - \mathbb{E}_\gamma[(\mathbf{I} - \alpha\mathbf{Q}_\gamma)\mathbf{Q}_\gamma(\mathbf{I} - \alpha\mathbf{Q}_\gamma)\boldsymbol{\theta}_\gamma]) \right\|
$$
$$
+ N \left\| \sum_{j=1}^{M} ((\mathbf{I} - \alpha\mathbf{Q}_j)\mathbf{Q}_j(\mathbf{I} - \alpha\mathbf{Q}_j) - \mathbb{E}_\gamma[(\mathbf{I} - \alpha\mathbf{Q}_\gamma)\mathbf{Q}_\gamma(\mathbf{I} - \alpha\mathbf{Q}_\gamma)]) \right\| \|\boldsymbol{\theta}_{maml}^*(\alpha)\|
$$
$$
+ 2N\alpha \sum_{j=1}^{M} \|\mathbf{I} - \alpha\mathbf{Q}_j\| \|\mathbf{Q}_j\| \left\| \frac{\mathbf{X}_{j1}\mathbf{X}_{j1}^{\mathsf{T}}}{N} - \mathbf{Q}_j \right\| \|\boldsymbol{\theta}_j - \boldsymbol{\theta}_{maml}^*(\alpha)\|
$$
$$
+ N \sum_{j=1}^{M} \|\mathbf{I} - \alpha\mathbf{Q}_j\|^2 \left\| \frac{\mathbf{X}_{j2}\mathbf{X}_{j2}^{\mathsf{T}}}{N} - \mathbf{Q}_j \right\| \|\boldsymbol{\theta}_j - \boldsymbol{\theta}_{maml}^*(\alpha)\|
$$
$$
+ 2N\alpha \sum_{j=1}^{M} \|\mathbf{I} - \alpha\mathbf{Q}_j\| \left\| \frac{\mathbf{X}_{j1}\mathbf{X}_{j1}^{\mathsf{T}}}{N} - \mathbf{Q}_j \right\| \left\| \frac{\mathbf{X}_{j2}\mathbf{X}_{j2}^{\mathsf{T}}}{N} - \mathbf{Q}_j \right\| \|\boldsymbol{\theta}_j - \boldsymbol{\theta}_{maml}^*(\alpha)\|
$$
$$
+ N\alpha^2 \sum_{j=1}^{M} \|\mathbf{Q}_j\| \left\| \frac{\mathbf{X}_{j1}\mathbf{X}_{j1}^{\mathsf{T}}}{N} - \mathbf{Q}_j \right\|^2 \|\boldsymbol{\theta}_j - \boldsymbol{\theta}_{maml}^*(\alpha)\|
$$
$$
+ N\alpha^2 \sum_{j=1}^{M} \left\| \frac{\mathbf{X}_{j1}\mathbf{X}_{j1}^{\mathsf{T}}}{N} - \mathbf{Q}_j \right\|^2 \left\| \frac{\mathbf{X}_{j2}\mathbf{X}_{j2}^{\mathsf{T}}}{N} - \mathbf{Q}_j \right\| \|\boldsymbol{\theta}_j - \boldsymbol{\theta}_{maml}^*(\alpha)\|
$$
$$
+ \sum_{j=1}^{M} \|\mathbf{I} - \alpha\mathbf{Q}_j\| \|\mathbf{X}_{j2}\mathbf{e}_{j2}\| + \alpha \sum_{j=1}^{M} \|\mathbf{I} - \alpha\mathbf{Q}_j\| \left\| \frac{\mathbf{X}_{j2}\mathbf{X}_{j2}^{\mathsf{T}}}{N} - \mathbf{Q}_j \right\| \|\mathbf{X}_{j1}\mathbf{e}_{j1}\|
$$
$$
+ \alpha \sum_{j=1}^{M} \|\mathbf{I} - \alpha\mathbf{Q}_j\| \|\mathbf{Q}_j\| \|\mathbf{X}_{j1}\mathbf{e}_{j1}\| + \alpha \sum_{j=1}^{M} \left\| \frac{\mathbf{X}_{j1}\mathbf{X}_{j1}^{\mathsf{T}}}{N} - \mathbf{Q}_j \right\| \|\mathbf{X}_{j2}\mathbf{e}_{j2}\|
$$
$$
+ \alpha^2 \sum_{j=1}^{M} \left\| \frac{\mathbf{X}_{j1}\mathbf{X}_{j1}^{\mathsf{T}}}{N} - \mathbf{Q}_j \right\| \left\| \frac{\mathbf{X}_{j2}\mathbf{X}_{j2}^{\mathsf{T}}}{N} - \mathbf{Q}_j \right\| \|\mathbf{X}_{j1}\mathbf{e}_{j1}\| + \alpha^2 \sum_{j=1}^{M} \left\| \frac{\mathbf{X}_{j1}\mathbf{X}_{j1}^{\mathsf{T}}}{N} - \mathbf{Q}_j \right\| \|\mathbf{Q}_j\| \|\mathbf{X}_{j1}\mathbf{e}_{j1}\|
$$

$$
(29)
$$

Let $\omega > 0$. Consider the following probabilistic events:

- (E1): $\left\| \sum_{j=1}^{M}((\mathbf{I} - \alpha\mathbf{Q}_j)\mathbf{Q}_j(\mathbf{I} - \alpha\mathbf{Q}_j) - \mathbb{E}_\gamma[(\mathbf{I} - \alpha\mathbf{Q}_\gamma)\mathbf{Q}_\gamma(\mathbf{I} - \alpha\mathbf{Q}_\gamma)]) \right\| \geq \frac{2\beta\mu^2\omega}{3} + \sqrt{2M\|\mathrm{Var}_\gamma[\mathbf{S}_\gamma]\|\omega}$, where $\|\mathbf{Q}_\gamma\| \leq \beta$ and $\|\mathbf{I} - \alpha\mathbf{Q}_\gamma\| \leq \mu$ with probability 1.

  This occurs with probability at most $2pe^{-\omega}$, from the remark after Lemma 2.

- (E2): $\left\| \sum_{j=1}^{M}((\mathbf{I} - \alpha\mathbf{Q}_j)\mathbf{Q}_j(\mathbf{I} - \alpha\mathbf{Q}_j)\boldsymbol{\theta}_j - \mathbb{E}_\gamma[(\mathbf{I} - \alpha\mathbf{Q}_\gamma)\mathbf{Q}_\gamma(\mathbf{I} - \alpha\mathbf{Q}_\gamma)\boldsymbol{\theta}_\gamma]) \right\| \geq \frac{2\beta\mu^2\eta\omega}{3} + \sqrt{2M\,\mathrm{tr}(\mathrm{Var}_\gamma[\mathbf{S}_\gamma\boldsymbol{\theta}_\gamma])\omega}$ where $\|\mathbf{Q}_\gamma\| \leq \beta$, $\|\mathbf{I} - \alpha\mathbf{Q}_\gamma\| \leq \mu$, and $\|\boldsymbol{\theta}_\gamma\| \leq \eta$ with probability 1.

  This occurs with probability at most $2(p+1)e^{-\omega}$, from the remark after Lemma 3.

- (E3-1, …, E3-2M): For $j = 1, \ldots, M$, $\left\| \dfrac{\mathbf{X}_{j1}\mathbf{X}_{j1}^{\mathsf{T}}}{N} - \mathbf{Q}_j \right\| \geq \beta C K^2 (\sqrt{\dfrac{p+\omega}{N}} + \dfrac{p+\omega}{N})$

  and $\left\| \dfrac{\mathbf{X}_{j2}\mathbf{X}_{j2}^{\mathsf{T}}}{N} - \mathbf{Q}_j \right\| \geq \beta C K^2 (\sqrt{\dfrac{p+\omega}{N}} + \dfrac{p+\omega}{N})$, where $\|\mathbf{Q}_\gamma\| \leq \beta$ with probability 1, the distribution of $\mathbf{x}_{\gamma,i}$ conditional on $\gamma$ is sub-Gaussian with parameter $K$, and $C$ is a constant.

  Each of the $2M$ events occurs with probability at most $2e^{-\omega}$, from Lemma 4.

- (E4-1, …, E4-2M): For $j = 1, \ldots, M$, $\|\mathbf{X}_{j1}\mathbf{e}_{j1}\| \geq \dfrac{2\xi\phi\omega}{3} + \sqrt{2N\,\mathrm{tr}(\mathbb{E}_\gamma[\sigma_\gamma^2 \mathbf{Q}_\gamma])\omega}$

  and $\|\mathbf{X}_{j2}\mathbf{e}_{j2}\| \geq \dfrac{2\xi\phi\omega}{3} + \sqrt{2N\,\mathrm{tr}(\mathbb{E}_\gamma[\sigma_\gamma^2 \mathbf{Q}_\gamma])\omega}$, where $\|\mathbf{x}_{\gamma,i}\| \leq \xi$ and $|\epsilon_{\gamma,i}| \leq \phi$ with probability 1.

  Each of the $2M$ events occurs with probability at most $2(p+1)e^{-\omega}$, from Lemma 5.

From the union bound, at least one of the events (E1), (E2), (E3-1), …, (E3-2M), (E4-1), …, (E4-2M) occurs with probability at most $2(2p+1+2M+2M(p+1))e^{-\omega}$. That is, none of the events occur with probability at least $1 - 2(2pM + 4M + 2p + 1)e^{-\omega}$.

From (29) and (29), with probability at least $1 - 2(2pM + 4M + 2p + 1)e^{-\omega}$,

$\lambda_{min}(\mathbf{W}\mathbf{W}^{\mathsf{T}})$

$\geq MN\lambda_{min}(\mathbb{E}_\gamma[(\mathbf{I} - \alpha\mathbf{Q}_\gamma)\mathbf{Q}_\gamma(\mathbf{I} - \alpha\mathbf{Q}_\gamma)]) - N(\dfrac{2\beta\mu^2\omega}{3} + \sqrt{2M\,\|\mathrm{Var}_\gamma[\mathbf{S}_\gamma]\|\,\omega})$

$\quad - 2\alpha MN\beta^2\mu C K^2 (\sqrt{\dfrac{p+\omega}{N}} + \dfrac{p+\omega}{N}) - MN\beta\mu^2 C K^2 (\sqrt{\dfrac{p+\omega}{N}} + \dfrac{p+\omega}{N})$

$\quad - 2\alpha MN\beta^2\mu C^2 K^4 (\sqrt{\dfrac{p+\omega}{N}} + \dfrac{p+\omega}{N})^2$

$= MN\Big( \lambda_{min}(\mathbb{E}_\gamma[\mathbf{S}_\gamma]) - (\sqrt{\dfrac{2\,\|\mathrm{Var}_\gamma[\mathbf{S}_\gamma]\|\,\omega}{M}} + \dfrac{2\beta\mu^2\omega}{3M})$

$\quad - (2\alpha\beta^2\mu + \beta\mu^2)C K^2 (\sqrt{\dfrac{p+\omega}{N}} + \dfrac{p+\omega}{N}) - 2\alpha\beta^2\mu C^2 K^4 (\sqrt{\dfrac{p+\omega}{N}} + \dfrac{p+\omega}{N})^2 \Big)$

and

$\Big\| \sum_{j=1}^{M}((\mathbf{I} - \dfrac{\alpha}{N}\mathbf{X}_{j1}\mathbf{X}_{j1}^{\mathsf{T}})\mathbf{X}_{j2}\mathbf{X}_{j2}^{\mathsf{T}}(\mathbf{I} - \dfrac{\alpha}{N}\mathbf{X}_{j1}\mathbf{X}_{j1}^{\mathsf{T}})(\boldsymbol{\theta}_j - \boldsymbol{\theta}_{maml}^*(\alpha))$

$\quad\quad + (\mathbf{I} - \dfrac{\alpha}{N}\mathbf{X}_{j1}\mathbf{X}_{j1}^{\mathsf{T}})(\mathbf{X}_{j2}\mathbf{e}_{j2} - \dfrac{\alpha}{N}\mathbf{X}_{j2}\mathbf{X}_{j2}^{\mathsf{T}}\mathbf{X}_{j1}\mathbf{e}_{j1})) \Big\|$

$\leq MN\Big( (\sqrt{\dfrac{2\,\mathrm{tr}(\mathrm{Var}_\gamma[\mathbf{S}_\gamma\boldsymbol{\theta}_\gamma])\omega}{M}} + \dfrac{2\beta\mu^2\eta\omega}{3M}) + \|\boldsymbol{\theta}_{maml}^*(\alpha)\| (\sqrt{\dfrac{2\,\|\mathrm{Var}_\gamma[\mathbf{S}_\gamma]\|\,\omega}{M}} + \dfrac{2\beta\mu^2\omega}{3M})$

$\quad + (2\alpha\beta + \mu)\tau'\beta\mu C K^2 (\sqrt{\dfrac{p+\omega}{N}} + \dfrac{p+\omega}{N}) + (2\mu + \alpha\beta)\alpha\tau'\beta^2 C^2 K^4 (\sqrt{\dfrac{p+\omega}{N}})^2$

$\quad + (1 + \alpha\beta)\mu(\sqrt{\dfrac{2\,\mathrm{tr}(\mathbb{E}_\gamma[\sigma_\gamma^2\mathbf{Q}_\gamma])\omega}{N}} + \dfrac{2\xi\phi\omega}{3N})$

$\quad + (\mu + 1 + \alpha\beta)\alpha\beta C K^2 \sqrt{\dfrac{p+\omega}{N}}\sqrt{\dfrac{2\,\mathrm{tr}(\mathbb{E}_\gamma[\sigma_\gamma^2\mathbf{Q}_\gamma])\omega}{N}} + o(\dfrac{1}{N}) \Big)$

where $\|\boldsymbol{\theta}_\gamma - \boldsymbol{\theta}_{maml}^*(\alpha)\| \leq \tau'$ with probability 1.

Thus, by (28), if

$$\lambda_{min}(\mathbb{E}_\gamma[\mathbf{S}_\gamma]) - \left(\sqrt{\frac{2\left\|\mathrm{Var}_\gamma[\mathbf{S}_\gamma]\right\|\omega}{M}} + \frac{2\beta\mu^2\omega}{3M}\right)$$
$$- (2\alpha\beta^2\mu + \beta\mu^2)CK^2\left(\sqrt{\frac{p+\omega}{N}} + \frac{p+\omega}{N}\right) - 2\alpha\beta^2\mu C^2K^4\left(\sqrt{\frac{p+\omega}{N}} + \frac{p+\omega}{N}\right)^2 > 0,$$

with probability at least $1 - 2(2pM + 4M + 2p + 1)e^{-\omega}$, $\left\|\hat{\boldsymbol{\theta}}_{maml}(\alpha) - \boldsymbol{\theta}^*_{maml}(\alpha)\right\|$ is bounded above by

$$(\lambda_{min}(\mathbb{E}_\gamma[\mathbf{S}_\gamma]) - o(1))^{-1}\left(\sqrt{\frac{2\,\mathrm{tr}(\mathrm{Var}_\gamma[\mathbf{S}_\gamma\boldsymbol{\theta}_\gamma])\omega}{M}} + \|\boldsymbol{\theta}^*_{maml}(\alpha)\|\sqrt{\frac{2\left\|\mathrm{Var}_\gamma[\mathbf{S}_\gamma]\right\|\omega}{M}}\right.$$
$$+ (2\alpha\beta + \mu)\tau'\beta\mu CK^2\sqrt{\frac{p+\omega}{N}} + (1+\alpha\beta)\mu\sqrt{\frac{2\,\mathrm{tr}(\mathbb{E}_\gamma[\sigma_\gamma^2\mathbf{Q}_\gamma])\omega}{N}} + \frac{2\beta\mu^2\eta\omega}{3M}$$
$$+ \|\boldsymbol{\theta}^*_{maml}(\alpha)\|\frac{2\beta\mu^2\omega}{3M} + ((2\alpha\beta + \mu)\mu + (2\mu + \alpha\beta)\alpha\beta CK^2)\tau'\beta CK^2\frac{p+\omega}{N}$$
$$\left.+ (1+\alpha\beta)\mu\frac{2\xi\phi\omega}{3N} + (\mu + 1 + \alpha + \beta)\alpha\beta CK^2\frac{\sqrt{p+\omega}\sqrt{2\,\mathrm{tr}(\mathbb{E}_\gamma[\sigma_\gamma^2\mathbf{Q}_\gamma])\omega}}{N} + o\left(\frac{1}{N}\right)\right)$$

Setting $\delta = 2(2pM + 4M + 2p + 1)e^{-\omega}$, we obtain $\omega = \ln(2pM + 4M + 2p + 1) - \ln(\delta/2)$.

Thus, if $\lambda_{min}(\mathbb{E}_\gamma[\mathbf{S}_\gamma]) - \tilde{o}(1) > 0$, with probability at least $1 - \delta$, $\left\|\hat{\boldsymbol{\theta}}_{maml}(\alpha) - \boldsymbol{\theta}^*_{maml}(\alpha)\right\|$ is bounded above by

$$(\lambda_{min}(\mathbb{E}_\gamma[\mathbf{S}_\gamma]) - \tilde{o}(1))^{-1}\left(\frac{c_1(\omega, \left\|\mathrm{Var}_\gamma[\mathbf{S}_\gamma]\right\|, \mathrm{tr}(\mathrm{Var}_\gamma[\mathbf{S}_\gamma\boldsymbol{\theta}_\gamma]), \boldsymbol{\theta}^*_{maml}(\alpha))}{\sqrt{M}}\right.$$
$$\left.+ \frac{(2\alpha\beta + \mu)\mu\tau'c_2(\omega) + \sqrt{2}(1+\alpha\beta)\mu c_3(\omega)}{\sqrt{N}} + \tilde{o}\left(\frac{1}{\sqrt{M}}\right) + \tilde{o}\left(\frac{1}{\sqrt{N}}\right)\right)$$

Note that $\|\mathbf{I} - \alpha\mathbf{Q}_\gamma\| \leq 1 + \alpha\beta$, so we can replace $\mu$ in the above bound by $1 + \alpha\beta$ to get the desired result.

$\square$

## C.4   Theorem 5

The precise form of the losses are as follows. For arbitrary $\theta$,

$$\mathbb{E}_\gamma[\mathcal{R}(\boldsymbol{\theta}; \gamma)] = \frac{1}{2}\boldsymbol{\theta}^\intercal\mathbb{E}_\gamma[\mathbf{Q}_\gamma]\boldsymbol{\theta} - \mathbb{E}_\gamma[\mathbf{Q}_\gamma\boldsymbol{\theta}_\gamma]^\intercal\boldsymbol{\theta} + \frac{1}{2}\mathbb{E}_\gamma[\boldsymbol{\theta}_\gamma^\intercal\mathbf{Q}_\gamma\boldsymbol{\theta}_\gamma] + \frac{1}{2}\mathbb{E}_\gamma[\sigma_\gamma^2]$$

$$\mathbb{E}_\gamma[\mathbb{E}_{\mathcal{O}_\gamma}[\mathcal{R}(\tilde{\boldsymbol{\theta}}_\gamma(\alpha); \gamma)]] = \frac{1}{2}\boldsymbol{\theta}^\intercal\mathbb{E}_\gamma[\mathbf{A}_\gamma(\alpha)]\boldsymbol{\theta} - \mathbb{E}_\gamma[\mathbf{A}_\gamma(\alpha)\boldsymbol{\theta}_\gamma]^\intercal\boldsymbol{\theta} + \frac{1}{2}\mathbb{E}_\gamma[\boldsymbol{\theta}_\gamma^\intercal\mathbf{A}_\gamma(\alpha)\boldsymbol{\theta}_\gamma]$$
$$+ \frac{1}{2}\mathbb{E}_\gamma[\sigma_\gamma^2] + \frac{1}{2}\frac{\alpha^2}{N}\mathbb{E}_\gamma[\sigma_\gamma^2\,\mathrm{tr}(\mathbf{Q}_\gamma^2)]$$

$$\text{where}\quad \mathbf{A}_\gamma(\alpha) = (\mathbf{I} - \alpha\mathbf{Q}_\gamma)\mathbf{Q}_\gamma(\mathbf{I} - \alpha\mathbf{Q}_\gamma) + \frac{\alpha^2}{N}(\mathbb{E}[\mathbf{x}_{\gamma,i}\mathbf{x}_{\gamma,i}^\intercal\mathbf{Q}_\gamma\mathbf{x}_{\gamma,i}\mathbf{x}_{\gamma,i}^\intercal] - \mathbf{Q}_\gamma^3)$$

*Proof.* The result for $\mathbb{E}_\gamma[\mathcal{R}(\boldsymbol{\theta}; \gamma)]$ follows from the proof of Equation 10. $\mathbb{E}_\gamma[\mathcal{R}(\boldsymbol{\theta}; \gamma)]$ is minimized by $\boldsymbol{\theta}^*_{drs}$ by definition.

Let $\mathbf{X}_\gamma$ be the $p \times N$ matrix with columns $\mathbf{x}_{\gamma,i}$ and $\mathbf{Y}_\gamma$ be the $N$-dimensional vector with entries $y_{\gamma,i}$. Recalling the definition of $\tilde{\boldsymbol{\theta}}_\gamma(\alpha)$, from (9)

$$\mathcal{R}(\tilde{\boldsymbol{\theta}}_\gamma(\alpha); \gamma) = \mathcal{R}(\boldsymbol{\theta} - \frac{\alpha}{N}(\mathbf{X}_\gamma\mathbf{X}_\gamma^\mathsf{T}\boldsymbol{\theta} - \mathbf{X}_\gamma\mathbf{Y}_\gamma); \gamma) = \mathcal{R}((\mathbf{I} - \frac{\alpha}{N}\mathbf{X}_\gamma\mathbf{X}_\gamma^\mathsf{T})\boldsymbol{\theta} + \frac{\alpha}{N}\mathbf{X}_\gamma\mathbf{Y}_\gamma); \gamma)$$

$$= \frac{1}{2}((\mathbf{I} - \frac{\alpha}{N}\mathbf{X}_\gamma\mathbf{X}_\gamma^\mathsf{T})\boldsymbol{\theta} + \frac{\alpha}{N}\mathbf{X}_\gamma\mathbf{Y}_\gamma))^\mathsf{T}\mathbf{Q}_\gamma((\mathbf{I} - \frac{\alpha}{N}\mathbf{X}_\gamma\mathbf{X}_\gamma^\mathsf{T})\boldsymbol{\theta} + \frac{\alpha}{N}\mathbf{X}_\gamma\mathbf{Y}_\gamma))$$

$$- \boldsymbol{\theta}_\gamma^\mathsf{T}\mathbf{Q}_\gamma((\mathbf{I} - \frac{\alpha}{N}\mathbf{X}_\gamma\mathbf{X}_\gamma^\mathsf{T})\boldsymbol{\theta} + \frac{\alpha}{N}\mathbf{X}_\gamma\mathbf{Y}_\gamma)) + \frac{1}{2}\boldsymbol{\theta}_\gamma^\mathsf{T}\mathbf{Q}_\gamma\boldsymbol{\theta}_\gamma + \frac{1}{2}\sigma_\gamma^2$$

$$= \frac{1}{2}\boldsymbol{\theta}^\mathsf{T}(\mathbf{I} - \frac{\alpha}{N}\mathbf{X}_\gamma\mathbf{X}_\gamma^\mathsf{T})\mathbf{Q}_\gamma(\mathbf{I} - \frac{\alpha}{N}\mathbf{X}_\gamma\mathbf{X}_\gamma^\mathsf{T})\boldsymbol{\theta} + \boldsymbol{\theta}^\mathsf{T}(\mathbf{I} - \frac{\alpha}{N}\mathbf{X}_\gamma\mathbf{X}_\gamma^\mathsf{T})\mathbf{Q}_\gamma\frac{\alpha}{N}\mathbf{X}_\gamma\mathbf{Y}_\gamma$$

$$+ \frac{1}{2}\frac{\alpha^2}{N^2}\mathbf{Y}_\gamma^\mathsf{T}\mathbf{X}_\gamma^\mathsf{T}\mathbf{Q}_\gamma\mathbf{X}_\gamma\mathbf{Y}_\gamma - \boldsymbol{\theta}_\gamma^\mathsf{T}\mathbf{Q}_\gamma(\mathbf{I} - \frac{\alpha}{N}\mathbf{X}_\gamma\mathbf{X}_\gamma^\mathsf{T})\boldsymbol{\theta} - \frac{\alpha}{N}\boldsymbol{\theta}_\gamma^\mathsf{T}\mathbf{Q}_\gamma\mathbf{X}_\gamma\mathbf{Y}_\gamma$$

$$+ \frac{1}{2}\boldsymbol{\theta}_\gamma^\mathsf{T}\mathbf{Q}_\gamma\boldsymbol{\theta}_\gamma + \frac{1}{2}\sigma_\gamma^2$$

First take the expectation with respect to $\mathbf{Y}_\gamma$. Let $\mathbf{e}_\gamma$ be the $N$-dimensional vector with entries $\epsilon_{\gamma,i}$.

$$\mathbb{E}_{\mathbf{Y}_\gamma}[\mathcal{R}(\tilde{\boldsymbol{\theta}}_\gamma(\alpha); \gamma)] = \frac{1}{2}\boldsymbol{\theta}^\mathsf{T}(\mathbf{I} - \frac{\alpha}{N}\mathbf{X}_\gamma\mathbf{X}_\gamma^\mathsf{T})\mathbf{Q}_\gamma(\mathbf{I} - \frac{\alpha}{N}\mathbf{X}_\gamma\mathbf{X}_\gamma^\mathsf{T})\boldsymbol{\theta} + \boldsymbol{\theta}^\mathsf{T}(\mathbf{I} - \frac{\alpha}{N}\mathbf{X}_\gamma\mathbf{X}_\gamma^\mathsf{T})\mathbf{Q}_\gamma\frac{\alpha}{N}\mathbf{X}_\gamma\mathbf{X}_\gamma^\mathsf{T}\boldsymbol{\theta}_\gamma$$

$$+ \frac{1}{2}\frac{\alpha^2}{N^2}\mathbb{E}_{\mathbf{e}_\gamma}[(\boldsymbol{\theta}_\gamma^\mathsf{T}\mathbf{X}_\gamma\mathbf{X}_\gamma^\mathsf{T} + \mathbf{e}_\gamma^\mathsf{T}\mathbf{X}_\gamma^\mathsf{T})\mathbf{Q}_\gamma(\mathbf{X}_\gamma\mathbf{X}_\gamma^\mathsf{T}\boldsymbol{\theta}_\gamma + \mathbf{X}_\gamma\mathbf{e}_\gamma)]$$

$$- \boldsymbol{\theta}_\gamma^\mathsf{T}\mathbf{Q}_\gamma(\mathbf{I} - \frac{\alpha}{N}\mathbf{X}_\gamma\mathbf{X}_\gamma^\mathsf{T})\boldsymbol{\theta} - \frac{\alpha}{N}\boldsymbol{\theta}_\gamma^\mathsf{T}\mathbf{Q}_\gamma\mathbf{X}_\gamma\mathbf{X}_\gamma^\mathsf{T}\boldsymbol{\theta}_\gamma + \frac{1}{2}\boldsymbol{\theta}_\gamma^\mathsf{T}\mathbf{Q}_\gamma\boldsymbol{\theta}_\gamma + \frac{1}{2}\sigma_\gamma^2$$

$$= \frac{1}{2}\boldsymbol{\theta}^\mathsf{T}(\mathbf{I} - \frac{\alpha}{N}\mathbf{X}_\gamma\mathbf{X}_\gamma^\mathsf{T})\mathbf{Q}_\gamma(\mathbf{I} - \frac{\alpha}{N}\mathbf{X}_\gamma\mathbf{X}_\gamma^\mathsf{T})\boldsymbol{\theta}$$

$$- \boldsymbol{\theta}^\mathsf{T}(\mathbf{I} - \frac{\alpha}{N}\mathbf{X}_\gamma\mathbf{X}_\gamma^\mathsf{T})\mathbf{Q}_\gamma(\mathbf{I} - \frac{\alpha}{N}\mathbf{X}_\gamma\mathbf{X}_\gamma^\mathsf{T})\boldsymbol{\theta}_\gamma$$

$$+ \frac{1}{2}\boldsymbol{\theta}_\gamma^\mathsf{T}(\mathbf{I} - \frac{\alpha}{N}\mathbf{X}_\gamma\mathbf{X}_\gamma^\mathsf{T})\mathbf{Q}_\gamma(\mathbf{I} - \frac{\alpha}{N}\mathbf{X}_\gamma\mathbf{X}_\gamma^\mathsf{T})\boldsymbol{\theta}_\gamma + \frac{1}{2}\sigma_\gamma^2 + \frac{1}{2}\frac{\alpha^2}{N^2}\sigma_\gamma^2\,\mathrm{tr}(\mathbf{X}_\gamma\mathbf{X}_\gamma^\mathsf{T}\mathbf{Q}_\gamma)$$

Using the linearity of trace and expectation, as well as the cyclic property of trace,

$$\mathbb{E}_{\mathbf{e}_\gamma}[\mathbf{e}_\gamma^\mathsf{T}\mathbf{X}_\gamma^\mathsf{T}\mathbf{Q}_\gamma\mathbf{X}_\gamma\mathbf{e}_\gamma] = \mathbb{E}_{\mathbf{e}_\gamma}[\mathrm{tr}(\mathbf{X}_\gamma^\mathsf{T}\mathbf{Q}_\gamma\mathbf{X}_\gamma\mathbf{e}_\gamma\mathbf{e}_\gamma^\mathsf{T})] = \mathrm{tr}(\mathbf{X}_\gamma^\mathsf{T}\mathbf{Q}_\gamma\mathbf{X}_\gamma\mathbb{E}_{\mathbf{e}_\gamma}[\mathbf{e}_\gamma\mathbf{e}_\gamma^\mathsf{T}]) = \sigma_\gamma^2\,\mathrm{tr}(\mathbf{X}_\gamma\mathbf{X}_\gamma^\mathsf{T}\mathbf{Q}_\gamma)$$

Using the previous equality,

$$\mathbb{E}_{\mathcal{O}_\gamma}[\mathcal{R}(\tilde{\boldsymbol{\theta}}_\gamma(\alpha); \gamma)] = \mathbb{E}_{\mathbf{X}_\gamma}[\mathbb{E}_{\mathbf{Y}_\gamma}[\mathcal{R}(\tilde{\boldsymbol{\theta}}_\gamma(\alpha); \gamma)]]$$

$$= \frac{1}{2}\boldsymbol{\theta}^\mathsf{T}\mathbb{E}_{\mathbf{X}_\gamma}[(\mathbf{I} - \frac{\alpha}{N}\mathbf{X}_\gamma\mathbf{X}_\gamma^\mathsf{T})\mathbf{Q}_\gamma(\mathbf{I} - \frac{\alpha}{N}\mathbf{X}_\gamma\mathbf{X}_\gamma^\mathsf{T})]\boldsymbol{\theta} - \boldsymbol{\theta}^\mathsf{T}\mathbb{E}_{\mathbf{X}_\gamma}[(\mathbf{I} - \frac{\alpha}{N}\mathbf{X}_\gamma\mathbf{X}_\gamma^\mathsf{T})\mathbf{Q}_\gamma(\mathbf{I} - \frac{\alpha}{N}\mathbf{X}_\gamma\mathbf{X}_\gamma^\mathsf{T})]\boldsymbol{\theta}_\gamma$$

$$+ \frac{1}{2}\boldsymbol{\theta}_\gamma^\mathsf{T}\mathbb{E}_{\mathbf{X}_\gamma}[(\mathbf{I} - \frac{\alpha}{N}\mathbf{X}_\gamma\mathbf{X}_\gamma^\mathsf{T})\mathbf{Q}_\gamma(\mathbf{I} - \frac{\alpha}{N}\mathbf{X}_\gamma\mathbf{X}_\gamma^\mathsf{T})]\boldsymbol{\theta}_\gamma + \frac{1}{2}\sigma_\gamma^2 + \frac{1}{2}\frac{\alpha^2}{N^2}\sigma_\gamma^2\mathbb{E}_{\mathbf{X}_\gamma}[\mathrm{tr}(\mathbf{X}_\gamma\mathbf{X}_\gamma^\mathsf{T}\mathbf{Q}_\gamma)]$$

We compute

$$\mathbb{E}_{\mathbf{X}_\gamma}[(\mathbf{I} - \frac{\alpha}{N}\mathbf{X}_\gamma\mathbf{X}_\gamma^\mathsf{T})\mathbf{Q}_\gamma(\mathbf{I} - \frac{\alpha}{N}\mathbf{X}_\gamma\mathbf{X}_\gamma^\mathsf{T})]$$

$$= \mathbf{Q}_\gamma - 2\alpha\mathbf{Q}_\gamma^2 + \frac{\alpha^2}{N^2}\mathbb{E}_{\mathbf{X}_\gamma}[\sum_{i=1}^{N}\sum_{j=1}^{N}\mathbf{x}_{\gamma,i}\mathbf{x}_{\gamma,i}^\mathsf{T}\mathbf{Q}_\gamma\mathbf{x}_{\gamma,j}\mathbf{x}_{\gamma,j}^\mathsf{T}]$$

$$= \mathbf{Q}_\gamma - 2\alpha\mathbf{Q}_\gamma^2 + \frac{\alpha^2}{N^2}\left(N^2\mathbf{Q}_\gamma^3 + \sum_{i=1}^{N}(\mathbb{E}_{\mathbf{x}_{\gamma,i}}[\mathbf{x}_{\gamma,i}\mathbf{x}_{\gamma,i}^\mathsf{T}\mathbf{Q}_\gamma\mathbf{x}_{\gamma,i}\mathbf{x}_{\gamma,i}^\mathsf{T}] - \mathbf{Q}_\gamma^3)\right)$$

$$= (\mathbf{I} - \alpha\mathbf{Q}_\gamma)\mathbf{Q}_\gamma(\mathbf{I} - \alpha\mathbf{Q}_\gamma) + \frac{\alpha^2}{N}(\mathbb{E}_{\mathbf{x}_{\gamma,i}}[\mathbf{x}_{\gamma,i}\mathbf{x}_{\gamma,i}^\mathsf{T}\mathbf{Q}_\gamma\mathbf{x}_{\gamma,i}\mathbf{x}_{\gamma,i}^\mathsf{T}] - \mathbf{Q}_\gamma^3) \equiv \mathbf{A}_\gamma(\alpha)$$

and

$$\mathbb{E}_{\mathbf{X}_\gamma}[\mathrm{tr}(\mathbf{X}_\gamma\mathbf{X}_\gamma^\mathsf{T}\mathbf{Q}_\gamma)] = \mathrm{tr}(\mathbb{E}_{\mathbf{X}_\gamma}[\mathbf{X}_\gamma\mathbf{X}_\gamma^\mathsf{T}]\mathbf{Q}_\gamma) = N\,\mathrm{tr}(\mathbf{Q}_\gamma^2)$$

Combining the previous two expressions,

$$\mathbb{E}_{\mathcal{O}_\gamma}[\mathcal{R}(\tilde{\boldsymbol{\theta}}_\gamma(\alpha);\gamma)] = \frac{1}{2}\boldsymbol{\theta}^\mathsf{T}\mathbf{A}_\gamma(\alpha)\boldsymbol{\theta} - \boldsymbol{\theta}^\mathsf{T}\mathbf{A}_\gamma(\alpha)\boldsymbol{\theta}_\gamma + \frac{1}{2}\boldsymbol{\theta}_\gamma^\mathsf{T}\mathbf{A}_\gamma(\alpha)\boldsymbol{\theta}_\gamma + \frac{1}{2}\sigma_\gamma^2 + \frac{1}{2}\frac{\alpha^2}{N}\sigma_\gamma^2\operatorname{tr}(\mathbf{Q}_\gamma^2)$$

Taking the expectation with respect to $\gamma$ gives

$$\mathbb{E}_\gamma[\mathbb{E}_{\mathcal{O}_\gamma}[\mathcal{R}(\tilde{\boldsymbol{\theta}}_\gamma(\alpha);\gamma)]] = \frac{1}{2}\boldsymbol{\theta}^\mathsf{T}\mathbb{E}_\gamma[\mathbf{A}_\gamma(\alpha)]\boldsymbol{\theta} - \boldsymbol{\theta}^\mathsf{T}\mathbb{E}_\gamma[\mathbf{A}_\gamma(\alpha)\boldsymbol{\theta}_\gamma] + \frac{1}{2}\mathbb{E}_\gamma[\boldsymbol{\theta}_\gamma^\mathsf{T}\mathbf{A}_\gamma(\alpha)\boldsymbol{\theta}_\gamma]$$
$$+ \frac{1}{2}\sigma_\gamma^2 + \frac{1}{2}\frac{\alpha^2}{N}\mathbb{E}_\gamma[\sigma_\gamma^2\operatorname{tr}(\mathbf{Q}_\gamma^2)]$$

Similar to the derivation of (10), $\mathbb{E}_\gamma[\mathbb{E}_{\mathcal{O}_\gamma}[\mathcal{R}(\tilde{\boldsymbol{\theta}}_\gamma(\alpha);\gamma)]]$ is minimized by $\mathbb{E}_\gamma[\mathbf{A}_\gamma(\alpha)]^{-1}\mathbb{E}_\gamma[\mathbf{A}_\gamma(\alpha)\boldsymbol{\theta}_\gamma]$.

From now on, assume $N \to \infty$. $\mathbf{A}_\gamma(\alpha) \to (\mathbf{I} - \alpha\mathbf{Q}_\gamma)\mathbf{Q}_\gamma(\mathbf{I} - \alpha\mathbf{Q}_\gamma) = \mathbf{S}_\gamma(\alpha)$, so $\mathbb{E}_\gamma[\mathbf{A}_\gamma(\alpha)]^{-1}\mathbb{E}_\gamma[\mathbf{A}_\gamma(\alpha)\boldsymbol{\theta}_\gamma] \to \boldsymbol{\theta}_{maml}^*(\alpha) = \mathbb{E}_\gamma[\mathbf{S}_\gamma(\alpha)]^{-1}\mathbb{E}_\gamma[\mathbf{S}_\gamma(\alpha)\boldsymbol{\theta}_\gamma]$. The expected loss after adaptation of $\boldsymbol{\theta}_{maml}^*(\alpha)$ is given by

$$\frac{1}{2}\boldsymbol{\theta}_{maml}^*(\alpha)^\mathsf{T}\mathbb{E}_\gamma[\mathbf{S}_\gamma(\alpha)]\boldsymbol{\theta}_{maml}^*(\alpha) - \mathbb{E}_\gamma[\mathbf{S}_\gamma(\alpha)\boldsymbol{\theta}_\gamma]^\mathsf{T}\boldsymbol{\theta}_{maml}^*(\alpha) + \frac{1}{2}\mathbb{E}_\gamma[\boldsymbol{\theta}_\gamma^\mathsf{T}\mathbf{S}_\gamma(\alpha)\boldsymbol{\theta}_\gamma]$$
$$+ \frac{1}{2}\mathbb{E}_\gamma[\sigma_\gamma^2]$$
$$= \frac{1}{2}\mathbb{E}_\gamma[\boldsymbol{\theta}_\gamma^\mathsf{T}\mathbf{S}_\gamma(\alpha)\boldsymbol{\theta}_\gamma] - \frac{1}{2}\mathbb{E}_\gamma[\mathbf{S}_\gamma(\alpha)\boldsymbol{\theta}_\gamma]^\mathsf{T}\mathbb{E}_\gamma[\mathbf{S}_\gamma(\alpha)]^{-1}\mathbb{E}_\gamma[\mathbf{S}_\gamma(\alpha)\boldsymbol{\theta}_\gamma] + \frac{1}{2}\mathbb{E}_\gamma[\sigma_\gamma^2] \qquad (30)$$

For $\alpha = 0$, (30) is equal to the expected loss before adaptation of $\boldsymbol{\theta}_{drs}^*$, $\mathbb{E}_\gamma[\mathcal{R}(\boldsymbol{\theta}_{drs}^*;\gamma)]$, i.e. the minimum possible loss before adaptation. Therefore, to show that for $0 < \alpha \le 1/\beta$, the expected loss after adaptation of $\boldsymbol{\theta}_{maml}^*(\alpha)$ is at most $\mathbb{E}_\gamma[\mathcal{R}(\boldsymbol{\theta}_{drs}^*;\gamma)]$, it suffices to show that (30) is nonincreasing in $\alpha$ on $[0, 1/\beta]$. We do so by computing its derivative with respect to $\alpha$ and showing that it is nonpositive on $[0, 1/\beta]$.

Using the chain rule of matrix calculus,

$$\frac{d(30)}{d\alpha} = \alpha\mathbb{E}_\gamma[\boldsymbol{\theta}_\gamma^\mathsf{T}\mathbf{Q}_\gamma^3\boldsymbol{\theta}_\gamma] - \mathbb{E}_\gamma[\boldsymbol{\theta}_\gamma^\mathsf{T}\mathbf{Q}_\gamma^2\boldsymbol{\theta}_\gamma] - \frac{1}{2}\frac{d\mathbb{E}_\gamma[\mathbf{S}_\gamma(\alpha)\boldsymbol{\theta}_\gamma]}{d\alpha}^\mathsf{T}\mathbb{E}_\gamma[\mathbf{S}_\gamma(\alpha)]^{-1}\mathbb{E}_\gamma[\mathbf{S}_\gamma(\alpha)\boldsymbol{\theta}_\gamma]$$
$$- \frac{1}{2}\mathbb{E}_\gamma[\mathbf{S}_\gamma(\alpha)\boldsymbol{\theta}_\gamma]^\mathsf{T}\frac{d\mathbb{E}_\gamma[\mathbf{S}_\gamma(\alpha)]^{-1}}{d\alpha}\mathbb{E}_\gamma[\mathbf{S}_\gamma(\alpha)\boldsymbol{\theta}_\gamma]$$
$$- \frac{1}{2}\mathbb{E}_\gamma[\mathbf{S}_\gamma(\alpha)\boldsymbol{\theta}_\gamma]^\mathsf{T}\mathbb{E}_\gamma[\mathbf{S}_\gamma(\alpha)]^{-1}\frac{d\mathbb{E}_\gamma[\mathbf{S}_\gamma(\alpha)\boldsymbol{\theta}_\gamma]}{d\alpha}$$
$$= -\mathbb{E}_\gamma[\boldsymbol{\theta}_\gamma^\mathsf{T}\mathbf{Q}_\gamma(\mathbf{I} - \alpha\mathbf{Q}_\gamma)\mathbf{Q}_\gamma\boldsymbol{\theta}_\gamma] + \mathbb{E}_\gamma[\mathbf{Q}_\gamma^2\boldsymbol{\theta}_\gamma - \alpha\mathbf{Q}_\gamma^3\boldsymbol{\theta}_\gamma]^\mathsf{T}\mathbb{E}_\gamma[\mathbf{S}_\gamma(\alpha)]^{-1}\mathbb{E}_\gamma[\mathbf{S}_\gamma(\alpha)\boldsymbol{\theta}_\gamma]$$
$$- \mathbb{E}_\gamma[\mathbf{S}_\gamma(\alpha)\boldsymbol{\theta}_\gamma]^\mathsf{T}\mathbb{E}_\gamma[\mathbf{S}_\gamma(\alpha)]^{-1}\mathbb{E}_\gamma[\mathbf{Q}_\gamma^2 - \alpha\mathbf{Q}_\gamma^3]\mathbb{E}_\gamma[\mathbf{S}_\gamma(\alpha)]^{-1}\mathbb{E}_\gamma[\mathbf{S}_\gamma(\alpha)\boldsymbol{\theta}_\gamma]$$
$$+ \mathbb{E}_\gamma[\mathbf{S}_\gamma(\alpha)\boldsymbol{\theta}_\gamma]^\mathsf{T}\mathbb{E}_\gamma[\mathbf{S}_\gamma(\alpha)]^{-1}\mathbb{E}_\gamma[\mathbf{Q}_\gamma^2\boldsymbol{\theta}_\gamma - \alpha\mathbf{Q}_\gamma^3\boldsymbol{\theta}_\gamma]$$
$$= -\mathbb{E}_\gamma[\boldsymbol{\theta}_\gamma^\mathsf{T}\mathbf{Q}_\gamma(\mathbf{I} - \alpha\mathbf{Q}_\gamma)\mathbf{Q}_\gamma\boldsymbol{\theta}_\gamma] + \mathbb{E}_\gamma[\mathbf{Q}_\gamma(\mathbf{I} - \alpha\mathbf{Q}_\gamma)\mathbf{Q}_\gamma\boldsymbol{\theta}_\gamma]^\mathsf{T}\mathbb{E}_\gamma[\mathbf{S}_\gamma(\alpha)]^{-1}\mathbb{E}_\gamma[\mathbf{S}_\gamma(\alpha)\boldsymbol{\theta}_\gamma]$$
$$- \mathbb{E}_\gamma[\mathbf{S}_\gamma(\alpha)\boldsymbol{\theta}_\gamma]^\mathsf{T}\mathbb{E}_\gamma[\mathbf{S}_\gamma(\alpha)]^{-1}\mathbb{E}_\gamma[\mathbf{Q}_\gamma(\mathbf{I} - \alpha\mathbf{Q}_\gamma)\mathbf{Q}_\gamma]\mathbb{E}_\gamma[\mathbf{S}_\gamma(\alpha)]^{-1}\mathbb{E}_\gamma[\mathbf{S}_\gamma(\alpha)\boldsymbol{\theta}_\gamma]$$
$$+ \mathbb{E}_\gamma[\mathbf{S}_\gamma(\alpha)\boldsymbol{\theta}_\gamma]^\mathsf{T}\mathbb{E}_\gamma[\mathbf{S}_\gamma(\alpha)]^{-1}\mathbb{E}_\gamma[\mathbf{Q}_\gamma(\mathbf{I} - \alpha\mathbf{Q}_\gamma)\mathbf{Q}_\gamma\boldsymbol{\theta}_\gamma]$$
$$= -\mathbb{E}_\gamma\left[(\boldsymbol{\theta}_\gamma - \mathbb{E}_\gamma[\mathbf{S}_\gamma(\alpha)]^{-1}\mathbb{E}_\gamma[\mathbf{S}_\gamma(\alpha)\boldsymbol{\theta}_\gamma])^\mathsf{T}\mathbf{Q}_\gamma(\mathbf{I} - \alpha\mathbf{Q}_\gamma)\mathbf{Q}_\gamma(\boldsymbol{\theta}_\gamma - \mathbb{E}_\gamma[\mathbf{S}_\gamma(\alpha)]^{-1}\mathbb{E}_\gamma[\mathbf{S}_\gamma(\alpha)\boldsymbol{\theta}_\gamma])\right]$$

which is the negative of an expectation of a quadratic form with inner matrix $\mathbf{Q}_\gamma(\mathbf{I} - \alpha\mathbf{Q}_\gamma)\mathbf{Q}_\gamma$. When $0 < \alpha \le 1/\beta$, $\mathbf{Q}_\gamma(\mathbf{I} - \alpha\mathbf{Q}_\gamma)\mathbf{Q}_\gamma$ is positive semidefinite for all $\gamma$, and the derivative is nonpositive.

$\square$