[Reviews · NeurIPS 2020]

Review 1

Summary and Contributions: This paper studies theoretical and empirical properties of two opposite methods, i.e. domain randomization and meta-learning, for pre-training of learning parameters in supervised and reinforcement learning. Theoretical results are derived for the case of linear regression and empirical results are obtained on well-known simulated locomotion, and numerical, problems. Results suggest that domain randomization is generally better than meta-learning, as it makes a better use of available sample and does not require the solving of a cumbersome bilevel optimization problem. --- Post rebuttal update --- After reading the rebuttal and other reviews, I still think the contribution of this paper is somewhat limited, but I consider the overall quality slightly above the threshold. Therefore, I'll keep my score of 6.

Strengths: This paper proposes an interesting analysis of the theoretical properties and empirical performance of the opposite methods, for parameters initialization, of domain randomization and meta-learning. Theoretical results are relevant to assess the conditions under which one method should be preferred over the other, and empirical results are obtained on well-known problems, e.g. RL locomotion tasks, whose complexity make the obtained results valuable.

Weaknesses: I think this work has two weaknesses: - The structure and presentation, together with the writing, can be improved. I find the order of sections chaotic, e.g. the alternating of empirical and theoretical results. Anyways, the paper is still readable. - The theoretical results are only obtained for the linear regression case, which I consider a significant limit of this work for being interesting for a general audience. The considered methods for parameters initialization, i.e. domain randomization and meta-learning, are mostly used because of lack of samples when dealing with complex and/or real world problems, that cannot be solved by linear regression methods; thus, I'd expect at least an analysis from the authors about how their results for linear regression may extend to more complex settings.

Correctness: The experimental settings is correct. Although the claims are justified by theorems, I didn't check carefully related proofs.

Clarity: The paper is generally well written, even if some parts could be better explained, e.g. a longer introduction and a better discussion of related works would be desirable.

Relation to Prior Work: Related works could be discussed better. The paper limits to briefly list the main related work, but does not provide further explanation about them and their significance.

Reproducibility: No

Additional Feedback: I suggest the authors to improve the presentation of the paper that can result a bit chaotic, e.g. the structure of the sections alternating empirical and theoretical results. I suggest to better address the analysis w.r.t. deep learning problems, since meta-learning approaches are mostly used in these scenarios. Indeed, I'm not sure about the importance of the results in the linear regression case, and I think this deserves a better discussion from the authors. Moreover, it seems that the take-home message of this work is that domain randomization, when enough samples are used, is a better alternative to meta-learning; however, given that both techniques are used to deal with problems where samples are few and/or costly, I'm not sure whether domain randomization can really be a preferable option w.r.t. meta-learning or not. The authors do not discuss the importance of similarity among the selected tasks, and how it affects the performance of both techniques. I know this is not an easy analysis to carry out, but I'd appreciate a further discussion about this.


Review 2

Summary and Contributions: This paper considers MAML and domain randomized search (DRS) for meta-learning. The former requires solving a bi-level optimization which is computationally expensive, and the latter is a simple empirical risk minimization (single-level) which can be solved efficiently. On the other hand. MAML is able to better leverage the structure among different tasks, while DRS ignore such correlation. The goal of this paper is to characterize the different trade-offs between MAML and DRS, i.e., optimization error, statistical error, and modeling error in meta-reinforcement learning (empirical) and a simple linear regression or meta-linear regression setting (empirical + theoretical). Empirically, it has been observed that DRS is competitive with and often outperforms MAML. In linear regression, while MAML can improve performance through test-time optimization it suffers from large statistical error when fed with small sample sizes.

Strengths: I think this paper investigates an important question on deeper understanding of MAML and DRS methods for meta-learning. However, there are few concerns that prevent me from giving it a high score (see below).

Weaknesses: Theorems 1 and 2 are asymptotic. Also, Assumption 2 assumes that for both DSR and MAML methods, the per-task optimal models are centered around the corresponding optimal solutions. This makes the interpretation of bounds hard. I was expecting a general assumption about tasks that applies for both algorithms and makes the comparison fair, e.g., low-rank or sparsity. Finally, the analysis for a convex squared loss and I am not sure to what extent it could confirm the claimed advantages of DSR compared to MAML. Minor comments: Line 34, (5) should refer to Eq (2) - Eq. 7, \mathcal{N} is missing from \epsilon - Eq. 10, would be better to be consistent in notation and use bold-face for identity matrix - Overall, it seems the paper is prepared in rush and needs a thorough proofreading ************************** Post Rebuttal ********************* Thanks authors for the responses. I read the other reviews, and while still believe the representation of statements can be significantly improved, but believe paper has enough contribution and thus I learn toward accepting the paper.

Correctness: As far as I checked the proofs hold.

Clarity: The paper is reasonably well-written but a through proofreading can make it more readable.

Relation to Prior Work: The claimed contributions are discussed in the light of existing results and the paper does survey related work appropriately.

Reproducibility: Yes

Additional Feedback:


Review 3

Summary and Contributions: This paper provides a comprehensive comparison between gradient-based meta-learning algorithms and domain randomization. The authors identify three sources of error in meta-learning: the modeling error, the statistical error and the optimization error, and show the trade-offs, both theoretically and empirically, between these three sources of error.

Strengths: The authors carefully built their experiments to control the different sources of error, to support their theoretical results. It really shows that the authors had a clear direction, where experimental results are there to prove (empirically) that their theoretical results are sound (as opposed to theory and experiments being disjoint). I also appreciated this overall comparison between meta-learning and domain randomization, which (as far as I know) was missing in the literature. It is interesting that both can fare comparably, especially in RL. It is also very interesting to see DRS as a meta-learning algorithms.

Weaknesses: A. Major concern 1. Lines 132-133: "we suspect that this is due to the reduction in the policy gradient variance resulting from the use of a trust region.". Contrary to what is implied in the submission (e.g. line 105), ProMP is more than just MAML+PPO. ProMP has a variance reduction term in its formulation (LVC vs. DICE formulations in (Rothfuss et al., 2018)). In addition, ProMP has a KL term which acts as a "soft local trust-region". Therefore, this alone cannot justify why MAML combined with TRPO fares better compared to DRS as opposed to ProMP. B. Moderate concern 1. Links (or at least section numbers) to the appropriate sections in the Appendix would have been appreciated. 2. The source-code was not provided as part of the Supplementary Material. C. Minor concern 1. Figures 1, 2, & 3 could really benefit from a diverging colormap (e.g. RdBu, or even better a colorblind alternative). I strongly encourage the authors to change the colormap to make the results clearer. 2. Line 168: the third inequality should read ||\theta_{\gamma} - \theta_{maml}^{*}|| < \tau' (instead \tau). 3. Line 88: "needs O(1/\epsilon^6) points.". Points here meaning "tasks"? This should be more explicit. (Rothfuss et al., 2018) Jonas Rothfuss, Dennis Lee, Ignasi Clavera, Tamim Asfour, Pieter Abbeel. ProMP: Proximal Meta-Policy Search

Correctness: The theoretical result are correct. I have lightly read the proofs provided in the Appendix, and they do not seem to have any major issue. The empirical methodology is correct.

Clarity: The paper is very clearly written.

Relation to Prior Work: The paper clearly discusses its contributions (both theoretical and empirical) within the existing literature. Although I am not sure if this would be relevant to this paper, maybe it could be worth having references to transductive learning (see e.g. (Dhillon et al., 2019) for a comparison with meta-learning), in the context of supervised learning. (Dhillon et al., 2019) Guneet S. Dhillon, Pratik Chaudhari, Avinash Ravichandran, Stefano Soatto. A Baseline for Few-Shot Image Classification

Reproducibility: No

Additional Feedback: After authors response -------------------------- After reading the authors response, I am happy to keep my initial score (7). I would still strongly encourage the authors to consider a different colormap for their plots in the final version of the paper (after seeing the additional plot provided in the authors response).


Review 4

Summary and Contributions: This paper examines the meta-learning performance and behavior of two methods: MAML and domain randomized search (i.e., joint training on the training tasks plus fine-tuning on the test task) with respect to the tradeoff between using a more accurate model (MAML) and using a model that is easier to optimize (DRS). These two methods are analyzed theoretically and empirically for meta-linear regression and empirically for meta-reinforcement learning. In the results on meta-RL, DRS tends to outperform MAML except when the meta-training budget is small and this is attributed to optimization error given the nonconvexity of the problem. In meta-linear regression, the constant factors of the statistical errors of MAML and DRS differ and MAML thus has a worse dependence on N (the number of data points per task / 2) when the curvature is high. However, the modeling error for DRS is worse than that of MAML, since MAML models the meta-structure of the problem, giving rise to the tradeoff. Examining this empirically shows (1) that DRS unsurprisingly outperforms MAML when no adaptation is done for test tasks, and (2) that MAML mostly outperforms DRS after test adaptation but this is not always true for small values of M (number of tasks) and small to medium values of N. Finally, Section 4 shows that, with larger training budgets, DRS will converge to a stationary point of the expected risk, making it a valid meta-learning algorithm given enough data.

Strengths: This paper provides an informative analysis of meta-linear regression and meta-RL using MAML and DRS. The resulting conclusion that DRS is a viable method for meta-learning (given enough training data) is interesting and potentially significant for those doing meta-learning with complex models that are challenging to optimize. Most works which include DRS (i.e., joint training) show that it does noticeably worse than other methods, so it is novel and useful to see problems in which it does well and, better still, to have an understanding for why that is the case.

Weaknesses: My concerns with this work are with its empirical evaluation and related work. In particular, I have concerns about (1) its treatment of the learning rate, \alpha, in the empirical evaluation and discussion, (2) a seeming disagreement between the empirical results presented, and (3) how it situates itself with respect to prior work. I expand on these below.

Correctness: The theoretical claims appear correct, although I did not re-derive the results in full. I have some questions about the empirical results.

Clarity: The paper is quite clear and well written.

Relation to Prior Work: Discussion of related work and situating itself within this context is lacking.

Reproducibility: Yes

Additional Feedback: Main concerns: 1. When trading off between MAML and DRS, the learning rate in MAML provides a simple and explicit mechanism to trade between these two algorithms (ignoring task dataset size). In particular, as the learning rate goes to zero, MAML becomes DRS. Thus, if proper hyperparameter tuning is performed on the learning rate, then I would expect MAML to always out-perform DRS given a reasonable number of samples. (a) I would like to see a discussion of this property and its implications, and perhaps an experiment or analysis showcasing this behavior with respect to \alpha. (b) I am a bit skeptical of the learning rates chosen for the empirical evaluations. In the meta-RL benchmarks, the original MAML paper [1] reports using a learning rate of 0.1 for TRPO-MAML but the largest value evaluated in this work is 0.01. Does that change the performance? [1] reports consistent improvements over joint training (i.e., DRS) in meta-RL. Why is the behavior seen here quite different? (c) Similarly, what happens if \alpha is reduced to 0.1 (or below) in the meta-linear regression experiments? The trend from the plots shown in Figure 3 indicates to me that MAML may more fully dominate DRS if the learning rate is tuned more. Is this true? If so, are the benefits of DRS observed in these experiments due to poor hyperparameters / overfitting instead of the claimed reasons? Are the actual losses recovered by MAML and DRS on the test tasks reasonable or poor? 2. (a) From my understanding of Figures 2, 5, and 6, the lines shown in Figures 5 and 6 should be essentially replicated in Figure 2 when Test Updates = 1. However, looking at the few cases where MAML outperforms DRS in Figures 5 and 6, this is not reflected in Figure 2 (e.g., Figs 5a, 6c,g). Why do these figures not align with each other? (b) In Section 2, DRS does better with more training steps (and thus after seeing more training tasks). However, in Section 3, DRS does better with *fewer* training tasks. However, again in Section 4, it's shown that DRS requires a large training budget to be successful at all. How do you reconcile these differing claims? 3. Joint training plus fine-tuning is commonly used as a baseline in meta-learning and it never does as well as it seems to in this paper (e.g., [1], [2], [3]). What have those works done differently or wrong? What makes this a good approach here? Additional data? (a) Many other works meta-train the embedding layers and then fix those at meta-test time and only update the linear embedding (e.g., [4], [5]). This seems like an interesting middle ground between model accuracy and optimization ease but these are never mentioned in this work. Further, [2] has an analysis in the supplement of MAML vs. joint training for linear regression. How does this differ from your analysis? (b) Similarly, Reptile [6] provides another interesting middle ground between these two ends of the spectrum, especially since it uses all 2N data points. How would Reptile behave in these evaluations? (c) I find it a bit odd to use domain randomization search as the name of this method, as the concept of joint training (plus fine-tuning) predates this particular work. Further, when applied to meta- and multi-task learning, joint training is both a more apt description and the name of a method within these spaces, whereas (while mathematically the same) DRS is defined more narrowly for Sim2Real with environments. Additional questions, concerns, and comments: 4. How do the results here change as the number of inner optimization steps used by MAML increases? This is relevant because most actual applications use more than one step to get good performance, and using more steps with a smaller learning rate should reduce the sensitivity to a large curvature. 5. Is the main reason for the difference in the constants in Theorems 1 and 2 due to the use of 2N data for DRS and only N data in the inner loop for MAML? In practice, MAML tends to benefit from the use of the validation data to prevent overfitting. Does anything related to this appear in your analysis? Minor: - Notationally, (2) and (3) are inconsistent as (2) is missing a \xi in the expectation. - A \mathcal{N} is missing from eq. (7). - It would be easier for the reader if the distinction between g and \gamma were made explicit. - In Theorem 3, S_{\gamma} is actually defined in Theorem 2, not Theorem 1. - The proof on page 18 of the supplement seems to be missing its statement. [1] Model-agnostic meta learning for Fast Adaptation of Deep Networks. Finn, Abbeel, and Levine. [2] Online meta-learning. Finn, Rajeswaran, Kakade, and Levine. [3] Meta-Dataset: A dataset of datasets for learning to learn from few examples. Triantafillou et al. [4] Meta-Learning with Differentiable Convex Optimization. Lee et al. [5] Rethinking Few-Shot Image Classification: a Good Embedding Is All You Need? Tian et al. [6] On First-Order Meta-Learning Algorithms. Nichol, Achiam, and Schulman. ------------------------------------------------- Update after author response: ------------------------------------------------- I am generally satisfied with the responses to my questions and have updated my score to a weak accept. I think there remain some unaddressed subtleties regarding the relationship between these two methods and I encourage the authors to highlight and discuss this in the revision.

[Author Response · NeurIPS 2020]

To all reviewers, thank you very much for your thoughtful comments and suggestions.

**R#1 & R#2: Limitations of the linear regression.** Although linear regression has limited application, it removes the
optimization error (problem has an analytical solution), and enable us to study statistical and modeling trade-off. Our
extensive empirical study on meta-RL complements this and fully describe the trade-off in a more general setting.

**R#1: *"...importance of similarity among the selected tasks..."*** In Theorem 1&2, similarity in the tasks can be described
by $\text{Var}_\gamma(\mathbf{Q}_\gamma)$ and $\text{Var}_\gamma(\mathbf{Q}_\gamma \boldsymbol{\theta}_\gamma)$, implying that greater similarity leads to smaller statistical error.

**R#1: *"...domain randomization, when enough samples are used, is a better alternative to meta-learning..."*** In many
practical deep-RL scenarios (highly nonlinear, low sample-case), we find optimization error dominates and DRS
outperforms MAML. Although Theorem 4 shows that DRS needs some small amount of training data to be effective,
empirical results (Fig 1&2) suggest that the amount of data required is as small as a few rollouts.

**R#2: *"...Theorems 1 and 2 are asymptotic..."*:** Only the first sentence of each theorem is asymptotic, the rest (starting
with "specifically") holds for finite samples. Indeed, asymptotic statements are obtained via the limits of those finite
bounds as $M, N \to \infty$. Hence, the theorems are NOT asymptotic. We will remove the asymptotic parts for clarity.

**R#2: *'Assumption 2 ... the per-task optimal models are centered around the corresponding optimal solutions.'*:** We
do NOT assume that the task optimal models are centered around the DRS/MAML optimal solutions, only that their
distance is bounded. This assumption can easily be dropped with the cost of including the distance as a term.

**R#3: *'...trust region alone cannot justify why ... TRPO fares better...'*** Thanks for this insightful comment. We agree
that the use of a trust region does not fully explain the behavior; we will investigate this and add further discussion.

**R#3: Other points.** We will add the links to the supplement. We will share the source code when the paper is public.

**R#4: *"...as the learning rate goes to zero, MAML becomes DRS..."*** We respectfully disagree that MAML with $\alpha = 0$
is the same as DRS. This statement is only true for the asymptotic objectives (Eq 9) and is not true in practice, when
there are finite samples. MAML uses some data for the inner optimization regardless if the parameters are updated
($\alpha > 0$) or not ($\alpha = 0$). Our main contribution is analyzing MAML and DRS in the practical, finite-sample case.

**R#4: *"...[1] reports consistent improvements over joint training..."*** Our meta-RL experiments are more extensive. [1]
considers locomotion with varying reward, while we consider locomotion and manipulation with varying reward or
dynamics. On the HalfCheetahRandVel environment, the only one that overlaps, our result (Fig 6c) is consistent with
[1] as MAML slightly outperforms DRS. Evaluations on a wider range of environments results in a different conclusion.

**R#4: *"...learning rates..."*** For TRPO-MAML's inner learning rate & trust region size, we use values from [16] for
the locomotion environments (as we use their algorithm implementations) and [25] for the others; DRS+TRPO's trust
region size was set to be the same as TRPO-MAML. The comparison is fair and follows practice of previous work.

**R#4: *"...Joint training plus fine-tuning ... never does as well as it seems to in this paper..."*** Our contribution is to
introduce and discuss the trade-off between DRS and MAML. Existing literature compare them on a single point within
this trade-off, whereas we explore the entire spectrum. Hence, those works happen to experiment on one side of the
spectrum. Our paper does not conflict with existing works, it complements them by considering a larger picture.

**R#4: *"... Figures 2, 5, and 6... Why do these figures not align with each other?..."*** The one-sided Welch t-test obtains
an estimate of the distribution of the difference between the average rewards of DRS and MAML. The variance of that
distribution combines the variances of the two average rewards and is fairly large. Thus, even if the average reward for
MAML is visually above that of DRS (Figure 5), the probability that DRS is better (Figure 2) may not be small. In
other words, the difference in Figure 5 is not statistically significant considering the variances.

**R#4: *"In Section 2, DRS does better with more training steps...Section 3, DRS does better with \*fewer\* training*
*tasks..."*** In Section 3, the optimization error is 0 for both methods (problem has analytical solution), and only the
statistical and modeling error matter. MAML has smaller modeling error and so it is better with more data. In Section 2,
the optimization error is significantly larger for MAML because of its bilevel structure; it dominates the trade-off for a
wide range of training budgets. The difference between these behaviors is one of the main points of this paper. We
show that the trade-off between DRS and MAML is not straightforward and requires careful analysis.

**R#4: *"...it's shown that DRS requires a large training budget to be successful...*** We showed that DRS needs some
data (can be very small). Results in Figures 5&6 suggest that this budget is very small (only a few rollouts).

**R#4: *"...[2] has an analysis..."*** For meta-linear regression, [2] derive the MAML and DRS estimates when the task-
specific losses are known. In contrast, we consider the practical scenario where the task-specific losses are not known,
analyzing the estimates, their finite-sample behaviors, and their meta-test performances. We will add a discussion.

**R#4: *"... if $\alpha$ is reduced to 0.1 (or below) ..."*** The adjacent figure
shows meta-linear regression results for $\alpha = 0.05, 0.1$. The same
conclusions hold, but the difference between MAML and DRS is
not as pronounced; MAML requires a larger data set to provide a
clear improvement over DRS after test-time optimization.

Meta-Learning is Better — Domain Randomization is Better

$\alpha = 0.05$, pre $\quad \alpha = 0.1$, pre $\quad \alpha = 0.05$, post $\quad \alpha = 0.1$, post

[Meta-Review · NeurIPS 2020]

This paper addresses a trade-off between MAML and DRS which are two opposite methods. Authors did a good job in the rebuttal which well answered most of reviewers’ concerns. Two of reviewers raised their scores to 6. The paper well investigates an important question on deeper understanding of MAML and DRS for meta-learning.